# Error Feedback Reloaded: From Quadratic to Arithmetic Mean of Smoothness Constants

**Peter Richtárik**
AI Initiative
KAUST,* Saudi Arabia

**Elnur Gasanov**
AI Initiative
KAUST, Saudi Arabia

**Konstantin Burlachenko**
AI Initiative
KAUST, Saudi Arabia

## Abstract

Error Feedback (EF) is a highly popular and immensely effective mechanism for fixing convergence issues which arise in distributed training methods (such as distributed GD or SGD) when these are enhanced with greedy communication compression techniques such as TopK. While EF was proposed almost a decade ago (Seide et al., 2014), and despite concentrated effort by the community to advance the theoretical understanding of this mechanism, there is still a lot to explore. In this work we study a modern form of error feedback called EF21 (Richtárik et al., 2021) which offers the currently best-known theoretical guarantees, under the weakest assumptions, and also works well in practice. In particular, while the theoretical communication complexity of EF21 depends on the *quadratic mean* of certain smoothness parameters, we improve this dependence to their *arithmetic mean*, which is always smaller, and can be substantially smaller, especially in heterogeneous data regimes. We take the reader on a journey of our discovery process. Starting with the idea of applying EF21 to an equivalent reformulation of the underlying problem which (unfortunately) requires (often impractical) machine *cloning*, we continue to the discovery of a new *weighted* version of EF21 which can (fortunately) be executed without any cloning, and finally circle back to an improved *analysis* of the original EF21 method. While this development applies to the simplest form of EF21, our approach naturally extends to more elaborate variants involving stochastic gradients and partial participation. Further, our technique improves the best-known theory of EF21 in the *rare features* regime (Richtárik et al., 2023). Finally, we validate our theoretical findings with suitable experiments.

## 1 Introduction

Due to their ability to harness the computational capabilities of modern devices and their capacity to extract value from the enormous data generated by organizations, individuals, and various digital devices and sensors, Machine Learning (ML) methods (Bishop, 2016; Shalev-Shwartz & Ben-David, 2014) have become indispensable in numerous practical applications (Krizhevsky et al., 2012; Lin et al., 2022; Vaswani et al., 2017; Onay & Öztürk, 2018; Poplin et al., 2017; Gavriluţ et al., 2009; Sun et al., 2017).

The necessity to handle large datasets has driven application entities to store and process their data in powerful computing centers (Yang et al., 2019; Dean et al., 2012; Verbraeken et al., 2020) via *distributed* training algorithms. Beside this industry-standard centralized approach, decentralized forms of distributed learning are becoming increasingly popular. For example, Federated Learning (FL) facilitates a collaborative learning process in which various clients, such as hospitals or owners of edge devices, collectively train a model

---

*King Abdullah University of Science and Technology

on their devices while retaining their data locally, without uploading it to a centralized location (Konečný et al., 2016b;a; McMahan et al., 2017; Li et al., 2020a; Kairouz et al., 2021; Wang et al., 2021).

Distributed training problems are typically formulated as optimization problems of the form

$$\min_{x \in \mathbb{R}^d} \left\{ f(x) \coloneqq \frac{1}{n} \sum_{i=1}^{n} f_i(x) \right\}, \tag{1}$$

where $n$ is the number of clients/workers/nodes, vector $x \in \mathbb{R}^d$ represents the $d$ trainable parameters, and $f_i(x)$ is the loss of the model parameterized by $x$ on the training data stored on client $i \in [n] \coloneqq \{1, \ldots, n\}$. One of the key issues in distributed training in general, and FL in particular, is the *communication bottleneck* (Konečný et al., 2016b; Kairouz et al., 2021). The overall efficiency of a distributed algorithm for solving (1) can be characterized by multiplying the number of communication rounds needed to find a solution of acceptable accuracy by the cost of each communication round:

communication complexity = # communication rounds × cost of 1 communication round. (2)

This simple formula clarifies the rationale behind two orthogonal approaches to alleviating the communication bottleneck. i) The first approach aims to minimize the first factor in (2). This is done by carefully deciding on *what work should be done* on the clients in each communication round in order for it to reduce the total number of communication rounds needed, and includes methods based on local training (Stich, 2018; Lin et al., 2018; Mishchenko et al., 2022; Condat et al., 2023; Li et al., 2019) and momentum (Nesterov, 1983; 2004; d'Aspremont et al., 2021). Methods in this class communicate dense $d$-dimensional vectors. ii) The second approach aims to minimize the second factor in (2). Methods in this category *compress the information* (typically $d$-dimensional vectors) transmitted between the clients and the server (Alistarh et al., 2017; Khirirat et al., 2018; Bernstein et al., 2018; Safaryan et al., 2021).

## 1.1 COMMUNICATION COMPRESSION

Vector compression can be achieved through the application of a compression operator. Below, we outline two primary classes of these operators: unbiased (with conically bounded variance) and contractive.

**Definition 1** (Compressors). A randomized mapping $\mathcal{C} : \mathbb{R}^d \to \mathbb{R}^d$ is called i) an *unbiased compressor* if for some $\omega > 0$ it satisfies

$$\mathbb{E}\left[\mathcal{C}(x)\right] = x, \quad \mathbb{E}\left[\|\mathcal{C}(x) - x\|^2\right] \le \omega\|x\|^2, \quad \forall x \in \mathbb{R}^d, \tag{3}$$

and ii) a *contractive compressor* if for some $\alpha \in (0, 1]$ it satisfies

$$\mathbb{E}\left[\|\mathcal{C}(x) - x\|^2\right] \le (1 - \alpha)\|x\|^2, \quad \forall x \in \mathbb{R}^d. \tag{4}$$

It is well known that whenever a compressor $\mathcal{C}$ satisfies (3), then the scaled compressor $\mathcal{C}/(\omega + 1)$ satisfies (4) with $\alpha = 1 - (\omega + 1)^{-1}$. In this sense, the class of contractive compressors includes all unbiased compressors as well. However, it is also strictly larger. For example, the Top$K$ compressor, which retains the $K$ largest elements in absolute value of the vector it is applied to and replaces the rest by zeros, and happens to be very powerful in practice (Alistarh et al., 2018), satisfies (4) with $\alpha = \frac{K}{d}$, but does not satisfy (3). From now on, we write $\mathbb{C}(\alpha)$ to denote the class of compressors satisfying (4).

It will be convenient to define the following functions of the contraction parameter

$$\theta = \theta(\alpha) \coloneqq 1 - \sqrt{1 - \alpha}; \quad \beta = \beta(\alpha) \coloneqq \frac{1 - \alpha}{1 - \sqrt{1 - \alpha}}; \quad \xi = \xi(\alpha) \coloneqq \sqrt{\frac{\beta(\alpha)}{\theta(\alpha)}} = \frac{1 + \sqrt{1 - \alpha}}{\alpha} - 1. \tag{5}$$

Note that $0 \le \xi(\alpha) < \frac{2}{\alpha} - 1$. The behavior of distributed algorithms utilizing unbiased compressors for solving (1) is relatively well-understood from a theoretical standpoint (Khirirat et al., 2018; Mishchenko et al., 2019; Li et al., 2020b; Gorbunov et al., 2021; Tyurin & Richtárik, 2023). By now, the community possesses a robust theoretical understanding of the advantages such methods can offer and the mechanisms behind their efficacy (Gorbunov et al., 2020; Khaled et al., 2023; Tyurin & Richtárik, 2023). However, it is well known that the class of contractive compressors includes some practically very powerful operators, such as the greedy sparsifier Top$K$ (Stich et al., 2018; Alistarh et al., 2018) and the low-rank approximator Rank$K$

---

**Algorithm 1** EF21: Error Feedback 2021

---

1: **Input:** initial model $x^0 \in \mathbb{R}^d$; initial gradient estimates $g_1^0, g_2^0, \ldots, g_n^0 \in \mathbb{R}^d$ stored at the server and the clients; stepsize $\gamma > 0$; number of iterations $T > 0$
2: **Initialize:** $g^0 = \frac{1}{n} \sum_{i=1}^{n} g_i^0$ on the server
3: **for** $t = 0, 1, 2, \ldots, T - 1$ **do**
4:     Server computes $x^{t+1} = x^t - \gamma g^t$ and broadcasts $x^{t+1}$ to all $n$ clients
5:     **for** $i = 1, \ldots, n$ **on the clients in parallel do**
6:         Compute $u_i^t = \mathcal{C}_i^t(\nabla f_i(x^{t+1}) - g_i^t)$ and update $g_i^{t+1} = g_i^t + u_i^t$
7:         Send the compressed message $u_i^t$ to the server
8:     **end for**
9:     Server updates $g_i^{t+1} = g_i^t + u_i^t$ for all $i \in [n]$, and computes $g^{t+1} = \frac{1}{n} \sum_{i=1}^{n} g_i^{t+1}$
10: **end for**
11: **Output:** Point $\hat{x}^T$ chosen from the set $\{x^0, \ldots, x^{T-1}\}$ uniformly at random

---

(Vogels et al., 2019; Safaryan et al., 2022), which are biased, and hence their behavior is not explainable by the above developments. These compressors have demonstrated surprisingly effective performance in practice (Seide et al., 2014; Alistarh et al., 2018), even when compared to the best results we can get with unbiased compressors (Szlendak et al., 2022), and are indispensable on difficult tasks such as the fine-tuning of foundation models in a geographically distributed manner over slow networks (Wang et al., 2023).

However, our theoretical understanding of algorithms based on contractive compressors in general, and these powerful biased compressors in particular, is very weak. Indeed, while the SOTA theory involving unbiased compressors offers significant and often several-degrees-of-magnitude improvements over the baseline methods that do not use compression (Mishchenko et al., 2019; Horváth et al., 2019b; Li et al., 2020b; Gorbunov et al., 2020; 2021; Tyurin & Richtárik, 2023), the best theory we currently have for methods that can provably work with contractive compressors, i.e., the theory behind the error feedback method called EF21 developed by Richtárik et al. (2021) (see Algorithm 1) and its variants (Fatkhullin et al., 2021; Condat et al., 2022; Fatkhullin et al., 2023), merely matches the communication complexity of the underlying methods that do not use any compression (Szlendak et al., 2022).

To the best of our knowledge, the only exception to this is the very recent work of Richtárik et al. (2023) showing that in a *rare features* regime, the EF21 method (Richtárik et al., 2021) outperforms gradient descent (which is a special case of EF21 when $\mathcal{C}_i^t(x) \equiv x$ for all $i \in [n]$ and $t \geq 0$) in theory. However, Richtárik et al. (2023) obtain no improvements upon the current best theoretical result for vanilla EF21 (Richtárik et al., 2021) in the general smooth nonconvex regime, outlined in Section 1.2, we investigate in this work.

## 1.2 Assumptions

We adopt the same very weak assumptions as those used by Richtárik et al. (2021) in their analysis of EF21.
**Assumption 1.** *The function $f$ is $L$-smooth, i.e., there exists $L > 0$ such that*
$$\|\nabla f(x) - \nabla f(y)\| \leq L \|x - y\|, \quad \forall x, y \in \mathbb{R}^d. \tag{6}$$
**Assumption 2.** *The functions $f_i$ are $L_i$-smooth, i.e., for all $i \in [n]$ there exists $L_i > 0$ such that*
$$\|\nabla f_i(x) - \nabla f_i(y)\| \leq L_i \|x - y\|, \quad \forall x, y \in \mathbb{R}^d. \tag{7}$$
Note that if (7) holds, then (6) holds, and $L \leq L_{\text{AM}} := \frac{1}{n} \sum_{i=1}^{n} L_i$. So, Assumption 1 does *not* further limit the class of functions already covered by Assumption 2. Indeed, it merely provides a new parameter $L$ better characterizing the smoothness of $f$ than the estimate $L_{\text{AM}}$ obtainable from Assumption 2 could.

Since our goal in (1) is to minimize $f$, the below assumption is necessary for the problem to be meaningful.
**Assumption 3.** *There exists $f^* \in \mathbb{R}$ such that $\inf f \geq f^*$.*

### 1.3 SUMMARY OF CONTRIBUTIONS

In our work we improve the current SOTA theoretical communication complexity guarantees for distributed algorithms that work with contractive compressors in general, and empirically powerful biased compressors such as Top$K$ and Rank$K$ in particular (Richtárik et al., 2021; Fatkhullin et al., 2021).

In particular, under Assumptions 1–3, the best known guarantees were obtained by Richtárik et al. (2021) for the EF21 method: to find a (random) vector $\hat{x}^T$ satisfying $\mathbb{E}\left[\left\|\nabla f(\hat{x}^T)\right\|^2\right] \leq \varepsilon$, Algorithm 1 requires

$$T = \mathcal{O}\left((L + L_{\mathrm{QM}}\xi(\alpha))\varepsilon^{-1}\right)$$

iterations, where $L_{\mathrm{QM}} \coloneqq \sqrt{\frac{1}{n}\sum_{i=1}^n L_i^2}$ is the Quadratic Mean of the smoothness constants $L_1, \ldots, L_n$. Our main finding is an improvement of this result to

$$T = \mathcal{O}\left((L + L_{\mathrm{AM}}\xi(\alpha))\varepsilon^{-1}\right), \tag{8}$$

where $L_{\mathrm{AM}} \coloneqq \frac{1}{n}\sum_{i=1}^n L_i$ is the Arithmetic Mean of the smoothness constants $L_1, \ldots, L_n$. We obtain this improvement in *three different ways:*

i) by *client cloning* (see Sections 2.1 and 2.2 and Theorem 2),

ii) by proposing a new *smoothness-weighted* variant of EF21 which we call EF21-W (see Section 2.3 and Theorem 3), and

iii) by a new *smoothness-weighted* analysis of classical EF21 (see Section 2.4 and Theorem 4).

We obtain refined linear convergence results cases under the Polyak-Łojasiewicz condition. Further, our analysis technique extends to many variants of EF21, including EF21-SGD which uses stochastic gradients instead of gradients (Section E), and EF21-PP which enables partial participation of clients (Section G). Our analysis also improves upon the results of Richtárik et al. (2023) who study EF21 in the *rare features* regime (Section H). Finally, we validate our theory with suitable computational experiments (Sections 3, I and J).

## 2 EF21 RELOADED: OUR DISCOVERY STORY

We now take the reader along on a ride of our discovery process.

### 2.1 STEP 1: CLONING THE CLIENT WITH THE WORSE SMOOTHNESS CONSTANT

The starting point of our journey is a simple observation described in the following example.

**Example 1.** *Let $n = 4$ and $f(x) = \frac{1}{4}(f_1(x) + f_2(x) + f_3(x) + f_4(x))$. Assume the smoothness constants $L_1, L_2, L_3$ of $f_1, f_2, f_3$ are equal to $1$, and $L_4$ is equal to $100$. In this case,EF21 needs to run for*

$$T_1 \coloneqq \mathcal{O}\left((L + L_{\mathrm{QM}}\xi(\alpha))\varepsilon^{-1}\right) = \mathcal{O}\left((L + \sqrt{2501.5}\xi(\alpha))\varepsilon^{-1}\right)$$

*iterations. Now, envision the existence of an additional machine capable of downloading the data from the fourth "problematic" machine. By rescaling local loss functions, we maintain the overall loss function as:*

$$f(x) = \frac{1}{4}(f_1(x) + f_2(x) + f_3(x) + f_4(x)) = \frac{1}{5}\left(\frac{5}{4}f_1(x) + \frac{5}{4}f_2(x) + \frac{5}{4}f_3(x) + \frac{5}{8}f_4(x) + \frac{5}{8}f_4(x)\right) \coloneqq \tilde{f}(x).$$

*Rescaling of the functions modifies the smoothness constants to $\hat{L}_i = \frac{5}{4}L_i$ for $i = 1, 2, 3$, and $\hat{L}_i = \frac{5}{8}L_4$ for $i = 4, 5$. EF21, launched on this setting of five nodes, requires*

$$T_2 \coloneqq \mathcal{O}\left(\left(L + \tilde{L}_{\mathrm{QM}}\xi(\alpha)\right)\varepsilon^{-1}\right) \approx \mathcal{O}\left((L + \sqrt{1564}\xi(\alpha))\varepsilon^{-1}\right)$$

*iterations, where $\tilde{L}_{\mathrm{QM}}$ is the quadratic mean of the new smoothness constants $\hat{L}_1, \ldots, \hat{L}_5$.*

This simple observation highlights that the addition of just one more client significantly enhances the convergence rate. Indeed, EF21 requires approximately $\frac{\xi(\alpha)}{\varepsilon}(\sqrt{2501.5} - \sqrt{1564}) \approx 10\frac{\xi(\alpha)}{\varepsilon}$ fewer iterations. We will generalize this client cloning idea in the next section.

## 2.2 STEP 2: GENERALIZING THE CLONING IDEA

We will now take the above motivating example further, allowing each client $i$ to be cloned arbitrarily many ($N_i$) times. Let us see where this gets us. For each $i \in [n]$, let $N_i$ denote a positive integer. We define $N := \sum_{i=1}^n N_i$ (the total number of clients after cloning), and observe that $f$ can be equivalently written as

$$f(x) \overset{(1)}{=} \tfrac{1}{n}\sum_{i=1}^n f_i(x) = \tfrac{1}{n}\sum_{i=1}^n \sum_{j=1}^{N_i} \tfrac{1}{N_i} f_i(x) = \tfrac{1}{N}\sum_{i=1}^n \sum_{j=1}^{N_i} \tfrac{N}{nN_i} f_i(x) = \tfrac{1}{N}\sum_{i=1}^n \sum_{j=1}^{N_i} f_{ij}(x), \tag{9}$$

where $f_{ij}(x) := \frac{N}{nN_i} f_i(x)$ for all $i \in [n]$ and $j \in [N_i]$. Notice that we scaled the functions as before, and that $f_{ij}$ is $L_{ij}$-smooth, where $L_{ij} := \frac{N}{nN_i} L_i$.

**Analysis of the convergence rate.** The performance of EF21, when applied to the problem (9) involving $N$ clients, depends on the quadratic mean of the new smoothness constants:

$$M(N_1, \ldots, N_n) := \sqrt{\tfrac{1}{N}\sum_{i=1}^n \sum_{j=1}^{N_i} L_{ij}^2} = \sqrt{\sum_{i=1}^n \tfrac{N}{n^2 N_i} L_i^2} = \tfrac{1}{n}\sqrt{\sum_{i=1}^n \tfrac{L_i^2}{N_i/N}}. \tag{10}$$

Note that if $N_i = 1$ for all $i \in [n]$, then $M(1, \ldots, 1) = L_{\mathrm{QM}}$.

**Optimal choice of cloning frequencies.** Our goal is to find integer values $N_1 \in \mathbb{N}, \ldots, N_n \in \mathbb{N}$ minimizing the function $M(N_1, \ldots, M_n)$ defined in (10). While we do not have a closed-form formula for the global minimizer, we are able to explicitly find a solution that is at most $\sqrt{2}$ times worse than the optimal one in terms of the objective value. In particular, if we let $N_i^\star = \lceil L_i/L_{\mathrm{AM}} \rceil$ for all $i \in [n]$, then

$$L_{\mathrm{AM}} \le \min_{N_1 \in \mathbb{N}, \ldots, N_n \in \mathbb{N}} M(N_1, \ldots, N_n) \le M(N_1^\star, \ldots, N_n^\star) \le \sqrt{2} L_{\mathrm{AM}},$$

and moreover, $n \le N^\star := \sum_i N_i^\star \le 2n$. That is, we need at most double the number of clients in our client cloning construction. See Lemma 2 in the Appendix for details.

By directly applying EF21 theory from (Richtárik et al., 2021) to problem (9) involving $N^\star$ clients, we obtain the advertised improvement from $L_{\mathrm{QM}}$ to $L_{\mathrm{AM}}$.

**Theorem 2** (**Convergence of** EF21 **applied to problem** (9) **with** $N^\star$ **machines**)**.** *Consider Algorithm 1 (*EF21*) applied to the "cloning reformulation" (9) of the distributed optimization problem (1), where $N_i^\star = \lceil L_i/L_{\mathrm{AM}} \rceil$ for all $i \in [n]$. Let Assumptions 1–3 hold, assume that $\mathcal{C}_{ij}^t \in \mathbb{C}(\alpha)$ for all $i \in [n]$, $j \in [N_i]$ and $t \ge 0$, set*

$$G^t := \tfrac{1}{N}\sum_{i=1}^N \sum_{j=1}^{N_i} \left\| g_{ij}^t - \nabla f_{ij}(x^t) \right\|^2,$$

*and let the stepsize satisfy $0 < \gamma \le \frac{1}{L + \sqrt{2}L_{\mathrm{AM}}\xi(\alpha)}$. If for $T \ge 1$ we define $\hat{x}^T$ as an element of the set $\{x^0, x^1, \ldots, x^{T-1}\}$ chosen uniformly at random, then*

$$\mathbb{E}\left[\left\| \nabla f(\hat{x}^T) \right\|^2\right] \le \tfrac{2(f(x^0) - f^*)}{\gamma T} + \tfrac{G^0}{\theta(\alpha)T}.$$

When we choose the largest allowed stepsize and $g_{ij}^0 = \nabla f_{ij}(x^0)$ for all $i, j$, this leads to the complexity (8); that is, by cloning client machines, we can replace $L_{\mathrm{QM}}$ in the standard rate with $\sqrt{2}L_{\mathrm{AM}}$. A similar result can be obtained even if we do not ignore the integrality constraint, but we do not include it for brevity

reasons. However, it is important to note that the cloning approach has several straightforward shortcomings, which we will address in the next section.[1]

## 2.3 STEP 3: FROM CLIENT CLONING TO UPDATE WEIGHTING

It is evident that employing client cloning improves the convergence rate. Nevertheless, there are obvious drawbacks associated with this approach. Firstly, it necessitates a larger number of computational devices, rendering its implementation less appealing from a resource allocation perspective. Secondly, the utilization of EF21 with cloned machines results in a departure from the principles of Federated Learning, as it inherently compromises user privacy – transferring data from one device to another is prohibited in FL.

However, a simpler approach to implementing the cloning idea emerges when we assume the compressors used to be *deterministic*. To illustrate this, let us initially examine how we would typically implement EF21 with cloned machines:

$$x^{t+1} = x^t - \gamma \frac{1}{N} \sum_{i=1}^{n} \sum_{j=1}^{N_i} g_{ij}^t, \tag{11}$$

$$g_{ij}^{t+1} = g_{ij}^t + \mathcal{C}_{ij}^t(\nabla f_{ij}(x^{t+1}) - g_{ij}^t), \quad i \in [n], \quad j \in [N_i]. \tag{12}$$

We will now rewrite the same method in a different way. Assume we choose $g_{ij}^0 = g_i^0$ for all $j \in [N_i]$. We show by induction that $g_{ij}^t$ is the same for all $j \in [N_i]$. We have just seen that this holds for $t = 0$. Assume this holds for some $t$. Then since $\nabla f_{ij}(x^{t+1}) = \frac{N}{nN_i} \nabla f_i(x^{t+1})$ for all $j \in [N_i]$ combined with the induction hypothesis, (12) and the determinism of $\mathcal{C}_{ij}^t$, we see that $g_{ij}^{t+1}$ is the same for all $j \in [N_i]$. Let us define $g_i^t \equiv g_{ij}^t$ for all $t$. This is a valid definition since we have shown that $g_{ij}^t$ does not depend on $j$. Because of all of the above, iterations (11)–(12) can be equivalently written in the form

$$x^{t+1} = x^t - \gamma \sum_{i=1}^{n} w_i g_i^t, \tag{13}$$

$$g_i^{t+1} = g_i^t + \mathcal{C}_i^t \left( \frac{1}{nw_i} \nabla f_i(x^{t+1}) - g_i^t \right), \quad i \in [n], \tag{14}$$

where $w_i = \frac{L_i}{\sum_j L_j}$. This transformation effectively enables us to operate the method on the original $n$ clients, eliminating the need for $N$ clients! This refinement has led to the creation of a new algorithm that outperforms EF21 in terms of convergence rate, which we call EF21-W (Algorithm 2). While we relied on assuming that the compressors are *deterministic* in order to motivate the transition from $N$ to $n$ clients, it turns out that EF21-W converges without the need to invoke this assumption.

**Theorem 3** (**Theory for** EF21-W). *Consider Algorithm 2 (EF21-W) applied to the distributed optimization problem* (1). *Let Assumptions 1–3 hold, assume that $\mathcal{C}_i^t \in \mathbb{C}(\alpha)$ for all $i \in [n]$ and $t \geq 0$, set*

$$G^t := \sum_{i=1}^{n} w_i \left\| g_i^t - \frac{1}{nw_i} \nabla f_i(x^t) \right\|^2,$$

---

[1]In our work, we address an optimization problem of the form $\min_{w_j \geq 0 \ \forall j \in [n]; \sum_{i=1}^{n} w_i = 1} \sum_{i=1}^{n} \frac{a_i^2}{w_i}$, where $a_i$ represent certain constants. This formulation bears a resemblance to the meta problem in the importance sampling strategy discussed in (Zhao & Zhang, 2015). Despite the apparent similarities in the abstract formulation, our approach and the one in the referenced work diverge significantly in both motivation and implementation. While Zhao & Zhang (2015) applies importance sampling to reduce the variance of a stochastic gradient estimator by adjusting sampling probabilities, our method involves adjusting client cloning weights without sampling. Furthermore, our gradient estimator is biased, unlike the unbiased estimator in the referenced paper, and we aim to minimize the quadratic mean of the smoothness constants, which is inherently different from the objectives in Zhao & Zhang (2015). Although both approaches can be expressed through a similar mathematical framework, they are employed in vastly different contexts, and any parallelism may be coincidental rather than indicative of a direct connection.

---

**Algorithm 2** EF21-W: Weighted Error Feedback 2021

---

1: **Input:** initial model parameters $x^0 \in \mathbb{R}^d$; initial gradient estimates $g_1^0, g_2^0, \ldots, g_n^0 \in \mathbb{R}^d$ stored at the server and the clients; weights $w_i = L_i / \sum_j L_j$; stepsize $\gamma > 0$; number of iterations $T > 0$
2: **Initialize:** $g^0 = \sum_{i=1}^n w_i g_i^0$ on the server
3: **for** $t = 0, 1, 2, \ldots, T-1$ **do**
4:     Server computes $x^{t+1} = x^t - \gamma g^t$ and broadcasts $x^{t+1}$ to all $n$ clients
5:     **for** $i = 1, \ldots, n$ **on the clients in parallel do**
6:         Compute $u_i^t = \mathcal{C}_i^t(\frac{1}{nw_i}\nabla f_i(x^{t+1}) - g_i^t)$ and update $g_i^{t+1} = g_i^t + u_i^t$
7:         Send the compressed message $u_i^t$ to the server
8:     **end for**
9:     Server updates $g_i^{t+1} = g_i^t + u_i^t$ for all $i \in [n]$, and computes $g^{t+1} = \sum_{i=1}^n w_i g_i^{t+1}$
10: **end for**
11: **Output:** Point $\hat{x}^T$ chosen from the set $\{x^0, \ldots, x^{T-1}\}$ uniformly at random

---

*where $w_i = \frac{L_i}{\sum_j L_j}$ for all $i \in [n]$, and let the stepsize satisfy $0 < \gamma \leq \frac{1}{L+L_{\text{AM}}\xi(\alpha)}$. If for $T > 1$ we define $\hat{x}^T$ as an element of the set $\{x^0, x^1, \ldots, x^{T-1}\}$ chosen uniformly at random, then*

$$\mathbb{E}\left[\left\|\nabla f(\hat{x}^T)\right\|^2\right] \leq \frac{2(f(x^0)-f^*)}{\gamma T} + \frac{G^0}{\theta(\alpha)T}. \tag{15}$$

### 2.4 Step 4: From weights in the algorithm to weights in the analysis

In the preceding section, we introduced a novel algorithm: EF21-W. While it bears some resemblance to the vanilla EF21 algorithm (Richtárik et al., 2021) (we recover it for uniform weights), the reliance on particular non-uniform weights enables it to achieve a faster convergence rate. However, this is not the end of the story as another insight reveals yet another surprise.

Let us consider the scenario when the compressors in Algorithm 2 are *positively homogeneous*[2]. Introducing the new variable $h_i^t = nw_i g_i^t$, we can reformulate the gradient update in Algorithm 2 to

$$h_i^{t+1} = nw_i g_i^{t+1} \stackrel{(14)}{=} nw_i \left[g_i^t + \mathcal{C}_i^t\left(\frac{\nabla f_i(x^{t+1})}{nw_i} - g_i^t\right)\right] = h_i^t + \mathcal{C}_i^t(\nabla f_i(x^t) - h_i^t),$$

indicating that $h_i^t$ adheres to the update rule of the vanilla EF21 method! Furthermore, the iterates $x^t$ also follow the same rule as EF21:

$$x^{t+1} \stackrel{(13)}{=} x^t - \gamma \sum_{i=1}^n w_i g^t = x^t - \gamma \sum_{i=1}^n w_i \frac{1}{nw_i} h_i^t = x^t - \gamma \frac{1}{n} \sum_{i=1}^n h_i^t.$$

So, what does this mean? One interpretation suggests that for positively homogeneous contractive compressors, the vanilla EF21 algorithm is equivalent to EF21-W, and hence inherits its faster convergence rate that depends on $L_{\text{AM}}$ rather than on $L_{\text{QM}}$. However, it turns out that we can establish the same result without having to resort to positive homogeneity altogether. For example, the "natural compression" quantizer, which rounds to one of the two nearest powers of two, is not positively homogeneous (Horváth et al., 2019a).

**Theorem 4** (**New theory for EF21**). *Consider Algorithm 1 (EF21) applied to the distributed optimization problem* (1). *Let Assumptions 1–3 hold, assume that $\mathcal{C}_i^t \in \mathbb{C}(\alpha)$ for all $i \in [n]$ and $t \geq 0$, set*

$$G^t := \frac{1}{n} \sum_{i=1}^n \frac{1}{nw_i} \left\|g_i^t - \nabla f_i(x^t)\right\|^2,$$

---

[2]A compressor $\mathcal{C} : \mathbb{R}^d \to \mathbb{R}^d$ is positively homogeneous if $\mathcal{C}(tx) = t\mathcal{C}(x)$ for all $t > 0$ and $x \in \mathbb{R}^d$.

*where $w_i = \frac{L_i}{\sum_j L_j}$ for all $i \in [n]$, and let the stepsize satisfy $0 < \gamma \leq \frac{1}{L+L_{\mathrm{AM}}\xi(\alpha)}$. If for $T > 1$ we define $\hat{x}^T$ as an element of the set $\{x^0, x^1, \ldots, x^{T-1}\}$ chosen uniformly at random, then*

$$\mathbb{E}\left[\|\nabla f(\hat{x}^T)\|^2\right] \leq \frac{2(f(x^0)-f^*)}{\gamma T} + \frac{G^0}{\theta(\alpha)T}. \tag{16}$$

This last result effectively pushes the weights from the algorithm in EF21-W to the proof, which enabled us to show that the original EF21 method also enjoys the same improvement: from $L_{\mathrm{QM}}$ to $L_{\mathrm{AM}}$.

## 3 EXPERIMENTS

### 3.1 NON-CONVEX LOGISTIC REGRESSION ON BENCHMARK DATASETS

In our first experiment, we employed a logistic regression model with a non-convex regularizer, i.e.,

$$f_i(x) \coloneqq \frac{1}{n_i} \sum_{j=1}^{n_i} \log\left(1 + \exp(-y_{ij} \cdot a_{ij}^\top x)\right) + \lambda \sum_{j=1}^{d} \frac{x_j^2}{x_j^2+1},$$

where $(a_{ij}, y_{ij}) \in \mathbb{R}^d \times \{-1, 1\}$ represents the $j$-th data point out from a set of $n_i$ data points stored at client $i$, and $\lambda > 0$ denotes a regularization coefficient. We utilized six datasets from LIBSVM (Chang & Lin, 2011). The dataset shuffling strategy, detailed in Appendix I.5, was employed to emulate heterogeneous data distribution. Each client was assigned the same number of data points. Figure 1 provides a comparison between EF21 employing the original stepsize (Richtárik et al., 2021) and EF21-W with the better stepsize. The initial gradient estimators were chosen as $g_i^0 = \nabla f_i(x^0)$ for all $i \in [n]$. As evidenced empirically, the EF21-W algorithm emerges as a practical choice when utilized in situations characterized by high variance in smoothness constants. As evident from the plots, the algorithm employing the new step size exhibits superior performance compared to its predecessor.

Next, we conducted a comparative analysis of the performance of EF21-W-PP and EF21-W-SGD, as elucidated in the appendix, compared to their non-weighted counterparts. In the EF21-PP/EF21-W-PP algorithms, each client participated independently in each round with probability $p_i = 0.5$. Moreover, in the case of EF21-SGD/EF21-W-SGD algorithms, a single data point was stochastically sampled from a uniform distribution at each client during each iteration of the algorithm. As observed in Figure 2, the algorithms employing the new learning rates demonstrate faster convergence. Notably, Figure 2 (c) depicts more pronounced oscillations with updated step sizes, as the new analysis permits larger step sizes, which can induce oscillations in stochastic methods.

### 3.2 NON-CONVEX LINEAR MODEL ON SYNTHETIC DATASETS

In our second set of experiments, we trained a linear regression model with a non-convex regularizer. The function $f_i$ for the linear regression problem is defined as follows: $f_i(x) \coloneqq \frac{1}{n_i}\|\mathbf{A}_i x - b_i\|^2 + \lambda \sum_{j=1}^{d} \frac{x_j^2}{x_j^2+1}$. Here, $\mathbf{A}_i \in \mathbb{R}^{n_i \times d}$ and $b_i \in \mathbb{R}^{n_i}$ represent the feature matrix and labels stored on client $i$ encompassing $n_i$ data points. The data employed in four experiments, as illustrated in Figure 3, was generated in such a manner that the smoothness constant $L$ remained fixed, while $L_i$ varied so that the difference between two crucial to analysis terms $L_{\mathrm{QM}}$ and $L_{\mathrm{AM}}$ changed from a relatively large value to negligible. As evident from Figure 3, the performance of EF21-W consistently matches or surpasses that of the original EF21, particularly in scenarios characterized by significant variations in the smoothness constants. For additional details and supplementary experiments, we refer the reader to Sections I and J.

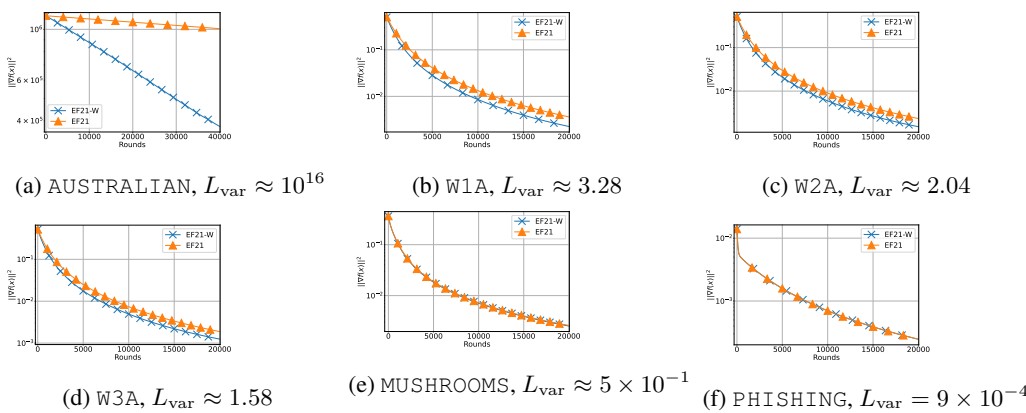

Figure 1: Comparison of EF21 versus our new EF21-W with the Top1 compressor on the non-convex logistic regression problem. The number of clients $n$ is $1,000$. The step size for EF21 is set according to (Richtárik et al., 2021), and the step size for EF21-W is set according to Theorem 3. The coefficient $\lambda$ for (b)–(f) is set to $0.001$, and for (a) is set to $1,000$ for numerical stability. We let $L_{\text{var}} := L_{\text{QM}}^2 - L_{\text{AM}}^2 = \frac{1}{n}\sum_{i=1}^n L_i^2 - \left(\frac{1}{n}\sum_{i=1}^n L_i\right)^2$.

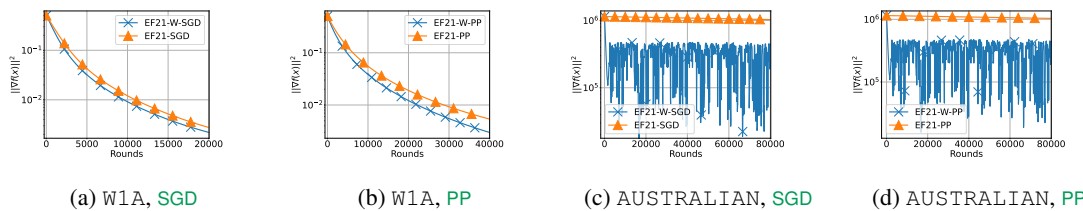

Figure 2: Comparison of EF21-W with partial partial participation (EF21-W-PP) or stochastic gradients (EF21-W-SGD) versus EF21 with partial partial participation (EF21-PP) or stochastic gradients (EF21-SGD) (Fatkhullin et al., 2021)). The Top1 compressor was employed in all experiments. The number of clients $n = 1,000$. All stepsizes are theoretical. The coefficient $\lambda$ was set to $0.001$ for (a), (b) and to $1,000$ for (c), (d).

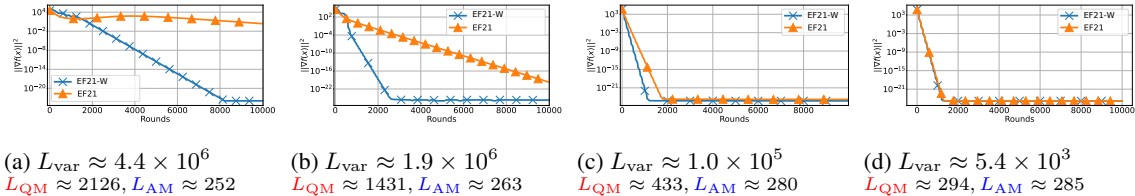

Figure 3: Comparison of EF21 vs. EF21-W with the Top1 compressor on the non-convex linear problem. The number of clients $n$ is $2,000$. The coefficient $\lambda$ has been set to $100$. The step size for EF21 is set according to (Richtárik et al., 2021), and the step size for EF21-W is set according to Theorem 3. In all cases, the smoothness constant $L$ equals $50$.

## ACKNOWLEDGEMENTS

This work of all authors was supported by the KAUST Baseline Research Scheme (KAUST BRF). The work Peter Richtárik and Konstantin Burlachenko was also supported by the SDAIA-KAUST Center of Excellence in Data Science and Artificial Intelligence (SDAIA-KAUST AI). We wish to thank Babis Kostopoulos—a VSRP intern at KAUST who spent some time working on this project in Summer 2023—for helping with some parts of the project. We offered Babis co-authorship, but he declined.

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

# A BASIC RESULTS AND LEMMAS

In this section, we offer a few results that serve as essential prerequisites for establishing the main findings in the paper.

## A.1 OPTIMAL CLIENT CLONING FREQUENCIES

**Lemma 1** (Optimal weights). *Let $a_i > 0$ for $i \in [n]$. Then*

$$\min_{\substack{w_1 > 0, \ldots, w_n > 0 \\ \sum_{i=1}^n w_i = 1}} \sum_{i=1}^n \frac{a_i^2}{w_i} = \left( \sum_{i=1}^n a_i \right)^2, \tag{17}$$

*which is achieved when $w_i^* = \frac{a_i}{\sum_j a_j}$. This means that*

$$\min_{\substack{w_1 > 0, \ldots, w_n > 0 \\ \sum_{i=1}^n w_i = 1}} \frac{1}{n} \sqrt{\sum_{i=1}^n \frac{a_i^2}{w_i}} = \frac{1}{n} \sum_{i=1}^n a_i. \tag{18}$$

We now show that the cloning frequencies given by $N_i^\star = \left\lceil \frac{L_i}{L_{\mathrm{AM}}} \right\rceil$ form a $\sqrt{2}$-approximation for the optimization problem of finding the optimal integer client frequencies.

**Lemma 2** ($\sqrt{2}$-approximation). *If we let $N_i^\star = \left\lceil \frac{L_i}{L_{\mathrm{AM}}} \right\rceil$ for all $i \in [n]$, then*

$$L_{\mathrm{AM}} \le \min_{N_1 \in \mathbb{N}, \ldots, N_n \in \mathbb{N}} M(N_1, \ldots, N_n) \le M(N_1^\star, \ldots, N_n^\star) \le \sqrt{2} L_{\mathrm{AM}}.$$

*Proof.* Recall that

$$M(N_1, \ldots, N_n) := \frac{1}{n} \sqrt{\sum_{i=1}^n \frac{L_i^2}{N_i/N}}.$$

The first inequality in the claim follows by relaxing the integrality constraints, which gives us the bound

$$\min_{\substack{w_1 > 0, \ldots, w_n > 0 \\ \sum_{i=1}^n w_i = 1}} \frac{1}{n} \sqrt{\sum_{i=1}^n \frac{L_i^2}{w_i}} \le \min_{N_1 \in \mathbb{N}, \ldots, N_n \in \mathbb{N}} \frac{1}{n} \sqrt{\sum_{i=1}^n \frac{L_i^2}{N_i/N}},$$

and subsequently applying Lemma 17.

Next, we argue that the quantity $N^\star := \sum_i N_i^\star$ is at most $2n$. Indeed,

$$N^\star = \sum_{i=1}^n N_i^\star = \sum_{i=1}^n \left\lceil \frac{L_i}{L_{\mathrm{AM}}} \right\rceil \le \sum_{i=1}^n \left( \frac{L_i}{L_{\mathrm{AM}}} + 1 \right) = 2n. \tag{19}$$

We will now use this to bound $M(N_1^\star, \ldots, N_n^\star)$ from above:

$$M(N_1^\star, \ldots, N_n^\star) = \frac{1}{n} \sqrt{\sum_{i=1}^n \frac{L_i^2}{N_i^\star/N^\star}} \overset{(19)}{=} \frac{\sqrt{2}}{\sqrt{n}} \sqrt{\sum_{i=1}^n \frac{L_i^2}{N_i^\star}} = \frac{\sqrt{2}}{\sqrt{n}} \sqrt{\sum_{i=1}^n \frac{\frac{L_i}{L_{\mathrm{AM}}}}{N_i^\star} L_i L_{\mathrm{AM}}}.$$

Since $\frac{\frac{L_i}{L_{\mathrm{AM}}}}{N_i^\star} \le 1$ for all $i \in [n]$, the proof is finished as follows:

$$M(N_1^\star, \ldots, N_n^\star) \le \frac{\sqrt{2}}{\sqrt{n}} \sqrt{\sum_{i=1}^n L_i L_{\mathrm{AM}}} = \frac{\sqrt{2}}{\sqrt{n}} \sqrt{L_{\mathrm{AM}}} \sqrt{\sum_{i=1}^n L_i} = \sqrt{2} L_{\mathrm{AM}}.$$

$\square$

## A.2 DESCENT LEMMA

**Lemma 3** (Li et al. (2021)). *Let Assumption 1 hold and $x^{t+1} = x^t - \gamma g^t$, where $g^t \in \mathbb{R}^d$ is any vector, and $\gamma > 0$ is any scalar. Then, we have*

$$f(x^{t+1}) \leq f(x^t) - \frac{\gamma}{2}\|\nabla f(x^t)\|^2 - \left(\frac{1}{2\gamma} - \frac{L}{2}\right)\|x^{t+1} - x^t\|^2 + \frac{\gamma}{2}\|g^t - \nabla f(x^t)\|^2. \tag{20}$$

## A.3 YOUNG'S INEQUALITY

**Lemma 4** (Young's inequality). *For any $a, b \in \mathbb{R}^d$ and any positive scalar $s > 0$ it holds that*
$$\|a + b\|^2 \leq (1 + s)\|a\|^2 + (1 + s^{-1})\|b\|^2. \tag{21}$$

## A.4 2-SUBOPTIMAL BUT SIMPLE STEPSIZE RULE

**Lemma 5** (Lemma 5, Richtárik et al. (2021)). *Let $a, b > 0$. If $0 < \gamma \leq \frac{1}{\sqrt{a}+b}$, then $a\gamma^2 + b\gamma \leq 1$. Moreover, the bound is tight up to the factor of 2 since*

$$\frac{1}{\sqrt{a} + b} \leq \min\left\{\frac{1}{\sqrt{a}}, \frac{1}{b}\right\} \leq \frac{2}{\sqrt{a} + b}.$$

## A.5 OPTIMAL COEFFICIENT IN YOUNG'S INEQUALITY

**Lemma 6** (Lemma 3, Richtárik et al. (2021)). *Let $0 < \alpha \leq 1$ and for $s > 0$, let $\theta(\alpha, s) := 1 - (1 - \alpha)(1 + s)$ and $\beta(\alpha, s) := (1 - \alpha)\left(1 + s^{-1}\right)$. Then, the solution of the optimization problem*

$$\min_s\left\{\frac{\beta(\alpha, s)}{\theta(\alpha, s)} : 0 < s < \frac{\alpha}{1 - \alpha}\right\} \tag{22}$$

*is given by $s^* = \frac{1}{\sqrt{1-\alpha}} - 1$. Furthermore, $\theta(\alpha, s^*) = 1 - \sqrt{1 - \alpha}$, and $\beta(\alpha, s^*) = \frac{1-\alpha}{1-\sqrt{1-\alpha}}$.*

## B   CLONING REFORMULATION FOR POLYAK-ŁOJASCHEWITZ FUNCTIONS

For completeness, we also provide a series of convergence results under Polyak-Łojasiewicz condition. We commence our exposition with the subsequent definition.

**Assumption 4** (Polyak-Łojasiewicz). *There exists a positive scalar $\mu > 0$ such that for all points $x \in \mathbb{R}^d$, the following inequality is satisfied:*

$$f(x) - f(x^*) \leq \frac{1}{2\mu} \|\nabla f(x)\|^2, \tag{23}$$

*where $x^* \coloneqq \operatorname{argmin} f(x)$.*

**Theorem 5.** *Let Assumptions 1, 2, and 4 hold. Assume that $\mathcal{C}_i^t \in \mathbb{C}(\alpha)$ for all $i \in [n]$ and $t \geq 0$. Consider Algorithm 1 (EF21) applied to the "cloning" reformulation 9 of the distributed optimization problem* (1), *where $N_i^* = \lceil \frac{L_i}{L_{\mathrm{AM}}} \rceil$ for all $i \in [n]$. Let the stepsize be set as*

$$0 \leq \gamma \leq \min \left\{ \left( L + \sqrt{2} L_{\mathrm{AM}} \sqrt{\frac{2\beta}{\theta}} \right)^{-1}, \frac{\theta}{2\mu} \right\},$$

*where $\theta = 1 - \sqrt{1-\alpha}$ and $\beta = \frac{1-\alpha}{1-\sqrt{1-\alpha}}$. Let*

$$\Psi^t \coloneqq f(x^t) - f(x^*) + \frac{\gamma}{\theta} G^t.$$

*Then, for any $T \geq 0$, we have*

$$\mathbb{E}\left[ \Psi^T \right] \leq (1 - \gamma\mu)^T \Psi^0.$$

*Proof.* This theorem is a corollary of Theorem 2 in (Richtárik et al., 2021) and Lemma 2.  □

## C  PROOF OF THEOREM 3 (THEORY FOR EF21-W)

In this section, we present a proof for Theorem 3. To start this proof, we establish a corresponding contraction lemma. We define the following quantities:

$$G_i^t := \left\| g_i^t - \frac{\nabla f_i(x^t)}{nw_i} \right\|^2; \qquad G^t := \sum_{i=1}^n w_i G_i^t, \qquad (24)$$

where the weights $w_i$ are defined as specified in Algorithm 2, that is,

$$w_i = \frac{L_i}{\sum_{j=1}^n L_j}. \qquad (25)$$

### C.1  A LEMMA

With these definitions in place, we are now prepared to proceed to the lemma.

**Lemma 7.** *Let* $\mathcal{C}_i^t \in \mathbb{C}(\alpha)$ *for all* $i \in [n]$ *and* $t \geq 0$. *Let* $W^t := \{g_1^t, g_2^t, \ldots, g_n^t, x^t, x^{t+1}\}$. *Then, for iterates of Algorithm 2 we have*

$$\mathbb{E}\left[ G_i^{t+1} \mid W^t \right] \leq (1 - \theta(\alpha, s)) G_i^t + \beta(\alpha, s) \frac{1}{n^2 w_i^2} \left\| \nabla f_i(x^{t+1}) - \nabla f_i(x^t) \right\|^2, \qquad (26)$$

*and*

$$\mathbb{E}\left[ G^{t+1} \right] \leq (1 - \theta(\alpha, s)) \mathbb{E}\left[ G^t \right] + \beta(\alpha, s) L_{\mathrm{AM}}^2 \mathbb{E}\left[ \left\| x^{t+1} - x^t \right\|^2 \right], \qquad (27)$$

*where* $s > 0$ *is an arbitrary positive scalar, and*

$$\theta(\alpha, s) := 1 - (1 - \alpha)(1 + s), \qquad \text{and} \qquad \beta(\alpha, s) := (1 - \alpha)\left(1 + s^{-1}\right). \qquad (28)$$

*Proof.* The proof is straightforward and bears resemblance to a similar proof found in a prior work (Richtárik et al., 2021).

$$
\begin{aligned}
\mathbb{E}\left[ G_i^{t+1} \mid W^t \right] &\overset{(24)}{=} \mathbb{E}\left[ \left\| g_i^{t+1} - \frac{\nabla f_i(x^{t+1})}{nw_i} \right\|^2 \mid W^t \right] \\
&= \mathbb{E}\left[ \left\| g_i^t + \mathcal{C}_i^t\left( \frac{\nabla f_i(x^{t+1})}{nw_i} - g_i^t \right) - \frac{\nabla f_i(x^{t+1})}{nw_i} \right\|^2 \mid W^t \right] \\
&\overset{(4)}{\leq} (1 - \alpha) \left\| \frac{\nabla f_i(x^{t+1})}{nw_i} - g_i^t \right\|^2 \\
&= (1 - \alpha) \left\| \frac{\nabla f_i(x^{t+1})}{nw_i} - \frac{\nabla f_i(x^t)}{nw_i} + \frac{\nabla f_i(x^t)}{nw_i} - g_i^t \right\|^2 \\
&\overset{(21)}{\leq} (1 - \alpha)(1 + s) \left\| \frac{\nabla f_i(x^t)}{nw_i} - g_i^t \right\|^2 \\
&\quad + (1 - \alpha)\left(1 + s^{-1}\right) \frac{1}{n^2 w_i^2} \left\| \nabla f_i(x^{t+1}) - \nabla f_i(x^t) \right\|^2,
\end{aligned}
$$

with the final inequality holding for any positive scalar $s > 0$. Consequently, we have successfully established the first part of the lemma.

By employing (24) and the preceding inequality, we can derive the subsequent bound for the conditional expectation of $G^{t+1}$:

$$
\mathbb{E}\left[G^{t+1} \mid W^t\right] \overset{(24)}{=} \mathbb{E}\left[\sum_{i=1}^n w_i G_i^{t+1} \mid W^t\right]
$$

$$
\overset{(24)}{=} \sum_{i=1}^n w_i \mathbb{E}\left[\left\|g_i^{t+1} - \frac{\nabla f_i(x^{t+1})}{n w_i}\right\|^2 \mid W^t\right]
$$

$$
\overset{(26)}{\leq} (1 - \theta(\alpha, s)) \sum_{i=1}^n w_i \left\|g_i^t - \frac{\nabla f_i(x^t)}{n w_i}\right\|^2
$$

$$
+ \beta(\alpha, s) \sum_{i=1}^n \frac{w_i}{w_i^2 n^2} \left\|\nabla f_i(x^{t+1}) - \nabla f_i(x^t)\right\|^2. \tag{29}
$$

Applying Assumption 2 and (25), we further proceed to:

$$
\mathbb{E}\left[G^{t+1} \mid W^t\right] \overset{(29)}{\leq} (1 - \theta(\alpha, s)) \sum_{i=1}^n w_i \left\|g_i^t - \frac{\nabla f_i(x^t)}{n w_i}\right\|^2 + \beta(\alpha, s) \sum_{i=1}^n \frac{w_i}{w_i^2 n^2} \left\|\nabla f_i(x^{t+1}) - \nabla f_i(x^t)\right\|^2
$$

$$
\overset{(24)}{=} (1 - \theta(\alpha, s)) G^t + \beta(\alpha, s) \sum_{i=1}^n \frac{1}{w_i n^2} \left\|\nabla f_i(x^{t+1}) - \nabla f_i(x^t)\right\|^2
$$

$$
\overset{(7)}{\leq} (1 - \theta(\alpha, s)) G^t + \beta(\alpha, s) \left(\sum_{i=1}^n \frac{L_i^2}{w_i n^2}\right) \left\|x^{t+1} - x^t\right\|^2
$$

$$
\overset{(25)}{=} (1 - \theta(\alpha, s)) G^t + \beta(\alpha, s) \left(\sum_{i=1}^n \frac{L_i^2}{\frac{L_i}{\sum_{j=1}^n L_j} n^2}\right) \left\|x^{t+1} - x^t\right\|^2
$$

$$
= (1 - \theta(\alpha, s)) G^t + \beta(\alpha, s) \left(\sum_{i=1}^n \frac{L_i \sum_{j=1}^n L_j}{n^2}\right) \left\|x^{t+1} - x^t\right\|^2
$$

$$
= (1 - \theta(\alpha, s)) G^t + \beta(\alpha, s) L_{\mathrm{AM}}^2 \left\|x^{t+1} - x^t\right\|^2. \tag{30}
$$

Using the tower property, we get

$$
\mathbb{E}\left[G^{t+1}\right] = \mathbb{E}\left[\mathbb{E}\left[G^{t+1} \mid W^t\right]\right] \overset{(30)}{\leq} (1 - \theta(\alpha, s)) \mathbb{E}\left[G^t\right] + \beta(\alpha, s) L_{\mathrm{AM}}^2 \mathbb{E}\left[\left\|x^{t+1} - x^t\right\|^2\right],
$$

and this finalizes the proof. □

## C.2 MAIN RESULT

We are now prepared to establish the proof for Theorem 3.

*Proof.* Note that, according to (13), the gradient estimate for Algorithm 2 gets the following form:

$$
g^t = \sum_{i=1}^n w_i g_i^t. \tag{31}
$$

Using Lemma 3 and Jensen's inequality applied to the function $x \mapsto \|x\|^2$ (since $\sum_{i=1}^n w_i = 1$), we obtain the following bound:

$$
\begin{aligned}
f(x^{t+1}) \overset{(20)}{\le} \;\; & f(x^t) - \frac{\gamma}{2} \left\| \nabla f(x^t) \right\|^2 - \left( \frac{1}{2\gamma} - \frac{L}{2} \right) \left\| x^{t+1} - x^t \right\|^2 + \frac{\gamma}{2} \left\| g^t - \sum_{i=1}^n \nabla f_i(x^t) \right\|^2 \\
\overset{(31)}{=} \;\; & f(x^t) - \frac{\gamma}{2} \left\| \nabla f(x^t) \right\|^2 - \left( \frac{1}{2\gamma} - \frac{L}{2} \right) \left\| x^{t+1} - x^t \right\|^2 + \frac{\gamma}{2} \left\| \sum_{i=1}^n w_i \left( g_i^t - \frac{\nabla f_i(x^t)}{nw_i} \right) \right\|^2 \\
\le \;\; & f(x^t) - \frac{\gamma}{2} \left\| \nabla f(x^t) \right\|^2 - \left( \frac{1}{2\gamma} - \frac{L}{2} \right) \left\| x^{t+1} - x^t \right\|^2 + \frac{\gamma}{2} \sum_{i=1}^n w_i \left\| g_i^t - \frac{\nabla f_i(x^t)}{nw_i} \right\|^2 \\
\overset{(24)}{=} \;\; & f(x^t) - \frac{\gamma}{2} \left\| \nabla f(x^t) \right\|^2 - \left( \frac{1}{2\gamma} - \frac{L}{2} \right) \left\| x^{t+1} - x^t \right\|^2 + \frac{\gamma}{2} G^t. \tag{32}
\end{aligned}
$$

Subtracting $f^*$ from both sides and taking expectation, we get

$$
\begin{aligned}
\mathbb{E} \left[ f(x^{t+1}) - f^* \right] \le \; & \mathbb{E} \left[ f(x^t) - f^* \right] - \frac{\gamma}{2} \mathbb{E} \left[ \left\| \nabla f(x^t) \right\|^2 \right] \\
& - \left( \frac{1}{2\gamma} - \frac{L}{2} \right) \mathbb{E} \left[ \left\| x^{t+1} - x^t \right\|^2 \right] + \frac{\gamma}{2} \mathbb{E} \left[ G^t \right]. \tag{33}
\end{aligned}
$$

Let $\delta^t := \mathbb{E} \left[ f(x^t) - f^* \right]$, $s^t := \mathbb{E} \left[ G^t \right]$ and $r^t := \mathbb{E} \left[ \left\| x^{t+1} - x^t \right\|^2 \right]$. Subsequently, by adding (27) with a $\frac{\gamma}{2\theta(\alpha,s)}$ multiplier, we obtain

$$
\begin{aligned}
\delta^{t+1} + \frac{\gamma}{2\theta(\alpha, s)} s^{t+1} \overset{(33)}{\le} \;\; & \delta^t - \frac{\gamma}{2} \left\| \nabla f(x^t) \right\|^2 - \left( \frac{1}{2\gamma} - \frac{L}{2} \right) r^t + \frac{\gamma}{2} s^t + \frac{\gamma}{2\theta} s^{t+1} \\
\overset{(27)}{\le} \;\; & \delta^t - \frac{\gamma}{2} \left\| \nabla f(x^t) \right\|^2 - \left( \frac{1}{2\gamma} - \frac{L}{2} \right) r^t + \frac{\gamma}{2} s^t \\
& + \frac{\gamma}{2\theta(\alpha, s)} \left( \beta(\alpha, s) L_{\mathrm{AM}}^2 r^t + (1 - \theta(\alpha, s)) s^t \right) \\
= \;\; & \delta^t + \frac{\gamma}{2\theta(\alpha, s)} s^t - \frac{\gamma}{2} \left\| \nabla f(x^t) \right\|^2 - \left( \frac{1}{2\gamma} - \frac{L}{2} - \frac{\gamma}{2\theta(\alpha, s)} \beta(\alpha, s) L_{\mathrm{AM}}^2 \right) r^t \\
\le \;\; & \delta^t + \frac{\gamma}{2\theta(\alpha, s)} s^t - \frac{\gamma}{2} \left\| \nabla f(x^t) \right\|^2.
\end{aligned}
$$

The last inequality is a result of the bound $\gamma^2 \frac{\beta(\alpha,s) L_{\mathrm{AM}}^2}{\theta(\alpha,s)} + L\gamma \le 1$, which is satisfied for the stepsize

$$
\gamma \le \frac{1}{L + L_{\mathrm{AM}} \xi(\alpha, s)},
$$

where $\xi(\alpha, s) := \sqrt{\frac{\beta(\alpha,s)}{\theta(\alpha,s)}}$. Maximizing the stepsize bound over the choice of $s$ using Lemma 6, we obtain the final stepsize. By summing up inequalities for $t = 0, \ldots, T - 1$, we get

$$
0 \le \delta^T + \frac{\gamma}{2\theta} s^T \le \delta^0 + \frac{\gamma}{2\theta} s^0 - \frac{\gamma}{2} \sum_{t=0}^{T-1} \mathbb{E} \left[ \left\| \nabla f(x^t) \right\|^2 \right].
$$

Multiplying both sides by $\frac{2}{\gamma T}$, after rearranging we get

$$
\sum_{t=0}^{T-1} \frac{1}{T} \mathbb{E} \left[ \left\| \nabla f(x^t) \right\|^2 \right] \le \frac{2\delta^0}{\gamma T} + \frac{s^0}{\theta T}.
$$

It remains to notice that the left hand side can be interpreted as $\mathbb{E}\left[\left\|\nabla f(\hat{x}^T)\right\|^2\right]$, where $\hat{x}^T$ is chosen from $\{x^0, x^1, \ldots, x^{T-1}\}$ uniformly at random. $\qquad\square$

### C.3 MAIN RESULT FOR POLYAK-ŁOJASIEWICZ FUNCTIONS

The main result is presented next.

**Theorem 6.** *Let Assumptions 1, 2, and 4 hold. Assume that $\mathcal{C}_i^t \in \mathbb{C}(\alpha)$ for all $i \in [n]$ and $t \geq 0$. Let the stepsize in Algorithm 2 be set as*

$$0 < \gamma \leq \min\left\{\frac{1}{L + \sqrt{2}L_{\mathrm{AM}}\xi(\alpha)}, \frac{\theta(\alpha)}{2\mu}\right\}.$$

*Let*

$$\Psi^t := f(x^t) - f(x^*) + \frac{\gamma}{\theta}G^t.$$

*Then, for any $T > 0$ the following inequality holds:*

$$\mathbb{E}\left[\Psi^T\right] \leq (1 - \gamma\mu)^T \Psi^0. \tag{34}$$

*Proof.* We proceed as in the previous proof, starting from the descent lemma with the same vector but using the PL inequality and subtracting $f(x^\star)$ from both sides:

$$\mathbb{E}\left[f(x^{t+1}) - f(x^\star)\right] \overset{(20)}{\leq} \mathbb{E}\left[f(x^t) - f(x^\star)\right] - \frac{\gamma}{2}\left\|\nabla f(x^t)\right\|^2 - \left(\frac{1}{2\gamma} - \frac{L}{2}\right)\left\|x^{t+1} - x^t\right\|^2 + \frac{\gamma}{2}G^t$$

$$\overset{(23)}{\leq} (1 - \gamma\mu)\mathbb{E}\left[f(x^t) - f(x^\star)\right] - \left(\frac{1}{2\gamma} - \frac{L}{2}\right)\left\|x^{t+1} - x^t\right\|^2 + \frac{\gamma}{2}G^t. \tag{35}$$

Let $\delta^t := \mathbb{E}\left[f(x^t) - f(x^\star)\right]$, $s^t := \mathbb{E}\left[G^t\right]$ and $r^t := \mathbb{E}\left[\left\|x^{t+1} - x^t\right\|^2\right]$. Thus,

$$\delta^{t+1} + \frac{\gamma}{\theta}s^{t+1} \overset{(44)}{\leq} (1 - \gamma\mu)\delta^t - \left(\frac{1}{2\gamma} - \frac{L}{2}\right)r^t + \frac{\gamma}{2}s^t + \frac{\gamma}{\theta}s^{t+1}$$

$$\overset{(27)}{\leq} (1 - \gamma\mu)\delta^t - \left(\frac{1}{2\gamma} - \frac{L}{2}\right)r^t + \frac{\gamma}{2}s^t + \frac{\gamma}{\theta}\left((1 - \theta)s^t + \beta\left(\frac{1}{n}\sum_{i=1}^n L_i\right)^2 r^t\right)$$

$$= (1 - \gamma\mu)\delta^t + \frac{\gamma}{\theta}\left(1 - \frac{\theta}{2}\right)s^t - \left(\frac{1}{2\gamma} - \frac{L}{2} - \frac{\beta L_{\mathrm{AM}}^2\gamma}{\theta}\right)r^t,$$

where $\theta$ and $\beta$ are set as in Lemma 6. Note that our extra assumption on the stepsize implies that $1 - \frac{\theta}{2} \leq 1 - \gamma\mu$ and

$$\frac{1}{2\gamma} - \frac{L}{2} - \frac{\beta L_{\mathrm{AM}}^2\gamma}{\theta} \geq 0.$$

The last inequality follows from the bound $\gamma^2\frac{2\beta L_{\mathrm{AM}}^2}{\theta} + \gamma L \leq 1$. Thus,

$$\delta^{t+1} + \frac{\gamma}{\theta}s^{t+1} \leq (1 - \gamma\mu)\left(\delta^t + \frac{\gamma}{\theta}s^t\right).$$

It remains to unroll the recurrence. $\qquad\square$

## D  PROOF OF THEOREM 4 (IMPROVED THEORY FOR EF21)

We commence by redefining gradient distortion as follows:

$$G^t := \frac{1}{n^2} \sum_{i=1}^{n} \frac{1}{w_i} \|\nabla f_i(x^t) - g_i^t\|^2. \tag{36}$$

We recall that the gradient update step for standard EF21 (Algorithm 1) takes the following form:

$$g_i^{t+1} = g_i^t + \mathcal{C}_i^t(\nabla f_i(x^{t+1}) - g_i^t), \tag{37}$$

$$g^{t+1} = \frac{1}{n} \sum_{i=1}^{n} g_i^{t+1}. \tag{38}$$

### D.1  TWO LEMMAS

Once more, we start our proof with the contraction lemma.

**Lemma 8.** *Let $\mathcal{C}_i^t \in \mathbb{C}(\alpha)$ for all $i \in [n]$ and $t \geq 0$. Define $W^t := \{g_1^t, g_2^t, \ldots, g_n^t, x^t, x^{t+1}\}$. Let Assumption 2 hold. Then*

$$\mathbb{E}\left[G^{t+1} \mid W^t\right] \leq (1 - \theta(\alpha, s))G^t + \beta(\alpha, s)L_{\mathrm{AM}}^2 \|x^{t+1} - x^t\|^2, \tag{39}$$

*where $\theta(\alpha, s) := 1 - (1 - \alpha)(1 + s)$ and $\beta(\alpha, s) := (1 - \alpha)(1 + s^{-1})$ for any $s > 0$.*

*Proof.* The proof of this lemma starts as the similar lemma in the standard analysis of EF21:

$$
\begin{aligned}
\mathbb{E}\left[G^{t+1} \mid W^t\right] &\overset{(36)}{=} \frac{1}{n^2} \sum_{i=1}^{n} \frac{1}{w_i} \mathbb{E}\left[\|\nabla f_i(x^{t+1}) - g_i^{t+1}\|^2 \mid W^t\right] \\
&\overset{(37)}{=} \frac{1}{n^2} \sum_{i=1}^{n} \frac{1}{w_i} \mathbb{E}\left[\|g_i^t + \mathcal{C}_i^t(\nabla f_i(x^{t+1}) - g_i^t) - \nabla f_i(x^{t+1})\|^2 \mid W^t\right] \\
&\overset{(4)}{\leq} \frac{1}{n^2} \sum_{i=1}^{n} \frac{1 - \alpha}{w_i} \|\nabla f_i(x^{t+1}) - g_i^t)\|^2 \\
&= \frac{1}{n^2} \sum_{i=1}^{n} \frac{1 - \alpha}{w_i} \|\nabla f_i(x^{t+1}) - \nabla f_i(x^t) + \nabla f_i(x^t) - g_i^t)\|^2 \\
&\overset{(21)}{\leq} \frac{1}{n^2} \sum_{i=1}^{n} \frac{1 - \alpha}{w_i} \left((1 + s^{-1})\|\nabla f_i(x^{t+1}) - \nabla f_i(x^t))\|^2 + (1 + s)\|g_i^t - \nabla f_i(x^t)\|^2\right)
\end{aligned}
$$

$$\tag{40}$$

for all $s > 0$. We proceed the proof as follows:

$$\mathbb{E}\left[G^{t+1} \mid W^t\right] \overset{(40)}{\leq} \frac{1}{n^2} \sum_{i=1}^{n} \frac{1-\alpha}{w_i} \left((1+s^{-1})\|\nabla f_i(x^{t+1}) - \nabla f_i(x^t)\|^2 + (1+s)\|g_i^t - \nabla f_i(x^t)\|^2\right)$$

$$= (1-\theta(\alpha,s))\frac{1}{n^2}\sum_{i=1}^{n}\frac{1}{w_i}\|g_i^t - \nabla f_i(x^t)\|^2 + \frac{\beta(\alpha,s)}{n^2}\sum_{i=1}^{n}\frac{1}{w_i}\|\nabla f_i(x^{t+1}) - \nabla f_i(x^t)\|^2$$

$$\overset{(36)}{=} (1-\theta(\alpha,s))G^t + \frac{\beta(\alpha,s)}{n^2}\sum_{i=1}^{n}\frac{1}{w_i}\|\nabla f_i(x^{t+1}) - \nabla f_i(x^t)\|^2$$

$$\overset{(7)}{\leq} (1-\theta(\alpha,s))G^t + \frac{\beta(\alpha,s)}{n^2}\sum_{i=1}^{n}\frac{L_i^2}{w_i}\|x^{t+1} - x^t\|^2. \tag{41}$$

Note that this is the exact place where the current analysis differs from the standard one. It fully coincides with it when $w_i = \frac{1}{n}$, i.e., when we assign the same weight for each individual gradient distortion $\|g_i^t - \nabla f_i(x^t)\|^2$. However, applying weights according to "importance" of each function, we proceed as follows:

$$\mathbb{E}\left[G^{t+1} \mid W^t\right] \overset{(41)}{\leq} (1-\theta(\alpha,s))G^t + \frac{\beta(\alpha,s)}{n^2}\sum_{i=1}^{n}\frac{L_i^2}{w_i}\|x^{t+1} - x^t\|^2$$

$$\overset{(25)}{=} (1-\theta(\alpha,s))G^t + \frac{\beta(\alpha,s)}{n^2}\sum_{i=1}^{n}\frac{L_i^2}{L_i}\left(\sum_{i=1}^{n}L_i\right)\|x^{t+1} - x^t\|^2$$

$$= (1-\theta(\alpha,s))G^t + \frac{\beta(\alpha,s)}{n^2}\sum_{j}L_j\left(\sum_{i=1}^{n}L_i\right)\|x^{t+1} - x^t\|^2$$

$$= (1-\theta(\alpha,s))G^t + \beta(\alpha,s)L_{\mathrm{AM}}^2\|x^{t+1} - x^t\|^2,$$

what finishes the proof. $\qquad\square$

To prove the main convergence theorem, we also need the following lemma.

**Lemma 9.** *For the variable $g^t$ from Algorithm 1, the following inequality holds:*

$$\|g^t - \nabla f(x^t)\|^2 \leq G^t. \tag{42}$$

*Proof.* The proof is straightforward:

$$\|g^t - \nabla f(x^t)\|^2 \overset{(38)}{=} \left\|\sum_{i=1}^{n}\frac{1}{n}\left(g_i^t - \nabla f_i(x^t)\right)\right\|^2$$

$$= \left\|\sum_{i=1}^{n}w_i\frac{1}{w_i n}\left(g_i^t - \nabla f_i(x^t)\right)\right\|^2$$

$$\leq \sum_{i=1}^{n}w_i\left\|\frac{1}{w_i n}\left(g_i^t - \nabla f_i(x^t)\right)\right\|^2$$

$$= \sum_{i=1}^{n}\frac{1}{w_i n^2}\|g^t - \nabla f_i(x^t)\|^2 \overset{(36)}{=} G^t,$$

where the only inequality in this series of equations is derived using Jensen's inequality. $\qquad\square$

## D.2 MAIN RESULT

We are now equipped with all the necessary tools to establish the convergence theorem.

*Proof.* Let us define the Lyapunov function

$$\Phi^t := f(x^t) - f^* + \frac{\gamma}{2\theta(\alpha, s)} G^t.$$

Let us also define $W^t := \{g_1^t, g_2^t, \ldots, g_n^t, x^t, x^{t+1}\}$. We start as follows:

$\mathbb{E}\left[\Phi^{t+1} \mid W^t\right]$

$= \mathbb{E}\left[f(x^{t+1}) - f^* \mid W^t\right] + \frac{\gamma}{2\theta(\alpha, s)}\mathbb{E}\left[G^{t+1} \mid W^t\right]$

$\overset{(20)}{\leq} f(x^t) - f^* - \frac{\gamma}{2}\|\nabla f(x^t)\|^2 - \left(\frac{1}{2\gamma} - \frac{L}{2}\right)\|x^{t+1} - x^t\|^2 + \frac{\gamma}{2}\|g^t - \nabla f(x^t)\|^2$

$\qquad + \frac{\gamma}{2\theta(\alpha, s)}\mathbb{E}\left[G^{t+1} \mid W^t\right]$

$\overset{(42)}{\leq} f(x^t) - f^* - \frac{\gamma}{2}\|\nabla f(x^t)\|^2 - \left(\frac{1}{2\gamma} - \frac{L}{2}\right)\|x^{t+1} - x^t\|^2 + \frac{\gamma}{2}G^t$

$\qquad + \frac{\gamma}{2\theta(\alpha, s)}\mathbb{E}\left[G^{t+1} \mid W^t\right]$

$\overset{(39)}{\leq} f(x^t) - f^* - \frac{\gamma}{2}\|\nabla f(x^t)\|^2 - \left(\frac{1}{2\gamma} - \frac{L}{2}\right)\|x^{t+1} - x^t\|^2 + \frac{\gamma}{2}G^t$

$\qquad + \frac{\gamma}{2\theta(\alpha, s)}\left((1 - \theta(\alpha, s))G^t + \beta(\alpha, s)L_{\text{AM}}^2\|x^{t+1} - x^t\|^2\right)$

$= f(x^t) - f^* + \frac{\gamma}{2\theta(\alpha, s)}G^t - \frac{\gamma}{2}\|\nabla f(x^t)\|^2 - \underbrace{\left(\frac{1}{2\gamma} - \frac{L}{2} - \frac{\gamma\beta(\alpha, s)}{2\theta(\alpha, s)}L_{\text{AM}}^2\right)}_{\geq 0}\|x^{t+1} - x^t\|^2$

$\leq f(x^t) - f^* + \frac{\gamma}{2\theta(\alpha, s)}G^t - \frac{\gamma}{2}\|\nabla f(x^t)\|^2$

$= \Phi^t - \frac{\gamma}{2}\|\nabla f(x^t)\|^2.$

The inequality in the last but one line is valid if

$$\gamma \leq \frac{1}{L + L_{\text{AM}}\sqrt{\frac{\beta(\alpha, s)}{\theta(\alpha, s)}}},$$

according to Lemma 5. By optimizing the stepsize bound through the selection of $s$ in accordance with Lemma 6, we derive the final stepsize and establish the optimal value for $\theta$ in defining the Lyapunov function. Applying the tower property and unrolling the recurrence, we finish the proof. $\qquad\square$

## D.3 MAIN RESULT FOR POLYAK-ŁOJASIEWICZ FUNCTIONS

For completeness, we also provide a convergence result under Polyak-Łojasiewicz condition (Assumption 4). The main result is presented next.

**Theorem 7.** *Let Assumptions 1, 2, and 4 hold. Assume that $\mathcal{C}_i^t \in \mathbb{C}(\alpha)$ for all $i \in [n]$ and $t \geq 0$. Let the stepsize in Algorithm 2 be set as*

$$0 < \gamma \leq \min\left\{\frac{1}{L + \sqrt{2}L_{\mathrm{AM}}\xi(\alpha)}, \frac{\theta(\alpha, s)}{2\mu}\right\}.$$

*Let*

$$\Psi^t := f(x^t) - f(x^*) + \frac{\gamma}{\theta(\alpha, s)}G^t.$$

*Then, for any $T > 0$ the following inequality holds:*

$$\mathbb{E}\left[\Psi^T\right] \leq (1 - \gamma\mu)^T\Psi^0. \tag{43}$$

*Proof.* We proceed as in the previous proof, starting from the descent lemma with the same vector but using the PL inequality and subtracting $f(x^\star)$ from both sides:

$$\mathbb{E}\left[f(x^{t+1}) - f(x^\star)\right] \overset{(20)}{\leq} \mathbb{E}\left[f(x^t) - f(x^\star)\right] - \frac{\gamma}{2}\left\|\nabla f(x^t)\right\|^2 - \left(\frac{1}{2\gamma} - \frac{L}{2}\right)\left\|x^{t+1} - x^t\right\|^2 + \frac{\gamma}{2}G^t$$

$$\overset{(23)}{\leq} (1 - \gamma\mu)\mathbb{E}\left[f(x^t) - f(x^\star)\right] - \left(\frac{1}{2\gamma} - \frac{L}{2}\right)\left\|x^{t+1} - x^t\right\|^2 + \frac{\gamma}{2}G^t. \tag{44}$$

Let $\delta^t := \mathbb{E}\left[f(x^t) - f(x^\star)\right]$, $s^t := \mathbb{E}\left[G^t\right]$ and $r^t := \mathbb{E}\left[\left\|x^{t+1} - x^t\right\|^2\right]$. Thus,

$$\delta^{t+1} + \frac{\gamma}{\theta(\alpha, s)}s^{t+1} \overset{(44)}{\leq} (1 - \gamma\mu)\delta^t - \left(\frac{1}{2\gamma} - \frac{L}{2}\right)r^t + \frac{\gamma}{2}s^t + \frac{\gamma}{\theta(\alpha, s)}s^{t+1}$$

$$\overset{(39)}{\leq} (1 - \gamma\mu)\delta^t - \left(\frac{1}{2\gamma} - \frac{L}{2}\right)r^t + \frac{\gamma}{2}s^t$$

$$+ \frac{\gamma}{\theta(\alpha, s)}\left((1 - \theta(\alpha, s))s^t + \beta\left(\frac{1}{n}\sum_{i=1}^n L_i\right)^2 r^t\right)$$

$$= (1 - \gamma\mu)\delta^t + \frac{\gamma}{\theta(\alpha, s)}\left(1 - \frac{\theta(\alpha, s)}{2}\right)s^t - \left(\frac{1}{2\gamma} - \frac{L}{2} - \frac{\beta L_{\mathrm{AM}}^2\gamma}{\theta(\alpha, s)}\right)r^t.$$

Note that our extra assumption on the stepsize implies that $1 - \frac{\theta(\alpha, s)}{2} \leq 1 - \gamma\mu$ and

$$\frac{1}{2\gamma} - \frac{L}{2} - \frac{\beta L_{\mathrm{AM}}^2\gamma}{\theta(\alpha, s)} \geq 0.$$

The last inequality follows from the bound $\gamma^2\frac{2\beta L_{\mathrm{AM}}^2}{\theta(\alpha, s)} + \gamma L \leq 1$. Thus,

$$\delta^{t+1} + \frac{\gamma}{\theta(\alpha, s)}s^{t+1} \leq (1 - \gamma\mu)\left(\delta^t + \frac{\gamma}{\theta(\alpha, s)}s^t\right).$$

It remains to unroll the recurrence which finishes the prove. $\qquad\square$

# E  EF21-W-SGD: WEIGHTED ERROR FEEDBACK 2021 WITH STOCHASTIC SUBSAMPLED GRADIENTS

The EF21-W algorithm assumes that all clients can compute the exact gradient in each round. In some scenarios, the exact gradients may be unavailable or too costly to compute, and only approximate gradient estimators can be obtained. In this section, we present the convergence result for EF21-W in the setting where the gradient computation on the clients, $\nabla f_i(x^{t+1})$, is replaced by a specific stochastic gradient estimator. For a variation of EF21-W-SGD which is working under a more general setting please see Appendix F.

## E.1  ALGORITHM

In this section, we extend EF21-W to handle stochastic gradients, and we call the resulting algorithm EF21-W-SGD (Algorithm 3). Our analysis of this extension follows a similar approach as the one used by Fatkhullin et al. (2021) for studying the stochastic gradient version of the vanilla EF21 algorithm, which they called EF21-SGD. Analysis of EF21-W-SGD has two important differences with vanilla EF21-SGD:

1. Vanilla EF21-SGD provides maximum theoretically possible $\gamma = \left( L + L_{\mathrm{QM}} \sqrt{\frac{\beta_1}{\theta}} \right)^{-1}$, where

   EF21-W-SGD has $\gamma = \left( L + L_{\mathrm{AM}} \sqrt{\frac{\beta_1}{\theta}} \right)^{-1}$

2. Vanilla EF21-SGD and EF21-W-SGD formally differs in a way how it reports iterate $x^T$ which minimizes $\mathbb{E}\left[ \left\| \nabla f(x^T) \right\|^2 \right]$ due to a slightly different definition of $\widetilde{A}$. The EF21-W-SGD (Algorithm 3) requires output iterate $\hat{x}^T$ randomly according to the probability mass function described by (49).

---

**Algorithm 3** EF21-W-SGD: Weighted Error Feedback 2021 with Stochastic Gradients

1: **Input:** initial model $x^0 \in \mathbb{R}^d$; initial gradient estimates $g_1^0, g_2^0, \ldots, g_n^0 \in \mathbb{R}^d$ stored at the server and the clients; stepsize $\gamma > 0$; number of iterations $T > 0$; weights $w_i = \frac{L_i}{\sum_j L_j}$ for $i \in [n]$
2: **Initialize:** $g^0 = \sum_{i=1}^n w_i g_i^0$ on the server
3: **for** $t = 0, 1, 2, \ldots, T-1$ **do**
4:   Server computes $x^{t+1} = x^t - \gamma g^t$ and broadcasts $x^{t+1}$ to all $n$ clients
5:   **for** $i = 1, \ldots, n$ **on the clients in parallel do**
6:     Compute a stochastic estimator $\hat{g}_i(x^{t+1}) = \frac{1}{\tau_i} \sum_{j=1}^{\tau_i} \nabla f_{\xi_{ij}^t}(x^{t+1})$ of the gradient $\nabla f_i(x^{t+1})$
7:     Compute $u_i^t = \mathcal{C}_i^t \left( \frac{1}{n w_i} \hat{g}_i(x^{t+1}) - g_i^t \right)$ and update $g_i^{t+1} = g_i^t + u_i^t$
8:     Send the compressed message $u_i^t$ to the server
9:   **end for**
10:   Server updates $g_i^{t+1} = g_i^t + u_i^t$ for all $i \in [n]$, and computes $g^{t+1} = \sum_{i=1}^n w_i g_i^{t+1}$
11: **end for**
12: **Output:** Point $\hat{x}^T$ chosen from the set $\{x^0, \ldots, x^{T-1}\}$ randomly according to the law (49)

---

**Assumption 5** (General assumption for stochastic gradient estimators). *We assume that for all $i \in [n]$ there exist parameters $A_i, C_i \geq 0$, $B_i \geq 1$ such that*

$$\mathbb{E}\left[ \|\nabla f_{\xi_{ij}^t}(x)\|^2 \right] \leq 2 A_i \left( f_i(x) - f_i^{\mathrm{inf}} \right) + B_i \|\nabla f_i(x)\|^2 + C_i, \tag{45}$$

*holds for all $x \in \mathbb{R}^d$, where*[3] $f_i^{\mathrm{inf}} = \inf_{x \in \mathbb{R}^d} f_i(x) > -\infty$.

We study EF21-W-SGD under the same assumption as was used for analyzing Vanilla EF21-SGD, which we denote as Assumption 5. To the best of our knowledge, this assumption, which was originally presented as Assumption 2 by Khaled & Richtárik (2022), is the most general assumption for a stochastic gradient estimator in a non-convex setting.

Next, to be aligned with original Vanilla EF21-SGD (Fatkhullin et al., 2021) we have considered a specific form of gradient estimator. This specific form of gradient estimator from Vanilla EF21-SGD presented in Section 4.1.2. of Fatkhullin et al. (2021) where the stochastic gradient $\hat{g}_i$ has been computed as follows:

$$\hat{g}_i(x^{t+1}) = \frac{1}{\tau_i} \sum_{j=1}^{\tau_i} \nabla f_{\xi_{ij}^t}(x^{t+1}),$$

Here $\tau_i$ is a minibatch size of sampled datapoint indexed by $\xi_{ij}^t$ of client $i$ in iteration $t$. And $\xi_{ij}^t$ are independent random variables. For a version of EF21-W-SGD which is working under a more general setting please see Appendix F.

### E.2 A LEMMA

The contraction lemma in this case gets the following form:

**Lemma 10.** *Let $\mathcal{C}_i^t \in \mathbb{C}(\alpha)$ for all $i \in [n]$ and $t \geq 0$. Define*

$$G_i^t := \left\| g_i^t - \frac{\nabla f_i(x^t)}{n w_i} \right\|^2, \qquad G^t := \sum_{i=1}^n w_i G_i^t.$$

*Let Assumptions 2 and 5 hold. Then, for any $s, \nu > 0$ we have*

$$\mathbb{E}\left[G^{t+1}\right] \leq (1 - \hat{\theta})\mathbb{E}\left[G^t\right] + \hat{\beta}_1 L_{\mathrm{AM}}^2 \mathbb{E}\left[\left\|x^{t+1} - x^t\right\|^2\right] + \widetilde{A}\hat{\beta}_2 \mathbb{E}\left[f(x^{t+1}) - f^{\mathrm{inf}}\right] + \widetilde{C}\hat{\beta}_2, \qquad (46)$$

*where*

$$
\begin{aligned}
w_i &:= \frac{L_i}{\sum_j L_j}, \\
\hat{\theta} &:= 1 - (1 - \alpha)(1 + s)(1 + \nu), \\
\hat{\beta}_1 &:= 2(1 - \alpha)(1 + s)\left(s + \nu^{-1}\right), \\
\hat{\beta}_2 &:= 2(1 - \alpha)(1 + s)(1 + \nu^{-1}) + (1 + s^{-1}), \\
\widetilde{A} &:= \max_{i=1,\ldots,n} \left(\frac{2(A_i + L_i(B_i - 1))}{\tau_i} \frac{1}{n w_i}\right), \\
\widetilde{C} &:= \max_{i=1,\ldots,n} \left(\frac{C_i}{\tau_i} \frac{1}{n w_i}\right).
\end{aligned}
$$

---

[3]When $A_i = 0$ one can ignore the first term in the right-hand side of (45), i.e., assumption $\inf_{x \in \mathbb{R}^d} f_i(x) > -\infty$ is not required in this case.

*Proof.* Define $W^t := \{g_1^t, \ldots, g_n^t, x^t, x^{t+1}\}$. The proof starts as follows:

$$\mathbb{E}\left[G_i^{t+1} \mid W^t\right] \stackrel{(24)}{=} \mathbb{E}\left[\left\|g_i^{t+1} - \frac{\nabla f_i(x^{t+1})}{nw_i}\right\|^2 \mid W^t\right]$$

$$\stackrel{\text{line 7}}{=} \mathbb{E}\left[\left\|g_i^t + \mathcal{C}_i^t\left(\frac{\hat{g}_i(x^{t+1})}{nw_i} - g_i^t\right) - \frac{\nabla f_i(x^{t+1})}{nw_i}\right\|^2 \mid W^t\right]$$

$$= \mathbb{E}\left[\left\|\mathcal{C}_i^t\left(\frac{\hat{g}_i(x^{t+1})}{nw_i} - g_i^t\right) - \left(\frac{\hat{g}_i(x^{t+1})}{nw_i} - g_i^t\right) + \frac{\hat{g}_i(x^{t+1})}{nw_i} - \frac{\nabla f_i(x^{t+1})}{nw_i}\right\|^2 \mid W^t\right]$$

$$\stackrel{(21)}{\leq} (1+s)\mathbb{E}\left[\left\|\mathcal{C}_i^t\left(\frac{\hat{g}_i(x^{t+1})}{nw_i} - g_i^t\right) - \left(\frac{\hat{g}_i(x^{t+1})}{nw_i} - g_i^t\right)\right\|^2 \mid W^t\right]$$

$$+ (1+s^{-1})\mathbb{E}\left[\left\|\frac{\hat{g}_i(x^{t+1})}{nw_i} - \frac{\nabla f_i(x^{t+1})}{nw_i}\right\|^2 \mid W^t\right]$$

$$\stackrel{(4)}{\leq} (1-\alpha)(1+s)\mathbb{E}\left[\left\|\frac{\hat{g}_i(x^{t+1})}{nw_i} - \frac{\nabla f_i(x^t)}{nw_i} + \frac{\nabla f_i(x^t)}{nw_i} - g_i^t\right\|^2 \mid W^t\right]$$

$$+ (1+s^{-1})\mathbb{E}\left[\left\|\frac{\hat{g}_i(x^{t+1})}{nw_i} - \frac{\nabla f_i(x^{t+1})}{nw_i}\right\|^2 \mid W^t\right]$$

$$\stackrel{(21)}{\leq} (1-\alpha)(1+s)(1+\nu)\mathbb{E}\left[\left\|g_i^t - \frac{\nabla f_i(x^t)}{nw_i}\right\|^2\right]$$

$$+ (1-\alpha)(1+s)(1+\nu^{-1})\mathbb{E}\left[\left\|\frac{\nabla f_i(x^t)}{nw_i} - \frac{\hat{g}_i(x^{t+1})}{nw_i}\right\|^2 \mid W^t\right]$$

$$+ (1+s^{-1})\mathbb{E}\left[\left\|\frac{\hat{g}_i(x^{t+1})}{nw_i} - \frac{\nabla f_i(x^{t+1})}{nw_i}\right\|^2 \mid W^t\right]$$

$$\stackrel{(21)}{\leq} (1-\alpha)(1+s)(1+\nu)\mathbb{E}\left[\left\|g_i^t - \frac{\nabla f_i(x^t)}{nw_i}\right\|^2 \mid W^t\right]$$

$$+ 2(1-\alpha)(1+s)(1+\nu^{-1})\mathbb{E}\left[\left\|\frac{\nabla f_i(x^{t+1})}{nw_i} - \frac{\hat{g}_i(x^{t+1})}{nw_i}\right\|^2 \mid W^t\right]$$

$$+ 2(1-\alpha)(1+s)(1+\nu^{-1})\left\|\frac{\nabla f_i(x^{t+1})}{nw_i} - \frac{\nabla f_i(x^t)}{nw_i}\right\|^2$$

$$+ (1+s^{-1})\mathbb{E}\left[\left\|\frac{\hat{g}_i(x^{t+1})}{nw_i} - \frac{\nabla f_i(x^{t+1})}{nw_i}\right\|^2 \mid W^t\right].$$

To further bound the last term, which contains multiple $(1+s^{-1})$ factors, we leverage the property that $\hat{g}_i(x^{t+1})$ is a random variable serving as an unbiased estimator of $\nabla f_i(x^{t+1})$, taking the form

$$\hat{g}_i(x^{t+1}) = \frac{1}{\tau_i}\sum_{j=1}^{\tau_i} \nabla f_{\xi_{ij}^t}(x^{t+1}),$$

where $\xi_{ij}^t$ are independent random variables. Next, we can continue as follows:

$$
\begin{aligned}
\mathbb{E}\left[G_i^{t+1} \mid W^t\right] &\leq (1-\hat{\theta})\mathbb{E}\left[G_i^t \mid W^t\right] + \hat{\beta}_1 \frac{1}{n^2 w_i^2} \left\|\nabla f_i(x^{t+1}) - \nabla f_i(x^t)\right\|^2 \\
&\quad + \frac{\hat{\beta}_2}{(nw_i)^2} \left( \mathbb{E}\left[\left\|\frac{1}{\tau_i}\sum_{j=1}^{\tau_i} \nabla f_{\xi_{ij}^t}(x^{t+1}) - \frac{1}{\tau_i}\sum_{j=1}^{\tau_i} \nabla f_i(x^{t+1})\right\|^2 \mid W^t\right]\right) \\
&= (1-\hat{\theta})\mathbb{E}\left[G_i^t \mid W^t\right] + \hat{\beta}_1 \frac{1}{n^2 w_i^2} \left\|\nabla f_i(x^{t+1}) - \nabla f_i(x^t)\right\|^2 \\
&\quad + \frac{\hat{\beta}_2}{(nw_i)^2 \tau^2} \left( \mathbb{E}\left[\left\|\sum_{j=1}^{\tau_i}\left(\nabla f_{\xi_{ij}^t}(x^{t+1}) - \nabla f_i(x^{t+1})\right)\right\|^2 \mid W^t\right]\right) \\
&= (1-\hat{\theta})\mathbb{E}\left[G_i^t \mid W^t\right] + \hat{\beta}_1 \frac{1}{n^2 w_i^2} \left\|\nabla f_i(x^{t+1}) - \nabla f_i(x^t)\right\|^2 \\
&\quad + \frac{\hat{\beta}_2}{(nw_i)^2 \tau_i^2} \sum_{j=1}^{\tau_i}\left(\mathbb{E}\left[\left\|\nabla f_{\xi_{ij}^t}(x^{t+1})\right\|^2 \mid W^t\right] - \left\|\mathbb{E}\left[\nabla f_{\xi_{ij}^t}(x^{t+1}) \mid W^t\right]\right\|^2\right) \\
&\leq (1-\hat{\theta})\mathbb{E}\left[G_i^t \mid W^t\right] + \hat{\beta}_1 \frac{1}{n^2 w_i^2} \left\|\nabla f_i(x^{t+1}) - \nabla f_i(x^t)\right\|^2 \\
&\quad + \frac{\hat{\beta}_2}{(nw_i)^2 \tau_i^2} \sum_{j=1}^{\tau_i}\left(2A_i\left(f_i(x^{t+1}) - f_i^{\inf}\right) + B_i\|\nabla f_i(x^{t+1})\|^2 + C_i - \left\|\nabla f_i(x^{t+1}\|^2\right)\right) \\
&= (1-\hat{\theta})\mathbb{E}\left[G_i^t \mid W^t\right] + \hat{\beta}_1 \frac{1}{n^2 w_i^2} \left\|\nabla f_i(x^{t+1}) - \left\|\nabla f_i(x^t)\right\|^2\right\|^2 \\
&\quad + \frac{2A_i\hat{\beta}_2}{(nw_i)^2 \tau_i}\left(f_i(x^{t+1}) - f_i^{\inf}\right) + \frac{2(B_i-1)\hat{\beta}_2}{(nw_i)^2 \tau_i}\left(\frac{1}{2}\|\nabla f_i(x^{t+1})\|^2\right) + \frac{C_i\hat{\beta}_2}{(nw_i)^2 \tau_i} \\
&\leq (1-\hat{\theta})\mathbb{E}\left[G_i^t \mid W^t\right] + \hat{\beta}_1 \frac{1}{n^2 w_i^2} \left\|\nabla f_i(x^{t+1}) - \nabla f_i(x^t)\right\|^2 \\
&\quad + \frac{2A_i\hat{\beta}_2}{(nw_i)^2 \tau_i}\left(f_i(x^{t+1}) - f_i^{\inf}\right) + \frac{2(B_i-1)\hat{\beta}_2}{(nw_i)^2 \tau_i} L_i\left(f_i(x^{t+1}) - f_i^{\inf}\right) + \frac{C_i\hat{\beta}_2}{(nw_i)^2 \tau_i} \\
&= (1-\hat{\theta})\mathbb{E}\left[G_i^t \mid W^t\right] + \hat{\beta}_1 \frac{1}{n^2 w_i^2} \left\|\nabla f_i(x^{t+1}) - \nabla f_i(x^t)\right\|^2 \\
&\quad + \frac{2(A_i + L_i(B_i-1))\hat{\beta}_2}{(nw_i)^2 \tau_i}\left(f_i(x^{t+1}) - f_i^{\inf}\right) + \frac{C_i\hat{\beta}_2}{(nw_i)^2 \tau_i}.
\end{aligned}
$$

Furthermore, as a result of leveraging Assumption 2, we can derive the subsequent bound:

$$
\begin{aligned}
\mathbb{E}\left[G_i^{t+1} \mid W^t\right] &\leq (1-\hat{\theta})G_i^t + \frac{\hat{\beta}_1 L_i^2}{n^2 w_i^2} \left\|x^{t+1} - x^t\right\|^2 \\
&\quad + \frac{2(A_i + L_i(B_i-1))\hat{\beta}_2}{(nw_i)^2 \tau_i}\left(f_i(x^{t+1}) - f_i^{\inf}\right) + \frac{C_i\hat{\beta}_2}{(nw_i)^2 \tau_i}.
\end{aligned}
$$

Applying the tower property and subsequently taking the expectation, we obtain:

$$\mathbb{E}\left[G_i^{t+1}\right] \le (1-\hat{\theta})\mathbb{E}\left[G_i^t\right] + \hat{\beta}_1 \frac{1}{n^2 w_i^2} L_i^2 \mathbb{E}\left[\left\|x^{t+1} - x^t\right\|^2\right]$$

$$+ \frac{2(A_i + L_i(B_i - 1))\hat{\beta}_2}{(nw_i)^2 \tau_i} \mathbb{E}\left[f_i(x^{t+1}) - f_i^{\inf}\right] + \frac{C_i \hat{\beta}_2}{(nw_i)^2 \tau_i}. \tag{47}$$

Regarding the expectation of $G^{t+1}$, we derive the subsequent bound:

$$\mathbb{E}\left[G^{t+1}\right] = \mathbb{E}\left[\sum_{i=1}^n w_i G_i^{t+1}\right]$$

$$= \sum_{i=1}^n w_i \mathbb{E}\left[G_i^{t+1}\right]$$

$$\overset{(47)}{\le} (1-\hat{\theta})\sum_{i=1}^n w_i \mathbb{E}\left[G_i^t\right] + \sum_{i=1}^n w_i \hat{\beta}_1 \frac{1}{n^2 w_i^2} L_i^2 \cdot \mathbb{E}\left[\left\|x^{t+1} - x^t\right\|^2\right]$$

$$+ \sum_{i=1}^n w_i \frac{2(A_i + L_i(B_i - 1))\hat{\beta}_2}{(nw_i)^2 \tau_i} \cdot \mathbb{E}\left[f_i(x^{t+1}) - f_i^{\inf}\right] + \sum_{i=1}^n w_i \frac{C_i \hat{\beta}_2}{(nw_i)^2 \tau_i}$$

$$= (1-\hat{\theta})\mathbb{E}\left[G^t\right] + \sum_{i=1}^n \hat{\beta}_1 \frac{1}{n^2 w_i} L_i^2 \cdot \mathbb{E}\left[\left\|x^{t+1} - x^t\right\|^2\right]$$

$$+ \sum_{i=1}^n \frac{2(A_i + L_i(B_i - 1))\hat{\beta}_2}{n^2 w_i \tau_i} \cdot \mathbb{E}\left[f_i(x^{t+1}) - f_i^{\inf}\right] + \sum_{i=1}^n \frac{C_i \hat{\beta}_2}{n^2 w_i \tau_i}.$$

Employing quantities $\tilde{A}$ and $\tilde{C}$, the final bound can be reformulated as follows:

$$\mathbb{E}\left[G^{t+1}\right] \le (1-\hat{\theta})\mathbb{E}\left[G^t\right] + \sum_{i=1}^n \hat{\beta}_1 \frac{1}{n^2 w_i} L_i^2 \cdot \mathbb{E}\left[\left\|x^{t+1} - x^t\right\|^2\right]$$

$$+ \frac{1}{n}\sum_{i=1}^n \tilde{A}\hat{\beta}_2 \cdot \mathbb{E}\left[f_i(x^{t+1}) - f_i^{\inf}\right] + \tilde{C}\hat{\beta}_2$$

$$\le (1-\hat{\theta})\mathbb{E}\left[G^t\right] + \sum_{i=1}^n \hat{\beta}_1 \frac{1}{n^2 w_i} L_i^2 \cdot \mathbb{E}\left[\left\|x^{t+1} - x^t\right\|^2\right]$$

$$+ \frac{1}{n}\sum_{i=1}^n \tilde{A}\hat{\beta}_2 \cdot \mathbb{E}\left[f_i(x^{t+1}) - f^{\inf}\right] + \tilde{C}\hat{\beta}_2$$

$$\le (1-\hat{\theta})\mathbb{E}\left[G^t\right] + \sum_{i=1}^n \hat{\beta}_1 \frac{1}{n^2 w_i} L_i^2 \cdot \mathbb{E}\left[\left\|x^{t+1} - x^t\right\|^2\right]$$

$$+ \tilde{A}\hat{\beta}_2 \mathbb{E}\left[f(x^{t+1}) - f^{\inf}\right] + \tilde{C}\hat{\beta}_2.$$

Given that $w_i = \frac{L_i}{\sum_j L_j}$, we have:

$$
\begin{aligned}
\mathbb{E}\left[G^{t+1}\right] &\leq (1-\hat{\theta})\mathbb{E}\left[G^t\right] + \frac{1}{n}\sum_{i=1}^{n}\hat{\beta}_1\frac{\sum_j L_j}{n}L_i\mathbb{E}\left[\left\|x^{t+1}-x^t\right\|^2\right] \\
&\quad + \widetilde{A}\hat{\beta}_2\mathbb{E}\left[f(x^{t+1})-f^{\mathrm{inf}}\right] + \widetilde{C}\hat{\beta}_2 \\
&= (1-\hat{\theta})\mathbb{E}\left[G^t\right] + \hat{\beta}_1\left(\frac{1}{n}\sum_{i=1}^{n}L_i\right)^2\cdot\mathbb{E}\left[\left\|x^{t+1}-x^t\right\|^2\right] \\
&\quad + \widetilde{A}\hat{\beta}_2\mathbb{E}\left[f(x^{t+1})-f^{\mathrm{inf}}\right] + \widetilde{C}\hat{\beta}_2,
\end{aligned}
$$

what completes the proof. $\qquad\square$

### E.3 MAIN RESULT

Now we are ready to prove the main convergence theorem.

**Theorem 8.** *Let $\mathcal{C}_i^t \in \mathbb{C}(\alpha)$ for all $\in [n]$ and $t \geq 0$ in Algorithm 3. Set the following quantities:*

$$
\begin{aligned}
\hat{\theta} &:= 1-(1-\alpha)(1+s)(1+\nu), \\
\hat{\beta}_1 &:= 2(1-\alpha)(1+s)\left(s+\nu^{-1}\right), \\
\hat{\beta}_2 &:= 2(1-\alpha)(1+s)(1+\nu^{-1})+(1+s^{-1}), \\
w_i &:= \frac{L_i}{\sum_j L_j}, \\
\widetilde{A} &:= \max_{i=1,\ldots,n}\left(\frac{2(A_i+L_i(B_i-1))}{\tau_i}\frac{1}{nw_i}\right), \\
\widetilde{C} &:= \max_{i=1,\ldots,n}\left(\frac{C_i}{\tau_i}\frac{1}{nw_i}\right).
\end{aligned}
$$

*Under Assumptions 1, 2, and 5, and selection of $s > 0$, $\mu > 0$ such that $(1+s)(1+\mu) < \frac{1}{1-\alpha}$ set the stepsize in the following way:*

$$
\gamma \leq \frac{1}{L + L_{\mathrm{AM}}\sqrt{\frac{\hat{\beta}_1}{\hat{\theta}}}}. \tag{48}
$$

*Choose an iterate $\hat{x}^T$ from $\{x^0, x^1, \ldots, x^{T-1}\}$ with probability*

$$
\mathbf{Prob}(\hat{x}^T = x^t) = \frac{v_t}{V_T}, \tag{49}
$$

*where*

$$
v_t := \left(1-\frac{\gamma\widetilde{A}\tilde{\beta}_2}{2\theta}\right)^t; \qquad V_T := \sum_{t=0}^{T-1}v_t.
$$

*Then,*

$$
\mathbb{E}\left[\left\|\nabla f(\hat{x}^T)\right\|^2\right] \leq \frac{2(f(x^0)-f^{inf})}{\gamma T\left(1-\frac{\gamma\widetilde{A}\tilde{\beta}_2}{2\theta}\right)^T} + \frac{G^0}{\hat{\theta}T\left(1-\frac{\gamma\widetilde{A}\tilde{\beta}_2}{2\theta}\right)^T} + \frac{\widetilde{C}\beta_2}{\hat{\theta}}, \tag{50}
$$

*where $G^0 := \sum_{i=1}^{n}w_i\|g_i^0 - \frac{1}{nw_i}\nabla f_i(x^0)\|^2$.*

*Proof.* In the derivation below, we use Lemma 3 for

$$g^t = \sum_{i=1}^{n} w_i g_i^t. \tag{51}$$

We start as follows:

$$
\begin{aligned}
f(x^{t+1}) &\overset{(20)}{\leq} f(x^t) - \frac{\gamma}{2} \left\| \nabla f(x^t) \right\|^2 - \left( \frac{1}{2\gamma} - \frac{L}{2} \right) \left\| x^{t+1} - x^t \right\|^2 + \frac{\gamma}{2} \left\| g^t - \sum_{i=1}^{n} \nabla f_i(x^t) \right\|^2 \\
&\overset{(51)}{=} f(x^t) - \frac{\gamma}{2} \left\| \nabla f(x^t) \right\|^2 - \left( \frac{1}{2\gamma} - \frac{L}{2} \right) \left\| x^{t+1} - x^t \right\|^2 + \frac{\gamma}{2} \left\| \sum_{i=1}^{n} w_i \left( g_i^t - \frac{\nabla f_i(x^t)}{n w_i} \right) \right\|^2 \\
&\leq f(x^t) - \frac{\gamma}{2} \left\| \nabla f(x^t) \right\|^2 - \left( \frac{1}{2\gamma} - \frac{L}{2} \right) \left\| x^{t+1} - x^t \right\|^2 + \frac{\gamma}{2} \sum_{i=1}^{n} w_i \left\| g_i^t - \frac{\nabla f_i(x^t)}{n w_i} \right\|^2 \\
&= f(x^t) - \frac{\gamma}{2} \left\| \nabla f(x^t) \right\|^2 - \left( \frac{1}{2\gamma} - \frac{L}{2} \right) \left\| x^{t+1} - x^t \right\|^2 + \frac{\gamma}{2} G^t. \tag{52}
\end{aligned}
$$

Subtracting $f^*$ from both sides and taking expectation, we get

$$
\begin{aligned}
\mathbb{E}\left[ f(x^{t+1}) - f^* \right] \leq\ & \mathbb{E}\left[ f(x^t) - f^* \right] - \frac{\gamma}{2} \mathbb{E}\left[ \left\| \nabla f(x^t) \right\|^2 \right] \\
& - \left( \frac{1}{2\gamma} - \frac{L}{2} \right) \mathbb{E}\left[ \left\| x^{t+1} - x^t \right\|^2 \right] + \frac{\gamma}{2} \mathbb{E}\left[ G^t \right]. \tag{53}
\end{aligned}
$$

Let $\delta^t := \mathbb{E}\left[ f(x^t) - f^* \right]$, $s^t := \mathbb{E}\left[ G^t \right]$ and $r^t := \mathbb{E}\left[ \left\| x^{t+1} - x^t \right\|^2 \right]$. Then by adding $\frac{\gamma}{2\hat{\theta}} s^{t+1}$ and employing (46), we obtain:

$$
\begin{aligned}
\delta^{t+1} + \frac{\gamma}{2\hat{\theta}} s^{t+1} \leq\ & \delta^t - \frac{\gamma}{2} \mathbb{E}\left[ \left\| \nabla f(x^t) \right\|^2 \right] - \left( \frac{1}{2\gamma} - \frac{L}{2} \right) r^t + \frac{\gamma}{2} s^t \\
& + \frac{\gamma}{2\hat{\theta}} \left( \hat{\beta}_1 L_{\mathrm{AM}}^2 r^t + (1-\hat{\theta}) s^t + \widetilde{A} \hat{\beta}_2 \delta^{t+1} + \widetilde{C} \hat{\beta}_2 \right) \\
=\ & \delta^t + \frac{\gamma}{2\hat{\theta}} s^t - \frac{\gamma}{2} \mathbb{E}\left[ \left\| \nabla f(x^t) \right\|^2 \right] - \left( \frac{1}{2\gamma} - \frac{L}{2} - \frac{\gamma}{2\hat{\theta}} \hat{\beta}_1 L_{\mathrm{AM}}^2 \right) r^t + \frac{\gamma \widetilde{A} \hat{\beta}_2}{2\hat{\theta}} \delta^{t+1} + \frac{\gamma \widetilde{C}}{2\hat{\theta}} \hat{\beta}_2 \\
\leq\ & \delta^t + \frac{\gamma}{2\hat{\theta}} s^t - \frac{\gamma}{2} \mathbb{E}\left[ \left\| \nabla f(x^t) \right\|^2 \right] + \frac{\gamma \widetilde{A} \hat{\beta}_2}{2\hat{\theta}} \delta^{t+1} + \frac{\gamma \widetilde{C}}{2\hat{\theta}} \hat{\beta}_2.
\end{aligned}
$$

The last inequality follows from the bound $\gamma^2 \frac{\hat{\beta}_1 L_{\mathrm{AM}}^2}{\hat{\theta}} + L\gamma \leq 1$, which holds due to Lemma 5 for $\gamma \leq \frac{1}{L + L_{\mathrm{AM}} \sqrt{\frac{\hat{\beta}_1}{\hat{\theta}}}}$. Subsequently, we will reconfigure the final inequality and perform algebraic manipulations, taking into account that $\frac{2}{\gamma} > 0$. In the final step of these algebraic transformations, we will leverage the fact that $s^t \geq 0$:

$$
\delta^{t+1} + \frac{\gamma}{2\hat{\theta}} s^{t+1} \leq\ \delta^t + \frac{\gamma}{2\hat{\theta}} s^t - \frac{\gamma}{2} \mathbb{E}\left[ \left\| \nabla f(x^t) \right\|^2 \right] + \frac{\gamma \widetilde{A} \hat{\beta}_2}{2\hat{\theta}} \delta^{t+1} + \frac{\gamma \widetilde{C}}{2\hat{\theta}} \hat{\beta}_2.
$$

Therefore,

$$
\frac{2}{\gamma} \delta^{t+1} + \frac{2}{\gamma} \frac{\gamma}{2\hat{\theta}} s^{t+1} \leq\ \frac{2}{\gamma} \delta^t + \frac{2}{\gamma} \frac{\gamma}{2\hat{\theta}} s^t - \mathbb{E}\left[ \left\| \nabla f(x^t) \right\|^2 \right] + \frac{2}{\gamma} \frac{\gamma \widetilde{A} \hat{\beta}_2}{2\hat{\theta}} \delta^{t+1} + \frac{2}{\gamma} \frac{\gamma \widetilde{C}}{2\hat{\theta}} \hat{\beta}_2.
$$

Further,

$$
\begin{aligned}
\mathbb{E}\left[\left\|\nabla f(x^t)\right\|^2\right] &\leq -\frac{2}{\gamma}\delta^{t+1} - \frac{2}{\gamma}\frac{\gamma}{2\hat{\theta}}s^{t+1} + \frac{2}{\gamma}\delta^t + \frac{2}{\gamma}\frac{\gamma}{2\hat{\theta}}s^t + \frac{2}{\gamma}\frac{\gamma\widetilde{A}\beta_2}{2\hat{\theta}}\delta^{t+1} + \frac{2}{\gamma}\frac{\gamma\widetilde{C}}{2\hat{\theta}}\beta_2 \\
&\leq -\frac{2}{\gamma}\delta^{t+1} - \frac{2}{\gamma}\frac{\gamma}{2\hat{\theta}}s^{t+1} + \frac{2}{\gamma}\left(\delta^t + \frac{\gamma}{2\hat{\theta}}s^t\right) + \frac{2}{\gamma}\frac{\gamma\widetilde{A}\beta_2}{2\hat{\theta}}\delta^{t+1} + \frac{\widetilde{C}\beta_2}{\hat{\theta}} \\
&\leq \frac{2}{\gamma}\left(\left(\delta^t + \frac{\gamma}{2\hat{\theta}}s^t\right) - 1\left(1 - \frac{\gamma\widetilde{A}\beta_2}{2\hat{\theta}}\right)\delta^{t+1} - \left(\frac{\gamma}{2\hat{\theta}}s^{t+1}\right)\right) + \frac{\widetilde{C}\beta_2}{\hat{\theta}} \\
&\leq \frac{2}{\gamma}\left(\left(\delta^t + \frac{\gamma}{2\hat{\theta}}s^t\right) - \left(1 - \frac{\gamma\widetilde{A}\beta_2}{2\hat{\theta}}\right)\left(\delta^{t+1} + \frac{\gamma}{2\hat{\theta}}s^{t+1}\right)\right) + \frac{\widetilde{C}\beta_2}{\hat{\theta}}.
\end{aligned}
$$

We sum up inequalities above with weights $v_t/V_T$, where $v_t := (1 - \frac{\gamma\widetilde{A}\hat{\beta}_2}{2\theta})^t$ and $V_T := \sum_{i=1}^{T} v_i$:

$$
\begin{aligned}
\mathbb{E}\left[\left\|\nabla f(\hat{x}^T)\right\|^2\right] &= \sum_{t=0}^{T}\frac{v_t}{V_T}\mathbb{E}\left[\left\|\nabla f(x^t)\right\|^2\right] \\
&= \frac{1}{V_T}\sum_{t=0}^{T}v_t\mathbb{E}\left[\left\|\nabla f(x^t)\right\|^2\right] \\
&\leq \frac{1}{V_T}\sum_{t=0}^{T}v_t\left(\frac{2}{\gamma}\left(\left(\delta^t + \frac{\gamma}{2\hat{\theta}}s^t\right) - \left(1 - \frac{\gamma\widetilde{A}\beta_2}{2\hat{\theta}}\right)\left(\delta^{t+1} + \frac{\gamma}{2\hat{\theta}}s^{t+1}\right)\right) + \frac{\widetilde{C}\beta_2}{\hat{\theta}}\right) \\
&= \frac{2}{\gamma V_T}\sum_{t=0}^{T}w_t\left(\left(\delta^t + \frac{\gamma}{2\hat{\theta}}s^t\right) - \left(1 - \frac{\gamma\widetilde{A}\beta_2}{2\hat{\theta}}\right)\left(\delta^{t+1} + \frac{\gamma}{2\hat{\theta}}s^{t+1}\right)\right) + \sum_{t=0}^{T}\frac{w_t}{W_T}\cdot\frac{\widetilde{C}\beta_2}{\hat{\theta}} \\
&= \frac{2}{\gamma V_T}\sum_{t=0}^{T}w_t\left(\left(\delta^t + \frac{\gamma}{2\hat{\theta}}s^t\right) - \left(1 - \frac{\gamma\widetilde{A}\beta_2}{2\hat{\theta}}\right)\left(\delta^{t+1} + \frac{\gamma}{2\hat{\theta}}s^{t+1}\right)\right) + \frac{\widetilde{C}\beta_2}{\hat{\theta}} \\
&= \frac{2}{\gamma V_T}\sum_{t=0}^{T}\left(w_t\left(\delta^t + \frac{\gamma}{2\hat{\theta}}s^t\right) - w_{t+1}\left(\delta^{t+1} + \frac{\gamma}{2\hat{\theta}}s^{t+1}\right)\right) + \frac{\widetilde{C}\beta_2}{\hat{\theta}} \\
&\leq \frac{2\delta^0}{\gamma V_T} + \frac{s^0}{\hat{\theta}V_T} + \frac{\widetilde{C}\beta_2}{\hat{\theta}}.
\end{aligned}
$$

Finally, we notice that $V_T = \sum_{t=1}^{T}(1 - \frac{\gamma\widetilde{A}\hat{\beta}_2}{2\theta})^t \geq T\cdot(1 - \frac{\gamma\widetilde{A}\hat{\beta}_2}{2\theta})^T$, what concludes the proof. $\square$

## F EF21-W-SGD: WEIGHTED ERROR FEEDBACK 2021 WITH STOCHASTIC GRADIENTS UNDER THE ABC ASSUMPTION

In this section, we present the convergence result for Weighted EF21 in the setting where the gradient computation on the clients is replaced with a pretty general unbiased stochastic gradient estimator.

### F.1 ALGORITHM

The EF21-W algorithm assumes that all clients can compute the exact gradient in each round. In some scenarios, the exact gradients may be unavailable or too costly to compute, and only approximate gradient estimators can be obtained. To have the ability for EF21-W to work in such circumstances we extended EF21-W to handle stochastic gradients. We called the resulting algorithm EF21-W-SGD (Algorithm 4).

---

**Algorithm 4** EF21-W-SGD: Weighted EF-21 with Stochastic Gradients under ABC assumption

---

1: **Input:** initial model $x^0 \in \mathbb{R}^d$; initial gradient estimates $g_1^0, g_2^0, \ldots, g_n^0 \in \mathbb{R}^d$ stored at the server and the clients; stepsize $\gamma > 0$; number of iterations $T > 0$; weights $w_i = \frac{L_i}{\sum_j L_j}$ for $i \in [n]$

2: **Initialize:** $g^0 = \sum_{i=1}^n w_i g_i^0$ on the server

3: **for** $t = 0, 1, 2, \ldots, T - 1$ **do**

4:     Server computes $x^{t+1} = x^t - \gamma g^t$ and broadcasts $x^{t+1}$ to all $n$ clients

5:     **for** $i = 1, \ldots, n$ **on the clients in parallel do**

6:         Compute a stochastic gradient $\hat{g}_i(x^{t+1})$ estimator of the gradient $\nabla f_i(x^{t+1})$

7:         Compute $u_i^t = \mathcal{C}_i^t \left( \frac{1}{n w_i} \hat{g}_i(x^{t+1}) - g_i^t \right)$ and update $g_i^{t+1} = g_i^t + u_i^t$

8:         Send the compressed message $u_i^t$ to the server

9:     **end for**

10:    Server updates $g_i^{t+1} = g_i^t + u_i^t$ for all $i \in [n]$, and computes $g^{t+1} = \sum_{i=1}^n w_i g_i^{t+1}$

11: **end for**

12: **Output:** Point $\hat{x}^T$ chosen from the set $\{x^0, \ldots, x^{T-1}\}$ randomly according to the law (59)

---

Our analysis of this extension follows a similar approach as the one used by Fatkhullin et al. (2021) for studying the stochastic gradient version under the name EF21-SGD. However, EF21-W-SGD has four important differences with vanilla EF21-SGD:

1. Vanilla EF21-SGD algorithm analyzed by Fatkhullin et al. (2021) worked under a specific sampling schema for a stochastic gradient estimator. Our analysis works under a more general ABC Assumption 6.

2. Vanilla EF21-SGD provides maximum theoretically possible $\gamma = \left( L + L_{\mathrm{QM}} \sqrt{\frac{\beta_1}{\theta}} \right)^{-1}$, where EF21-W-SGD has $\gamma = \left( L + L_{\mathrm{AM}} \sqrt{\frac{\beta_1}{\theta}} \right)^{-1}$.

3. In contrast to the original analysis Vanilla EF21-SGD our analysis provides a more aggressive $\beta_1$ parameter which is smaller by a factor of 2.

4. Vanilla EF21-SGD and EF21-W-SGD formally differs in a way how it reports iterate $x^T$ which minimizes $\mathbb{E}\left[ \left\| \nabla f(x^T) \right\|^2 \right]$ due to a slightly different definition of $\widetilde{A}$. The EF21-W-SGD (Algorithm 4) requires output iterate $\hat{x}^T$ randomly according to the probability mass function described by Equation (59).

**Assumption 6** (General assumption for stochastic gradient estimators). *We assume that for all $i \in [n]$ there exist parameters $A_i, C_i \geq 0$, $B_i \geq 1$ such that*

$$\mathbb{E}\left[\|\nabla \hat{g}_i(x)\|^2\right] \leq 2A_i\left(f_i(x) - f_i^{\text{inf}}\right) + B_i \|\nabla f_i(x)\|^2 + C_i, \tag{54}$$

*holds for all $x \in \mathbb{R}^d$, where[4] $f_i^{\text{inf}} = \inf_{x \in \mathbb{R}^d} f_i(x) > -\infty$.*

**Assumption 7** (Unbiased assumption for stochastic gradient estimators). *We assume that for all $i \in [n]$ there following holds for all $x \in \mathbb{R}^d$:*

$$\mathbb{E}\left[\hat{g}_i(x)\right] = \nabla f_i(x).$$

We study EF21-W-SGD under Assumption 6 and Assumption 7. To the best of our knowledge, this Assumption 6, which was originally presented as Assumption 2 by Khaled & Richtárik (2022), is the most general assumption for a stochastic gradient estimator in a non-convex setting. For a detailed explanation of the generality of this assumption see Figure 1 of Khaled & Richtárik (2022).

### F.2 A LEMMA

The contraction lemma in this case gets the following form:

**Lemma 11.** *Let $\mathcal{C}_i^t \in \mathbb{C}(\alpha)$ for all $i \in [n]$ and $t \geq 0$. Define*

$$G_i^t := \left\|g_i^t - \frac{\nabla f_i(x^t)}{nw_i}\right\|^2, \qquad G^t := \sum_{i=1}^n w_i G_i^t.$$

*Let Assumptions 2, 6, 7 hold. Then, for any $s > 0, \nu > 0$ during execution of the Algorithm 4 the following holds:*

$$\mathbb{E}\left[G^{t+1}\right] \leq (1 - \hat{\theta})\mathbb{E}\left[G^t\right] + \hat{\beta}_1 L_{\text{AM}}^2 \mathbb{E}\left[\left\|x^{t+1} - x^t\right\|^2\right] + \widetilde{A}\hat{\beta}_2 \mathbb{E}\left[f(x^{t+1}) - f^{\text{inf}}\right] + \widetilde{C}\hat{\beta}_2, \tag{55}$$

*where*

$$
\begin{aligned}
w_i &:= \frac{L_i}{\sum_j L_j}, \\
\hat{\theta} &:= 1 - (1 - \alpha)(1 + s)(1 + \nu) \\
\hat{\beta}_1 &:= (1 - \alpha)(1 + s)\left(s + \nu^{-1}\right), \\
\hat{\beta}_2 &:= (1 - \alpha)(1 + s) + (1 + s^{-1}), \\
\widetilde{A} &:= \max_{i=1,\dots,n}\left(2(A_i + L_i(B_i - 1))\frac{1}{nw_i}\right), \\
\widetilde{C} &:= \max_{i=1,\dots,n}\left(C_i \frac{1}{nw_i}\right).
\end{aligned}
$$

---

[4]When $A_i = 0$ one can ignore the first term in the right-hand side of (54), i.e., assumption $\inf_{x \in \mathbb{R}^d} f_i(x) > -\infty$ is not required in this case.

*Proof.* Define $W^t := \{g_1^t, \ldots, g_n^t, x^t, x^{t+1}\}$. The proof starts as follows:

$$
\mathbb{E}\left[G_i^{t+1} \mid W^t\right] \stackrel{(24)}{=} \mathbb{E}\left[\left\|g_i^{t+1} - \frac{\nabla f_i(x^{t+1})}{nw_i}\right\|^2 \mid W^t\right]
$$

$$
\stackrel{\text{line } 7}{=} \mathbb{E}\left[\left\|g_i^t + \mathcal{C}_i^t\left(\frac{\hat{g}_i(x^{t+1})}{nw_i} - g_i^t\right) - \frac{\nabla f_i(x^{t+1})}{nw_i}\right\|^2 \mid W^t\right]
$$

$$
= \mathbb{E}\left[\left\|\mathcal{C}_i^t\left(\frac{\hat{g}_i(x^{t+1})}{nw_i} - g_i^t\right) - \left(\frac{\hat{g}_i(x^{t+1})}{nw_i} - g_i^t\right) + \frac{\hat{g}_i(x^{t+1})}{nw_i} - \frac{\nabla f_i(x^{t+1})}{nw_i}\right\|^2 \mid W^t\right]
$$

$$
\stackrel{(21)}{\leq} (1+s)\mathbb{E}\left[\left\|\mathcal{C}_i^t\left(\frac{\hat{g}_i(x^{t+1})}{nw_i} - g_i^t\right) - \left(\frac{\hat{g}_i(x^{t+1})}{nw_i} - g_i^t\right)\right\|^2 \mid W^t\right]
$$

$$
+ (1+s^{-1})\mathbb{E}\left[\left\|\frac{\hat{g}_i(x^{t+1})}{nw_i} - \frac{\nabla f_i(x^{t+1})}{nw_i}\right\|^2 \mid W^t\right]
$$

$$
\stackrel{(4)}{\leq} (1-\alpha)(1+s)\mathbb{E}\left[\left\|\left(\frac{\hat{g}_i(x^{t+1})}{nw_i} - \frac{\nabla f_i(x^{t+1})}{nw_i}\right) + \left(\frac{\nabla f_i(x^{t+1})}{nw_i} - g_i^t\right)\right\|^2 \mid W^t\right]
$$

$$
+ (1+s^{-1})\mathbb{E}\left[\left\|\frac{\hat{g}_i(x^{t+1})}{nw_i} - \frac{\nabla f_i(x^{t+1})}{nw_i}\right\|^2 \mid W^t\right]
$$

$$
= (1-\alpha)(1+s)\mathbb{E}\left[\left\|g_i^t - \frac{\nabla f_i(x^{t+1})}{nw_i}\right\|^2 \mid W^t\right]
$$

$$
+ (1-\alpha)(1+s)\mathbb{E}\left[\left\|\frac{\nabla f_i(x^{t+1})}{nw_i} - \frac{\hat{g}_i(x^{t+1})}{nw_i}\right\|^2 \mid W^t\right]
$$

$$
+ (1+s^{-1})\mathbb{E}\left[\left\|\frac{\hat{g}_i(x^{t+1})}{nw_i} - \frac{\nabla f_i(x^{t+1})}{nw_i}\right\|^2 \mid W^t\right]
$$

$$
= (1-\alpha)(1+s)\mathbb{E}\left[\left\|g_i^t - \frac{\nabla f_i(x^t)}{nw_i} + \frac{\nabla f_i(x^t)}{nw_i} - \frac{\nabla f_i(x^{t+1})}{nw_i}\right\|^2 \mid W^t\right]
$$

$$
+ (1-\alpha)(1+s)\mathbb{E}\left[\left\|\frac{\nabla f_i(x^{t+1})}{nw_i} - \frac{\hat{g}_i(x^{t+1})}{nw_i}\right\|^2 \mid W^t\right]
$$

$$
+ (1+s^{-1})\mathbb{E}\left[\left\|\frac{\hat{g}_i(x^{t+1})}{nw_i} - \frac{\nabla f_i(x^{t+1})}{nw_i}\right\|^2 \mid W^t\right].
$$

Further, we continue as follows

$$\mathbb{E}\left[G_i^{t+1} \mid W^t\right] \overset{(21)}{\leq} (1-\alpha)(1+s)(1+\nu)\mathbb{E}\left[\left\|g_i^{\,t} - \frac{\nabla f_i(x^t)}{nw_i}\right\|^2 \mid W^t\right]$$

$$+(1-\alpha)(1+s)(1+\nu^{-1})\left\|\frac{\nabla f_i(x^{t+1})}{nw_i} - \frac{\nabla f_i(x^t)}{nw_i}\right\|^2$$

$$+\left((1+s^{-1}) + (1-\alpha)(1+s)\right)\mathbb{E}\left[\left\|\frac{\hat{g}_i(x^{t+1})}{nw_i} - \frac{\nabla f_i(x^{t+1})}{nw_i}\right\|^2 \mid W^t\right].$$

To further bound the last term, which contains multiple $(1+s^{-1})$ factors, we leverage the property that $\hat{g}_i(x^{t+1})$ is a random variable serving as an unbiased estimator of $\nabla f_i(x^{t+1})$. Our approach is as follows:

$$\mathbb{E}\left[G_i^{t+1} \mid W^t\right] \leq (1-\hat{\theta})\mathbb{E}\left[G_i^t \mid W^t\right] + \hat{\beta}_1 \frac{1}{n^2 w_i^2}\left\|\nabla f_i(x^{t+1}) - \nabla f_i(x^t)\right\|^2$$

$$+\frac{\hat{\beta}_2}{(nw_i)^2}\mathbb{E}\left[\left\|\hat{g}_i(x^{t+1}) - \nabla f_i(x^{t+1})\right\|^2 \mid W^t\right].$$

Now due to the requirement of unbiasedness of gradient estimators expressed in the form of Assumption 7 we have the following:

$$\mathbb{E}\left[\left\|\hat{g}_i(x^{t+1}) - \nabla f_i(x^{t+1})\right\|^2 \mid W^t\right] = \mathbb{E}\left[\left\|\hat{g}_i(x^{t+1})\right\|^2 \mid W^t\right] - \left\|\nabla f_i(x^{t+1})\right\|^2 \tag{56}$$

Using this variance decomposition, we can proceed as follows.

$$
\begin{aligned}
\mathbb{E}\left[G_i^{t+1} \mid W^t\right] \overset{(56)}{\leq}\ & (1-\hat{\theta})\mathbb{E}\left[G_i^t \mid W^t\right] + \hat{\beta}_1 \frac{1}{n^2 w_i^2}\left\|\nabla f_i(x^{t+1}) - \nabla f_i(x^t)\right\|^2 \\
& + \frac{\hat{\beta}_2}{(nw_i)^2}\left(\mathbb{E}\left[\|\hat{g}_i(x^{t+1})\|^2 \mid W^t\right] - \|\nabla f_i(x^{t+1})\|^2\right) \\
\overset{(54)}{\leq}\ & (1-\hat{\theta})\mathbb{E}\left[G_i^t \mid W^t\right] + \hat{\beta}_1 \frac{1}{n^2 w_i^2}\left\|\nabla f_i(x^{t+1}) - \nabla f_i(x^t)\right\|^2 \\
& + \frac{\hat{\beta}_2}{(nw_i)^2}\left(2A_i\left(f_i(x^{t+1}) - f_i^{\mathrm{inf}}\right) + B_i\|\nabla f_i(x^{t+1})\|^2 + C_i - \|\nabla f_i(x^{t+1}\|^2)\right) \\
=\ & (1-\hat{\theta})\mathbb{E}\left[G_i^t \mid W^t\right] + \hat{\beta}_1 \frac{1}{n^2 w_i^2}\left\|\nabla f_i(x^{t+1}) - \nabla f_i(x^t)\right\|^2 \\
& + \frac{2A_i\hat{\beta}_2}{(nw_i)^2}\left(f_i(x^{t+1}) - f_i^{\mathrm{inf}}\right) + \frac{2(B_i - 1)\hat{\beta}_2}{(nw_i)^2}\left(\frac{1}{2}\|\nabla f_i(x^{t+1})\|^2\right) + \frac{C_i\hat{\beta}_2}{(nw_i)^2} \\
\leq\ & (1-\hat{\theta})\mathbb{E}\left[G_i^t \mid W^t\right] + \hat{\beta}_1 \frac{1}{n^2 w_i^2}\left\|\nabla f_i(x^{t+1}) - \nabla f_i(x^t)\right\|^2 \\
& + \frac{2A_i\hat{\beta}_2}{(nw_i)^2}\left(f_i(x^{t+1}) - f_i^{\mathrm{inf}}\right) + \frac{2(B_i - 1)\hat{\beta}_2}{(nw_i)^2}L_i\left(f_i(x^{t+1}) - f_i^{\mathrm{inf}}\right) + \frac{C_i\hat{\beta}_2}{(nw_i)^2} \\
=\ & (1-\hat{\theta})\mathbb{E}\left[G_i^t \mid W^t\right] + \hat{\beta}_1 \frac{1}{n^2 w_i^2}\left\|\nabla f_i(x^{t+1}) - \nabla f_i(x^t)\right\|^2 \\
& + \frac{2(A_i + L_i(B_i - 1))\hat{\beta}_2}{(nw_i)^2}\left(f_i(x^{t+1}) - f_i^{\mathrm{inf}}\right) + \frac{C_i\hat{\beta}_2}{(nw_i)^2}.
\end{aligned}
$$

Next leveraging Assumption 2 we replace the second term in the last expression, and we can derive the subsequent bound:

$$
\begin{aligned}
\mathbb{E}\left[G_i^{t+1} \mid W^t\right] \overset{(7)}{\leq}\ & (1-\hat{\theta})G_i^t + \frac{\hat{\beta}_1 L_i^2}{n^2 w_i^2}\left\|x^{t+1} - x^t\right\|^2 \\
& + \frac{2(A_i + L_i(B_i - 1))\hat{\beta}_2}{(nw_i)^2}\left(f_i(x^{t+1}) - f_i^{\mathrm{inf}}\right) + \frac{C_i\hat{\beta}_2}{(nw_i)^2}.
\end{aligned}
$$

Applying the tower property and subsequently taking the expectation, we obtain:

$$
\begin{aligned}
\mathbb{E}\left[G_i^{t+1}\right] \leq\ & (1-\hat{\theta})\mathbb{E}\left[G_i^t\right] + \hat{\beta}_1 \frac{1}{n^2 w_i^2} L_i^2 \mathbb{E}\left[\left\|x^{t+1} - x^t\right\|^2\right] \\
& + \frac{2(A_i + L_i(B_i - 1))\hat{\beta}_2}{(nw_i)^2}\mathbb{E}\left[f_i(x^{t+1}) - f_i^{\mathrm{inf}}\right] + \frac{C_i\hat{\beta}_2}{(nw_i)^2}.
\end{aligned}
\tag{57}
$$

Next for the expectation of the main quantity of our interest $G^{t+1}$, we derive the subsequent bound:

$$
\begin{aligned}
\mathbb{E}\left[G^{t+1}\right] &= \mathbb{E}\left[\sum_{i=1}^{n} w_i G_i^{t+1}\right] \\
&= \sum_{i=1}^{n} w_i \mathbb{E}\left[G_i^{t+1}\right] \\
&\overset{(57)}{\leq} (1-\hat{\theta})\sum_{i=1}^{n} w_i \mathbb{E}\left[G_i^t\right] + \sum_{i=1}^{n} w_i \hat{\beta}_1 \frac{1}{n^2 w_i^2} L_i^2 \cdot \mathbb{E}\left[\left\|x^{t+1}-x^t\right\|^2\right] \\
&\quad + \sum_{i=1}^{n} w_i \frac{2(A_i+L_i(B_i-1))\hat{\beta}_2}{(nw_i)^2}\cdot \mathbb{E}\left[f_i(x^{t+1})-f_i^{\inf}\right] + \sum_{i=1}^{n} w_i \frac{C_i\hat{\beta}_2}{(nw_i)^2} \\
&= (1-\hat{\theta})\mathbb{E}\left[G^t\right] + \sum_{i=1}^{n} \hat{\beta}_1 \frac{1}{n^2 w_i} L_i^2 \cdot \mathbb{E}\left[\left\|x^{t+1}-x^t\right\|^2\right] \\
&\quad + \sum_{i=1}^{n} \frac{2(A_i+L_i(B_i-1))\hat{\beta}_2}{(n)^2 w_i}\cdot \mathbb{E}\left[f_i(x^{t+1})-f_i^{\inf}\right] + \sum_{i=1}^{n} \frac{C_i\hat{\beta}_2}{n^2 w_i}
\end{aligned}
$$

Employing quantities $\tilde{A}$ and $\tilde{C}$, the final bound can be reformulated as follows:

$$
\begin{aligned}
\mathbb{E}\left[G^{t+1}\right] &\leq (1-\hat{\theta})\mathbb{E}\left[G^t\right] + \sum_{i=1}^{n} \hat{\beta}_1 \frac{1}{n^2 w_i} L_i^2 \cdot \mathbb{E}\left[\left\|x^{t+1}-x^t\right\|^2\right] \\
&\quad + \frac{1}{n}\sum_{i=1}^{n} \widetilde{A}\hat{\beta}_2 \cdot \mathbb{E}\left[f_i(x^{t+1})-f_i^{\inf}\right] + \widetilde{C}\hat{\beta}_2 \\
&\leq (1-\hat{\theta})\mathbb{E}\left[G^t\right] + \sum_{i=1}^{n} \hat{\beta}_1 \frac{1}{n^2 w_i} L_i^2 \cdot \mathbb{E}\left[\left\|x^{t+1}-x^t\right\|^2\right] \\
&\quad + \frac{1}{n}\sum_{i=1}^{n} \widetilde{A}\hat{\beta}_2 \cdot \mathbb{E}\left[f_i(x^{t+1})-f^{\inf}\right] + \widetilde{C}\hat{\beta}_2 \\
&\leq (1-\hat{\theta})\mathbb{E}\left[G^t\right] + \sum_{i=1}^{n} \hat{\beta}_1 \frac{1}{n^2 w_i} L_i^2 \cdot \mathbb{E}\left[\left\|x^{t+1}-x^t\right\|^2\right] \\
&\quad + \widetilde{A}\hat{\beta}_2 \mathbb{E}\left[f(x^{t+1})-f^{\inf}\right] + \widetilde{C}\hat{\beta}_2.
\end{aligned}
$$

Given that $w_i = \frac{L_i}{\sum_j L_j}$, we have:

$$
\begin{aligned}
\mathbb{E}\left[G^{t+1}\right] &\leq (1-\hat{\theta})\mathbb{E}\left[G^t\right] + \frac{1}{n}\sum_{i=1}^{n} \hat{\beta}_1 \frac{\sum_j L_j}{n} L_i \mathbb{E}\left[\left\|x^{t+1}-x^t\right\|^2\right] \\
&\quad + \widetilde{A}\hat{\beta}_2 \mathbb{E}\left[f(x^{t+1})-f^{\inf}\right] + \widetilde{C}\hat{\beta}_2 \\
&= (1-\hat{\theta})\mathbb{E}\left[G^t\right] + \hat{\beta}_1\left(\frac{1}{n}\sum_{i=1}^{n} L_i\right)^2 \cdot \mathbb{E}\left[\left\|x^{t+1}-x^t\right\|^2\right] \\
&\quad + \widetilde{A}\hat{\beta}_2 \mathbb{E}\left[f(x^{t+1})-f^{\inf}\right] + \widetilde{C}\hat{\beta}_2,
\end{aligned}
$$

what completes the proof. $\qquad\square$

### F.3   MAIN RESULT

Now we are ready to prove the main convergence theorem.

**Theorem 9.** *Let $\mathcal{C}_i^t \in \mathbb{C}(\alpha)$ for all $\in [n]$ and $t \geq 0$ in Algorithm 4. set the following quantities:*

$$
\begin{aligned}
\hat{\theta} &:= 1 - (1 - \alpha)(1 + s)(1 + \nu), \\
\hat{\beta}_1 &:= (1 - \alpha)(1 + s)\left(s + \nu^{-1}\right), \\
\hat{\beta}_2 &:= (1 - \alpha)(1 + s) + (1 + s^{-1}), \\
w_i &:= \frac{L_i}{\sum_{j=1}^n L_j}, \\
\widetilde{A} &:= \max_{i=1,\ldots,n} \frac{2(A_i + L_i(B_i - 1))}{nw_i}, \\
\widetilde{C} &:= \max_{i=1,\ldots,n} \frac{C_i}{nw_i}.
\end{aligned}
$$

*Under Assumptions 1, 2, 6, 7, and selection of $s > 0, \nu > 0$ small enough such that $(1+s)(1+\nu) < \frac{1}{1-\alpha}$ holds, set the stepsize in the following way:*

$$
\gamma \leq \frac{1}{L + L_{\mathrm{AM}}\sqrt{\frac{\hat{\beta}_1}{\hat{\theta}}}}. \tag{58}
$$

*Choose an iterate $\hat{x}^T$ from $\{x^0, x^1, \ldots, x^{T-1}\}$ with probability*

$$
\mathbf{Prob}(\hat{x}^T = x^t) = \frac{v_t}{V_T}, \tag{59}
$$

*where*

$$
v_t := \left(1 - \frac{\gamma \tilde{A} \tilde{\beta}_2}{2\theta}\right)^t; \qquad V_T := \sum_{t=0}^{T-1} v_t.
$$

*Then,*

$$
\mathbb{E}\left[\left\|\nabla f(\hat{x}^T)\right\|^2\right] \leq \frac{2(f(x^0) - f^{inf})}{\gamma T \left(1 - \frac{\gamma \tilde{A} \hat{\beta}_2}{2\theta}\right)^T} + \frac{G^0}{\hat{\theta} T \left(1 - \frac{\gamma \tilde{A} \hat{\beta}_2}{2\theta}\right)^T} + \frac{\widetilde{C}\beta_2}{\hat{\theta}}, \tag{60}
$$

*where $G^0 := \sum_{i=1}^n w_i \left\| g_i^0 - \frac{1}{nw_i}\nabla f_i(x^0) \right\|^2$.*

*Proof.* In the derivation below, we use Lemma 3 for

$$
g^t = \sum_{i=1}^n w_i g_i^t. \tag{61}
$$

We start as follows:

$$
\begin{aligned}
f(x^{t+1}) \;\overset{(20)}{\leq}\;& f(x^t) - \frac{\gamma}{2}\left\|\nabla f(x^t)\right\|^2 - \left(\frac{1}{2\gamma} - \frac{L}{2}\right)\left\|x^{t+1} - x^t\right\|^2 + \frac{\gamma}{2}\left\|g^t - \frac{1}{n}\sum_{i=1}^n \nabla f_i(x^t)\right\|^2 \\
\overset{(51)}{=}\;& f(x^t) - \frac{\gamma}{2}\left\|\nabla f(x^t)\right\|^2 - \left(\frac{1}{2\gamma} - \frac{L}{2}\right)\left\|x^{t+1} - x^t\right\|^2 + \frac{\gamma}{2}\left\|\sum_{i=1}^n w_i\left(g_i^t - \frac{\nabla f_i(x^t)}{nw_i}\right)\right\|^2 \\
\leq\;& f(x^t) - \frac{\gamma}{2}\left\|\nabla f(x^t)\right\|^2 - \left(\frac{1}{2\gamma} - \frac{L}{2}\right)\left\|x^{t+1} - x^t\right\|^2 + \frac{\gamma}{2}\sum_{i=1}^n w_i\left\|g_i^t - \frac{\nabla f_i(x^t)}{nw_i}\right\|^2 \\
=\;& f(x^t) - \frac{\gamma}{2}\left\|\nabla f(x^t)\right\|^2 - \left(\frac{1}{2\gamma} - \frac{L}{2}\right)\left\|x^{t+1} - x^t\right\|^2 + \frac{\gamma}{2}G^t.
\end{aligned}
$$

Subtracting $f^*$ from both sides and taking expectation, we get

$$
\mathbb{E}\left[f(x^{t+1}) - f^*\right] \leq \mathbb{E}\left[f(x^t) - f^*\right] - \frac{\gamma}{2}\mathbb{E}\left[\left\|\nabla f(x^t)\right\|^2\right] - \left(\frac{1}{2\gamma} - \frac{L}{2}\right)\mathbb{E}\left[\left\|x^{t+1} - x^t\right\|^2\right] + \frac{\gamma}{2}\mathbb{E}\left[G^t\right].
$$

Let $\delta^t := \mathbb{E}\left[f(x^t) - f^*\right]$, $s^t := \mathbb{E}\left[G^t\right]$ and $r^t := \mathbb{E}\left[\left\|x^{t+1} - x^t\right\|^2\right]$. Then by adding $\frac{\gamma}{2\hat{\theta}}s^{t+1}$ and employing inequality (46), we obtain:

$$
\begin{aligned}
\delta^{t+1} + \frac{\gamma}{2\hat{\theta}}s^{t+1} \leq\;& \delta^t - \frac{\gamma}{2}\mathbb{E}\left[\left\|\nabla f(x^t)\right\|^2\right] - \left(\frac{1}{2\gamma} - \frac{L}{2}\right)r^t + \frac{\gamma}{2}s^t \\
& + \frac{\gamma}{2\hat{\theta}}\left(\hat{\beta}_1 L_{\mathrm{AM}}^2 r^t + (1-\hat{\theta})s^t + \widetilde{A}\hat{\beta}_2\delta^{t+1} + \widetilde{C}\hat{\beta}_2\right) \\
=\;& \delta^t + \frac{\gamma}{2\hat{\theta}}s^t - \frac{\gamma}{2}\mathbb{E}\left[\left\|\nabla f(x^t)\right\|^2\right] - \left(\frac{1}{2\gamma} - \frac{L}{2} - \frac{\gamma}{2\hat{\theta}}\hat{\beta}_1 L_{\mathrm{AM}}^2\right)r^t + \frac{\gamma\widetilde{A}\beta_2}{2\hat{\theta}}\delta^{t+1} + \frac{\gamma\widetilde{C}}{2\hat{\theta}}\beta_2 \\
\leq\;& \delta^t + \frac{\gamma}{2\hat{\theta}}s^t - \frac{\gamma}{2}\mathbb{E}\left[\left\|\nabla f(x^t)\right\|^2\right] + \frac{\gamma\widetilde{A}\beta_2}{2\hat{\theta}}\delta^{t+1} + \frac{\gamma\widetilde{C}}{2\hat{\theta}}\beta_2.
\end{aligned}
$$

The last inequality follows from the bound $\gamma^2\frac{\hat{\beta}_1 L_{\mathrm{AM}}^2}{\hat{\theta}} + L\gamma \leq 1$, which holds due to Lemma 5 for

$$
\gamma \leq \frac{1}{L + L_{\mathrm{AM}}\sqrt{\frac{\hat{\beta}_1}{\hat{\theta}}}}.
$$

Subsequently, we will reconfigure the final inequality and perform algebraic manipulations, taking into account that $\frac{2}{\gamma} > 0$. In the final step of these algebraic transformations, we will leverage the fact that $s^t \geq 0$:

$$
\delta^{t+1} + \frac{\gamma}{2\hat{\theta}}s^{t+1} \;\leq\; \delta^t + \frac{\gamma}{2\hat{\theta}}s^t - \frac{\gamma}{2}\mathbb{E}\left[\left\|\nabla f(x^t)\right\|^2\right] + \frac{\gamma\widetilde{A}\beta_2}{2\hat{\theta}}\delta^{t+1} + \frac{\gamma\widetilde{C}}{2\hat{\theta}}\beta_2.
$$

Therefore,

$$
\frac{2}{\gamma}\delta^{t+1} + \frac{2}{\gamma}\frac{\gamma}{2\hat{\theta}}s^{t+1} \;\leq\; \frac{2}{\gamma}\delta^t + \frac{2}{\gamma}\frac{\gamma}{2\hat{\theta}}s^t - \mathbb{E}\left[\left\|\nabla f(x^t)\right\|^2\right] + \frac{2}{\gamma}\frac{\gamma\widetilde{A}\beta_2}{2\hat{\theta}}\delta^{t+1} + \frac{2}{\gamma}\frac{\gamma\widetilde{C}}{2\hat{\theta}}\beta_2.
$$

Further,

$$
\begin{aligned}
\mathbb{E}\left[\left\|\nabla f(x^t)\right\|^2\right] &\leq -\frac{2}{\gamma}\delta^{t+1} - \frac{2}{\gamma}\frac{\gamma}{2\hat{\theta}}s^{t+1} + \frac{2}{\gamma}\delta^t + \frac{2}{\gamma}\frac{\gamma}{2\hat{\theta}}s^t + \frac{2}{\gamma}\frac{\gamma\widetilde{A}\beta_2}{2\hat{\theta}}\delta^{t+1} + \frac{2}{\gamma}\frac{\gamma\widetilde{C}}{2\hat{\theta}}\beta_2 \\
&\leq -\frac{2}{\gamma}\delta^{t+1} - \frac{2}{\gamma}\frac{\gamma}{2\hat{\theta}}s^{t+1} + \frac{2}{\gamma}\left(\delta^t + \frac{\gamma}{2\hat{\theta}}s^t\right) + \frac{2}{\gamma}\frac{\gamma\widetilde{A}\beta_2}{2\hat{\theta}}\delta^{t+1} + \frac{\widetilde{C}\beta_2}{\hat{\theta}} \\
&\leq \frac{2}{\gamma}\left(\left(\delta^t + \frac{\gamma}{2\hat{\theta}}s^t\right) - 1\left(1 - \frac{\gamma\widetilde{A}\beta_2}{2\hat{\theta}}\right)\delta^{t+1} - \left(\frac{\gamma}{2\hat{\theta}}s^{t+1}\right)\right) + \frac{\widetilde{C}\beta_2}{\hat{\theta}} \\
&\leq \frac{2}{\gamma}\left(\left(\delta^t + \frac{\gamma}{2\hat{\theta}}s^t\right) - \left(1 - \frac{\gamma\widetilde{A}\beta_2}{2\hat{\theta}}\right)\left(\delta^{t+1} + \frac{\gamma}{2\hat{\theta}}s^{t+1}\right)\right) + \frac{\widetilde{C}\beta_2}{\hat{\theta}}.
\end{aligned}
$$

We sum up inequalities above with weights $v_t/V_T$, where $v_t := (1 - \frac{\gamma\widetilde{A}\hat{\beta}_2}{2\theta})^t$ and $V_T := \sum_{i=1}^T v_i$:

$$
\begin{aligned}
\mathbb{E}\left[\left\|\nabla f(\hat{x}^T)\right\|^2\right] &= \sum_{t=0}^T \frac{v_t}{V_T}\mathbb{E}\left[\left\|\nabla f(x^t)\right\|^2\right] \\
&= \frac{1}{V_T}\sum_{t=0}^T v_t\mathbb{E}\left[\left\|\nabla f(x^t)\right\|^2\right] \\
&\leq \frac{1}{V_T}\sum_{t=0}^T v_t\left(\frac{2}{\gamma}\left(\left(\delta^t + \frac{\gamma}{2\hat{\theta}}s^t\right) - \left(1 - \frac{\gamma\widetilde{A}\beta_2}{2\hat{\theta}}\right)\left(\delta^{t+1} + \frac{\gamma}{2\hat{\theta}}s^{t+1}\right)\right) + \frac{\widetilde{C}\beta_2}{\hat{\theta}}\right) \\
&= \frac{2}{\gamma V_T}\sum_{t=0}^T w_t\left(\left(\delta^t + \frac{\gamma}{2\hat{\theta}}s^t\right) - \left(1 - \frac{\gamma\widetilde{A}\beta_2}{2\hat{\theta}}\right)\left(\delta^{t+1} + \frac{\gamma}{2\hat{\theta}}s^{t+1}\right)\right) + \sum_{t=0}^T \frac{w_t}{W_T}\cdot\frac{\widetilde{C}\beta_2}{\hat{\theta}} \\
&= \frac{2}{\gamma V_T}\sum_{t=0}^T w_t\left(\left(\delta^t + \frac{\gamma}{2\hat{\theta}}s^t\right) - \left(1 - \frac{\gamma\widetilde{A}\beta_2}{2\hat{\theta}}\right)\left(\delta^{t+1} + \frac{\gamma}{2\hat{\theta}}s^{t+1}\right)\right) + \frac{\widetilde{C}\beta_2}{\hat{\theta}} \\
&= \frac{2}{\gamma V_T}\sum_{t=0}^T \left(w_t\left(\delta^t + \frac{\gamma}{2\hat{\theta}}s^t\right) - w_{t+1}\left(\delta^{t+1} + \frac{\gamma}{2\hat{\theta}}s^{t+1}\right)\right) + \frac{\widetilde{C}\beta_2}{\hat{\theta}} \\
&\leq \frac{2\delta^0}{\gamma V_T} + \frac{s^0}{\hat{\theta}V_T} + \frac{\widetilde{C}\beta_2}{\hat{\theta}}.
\end{aligned}
$$

Finally, we notice that $V_T = \sum_{t=1}^T (1 - \frac{\gamma\widetilde{A}\hat{\beta}_2}{2\theta})^t \geq T\cdot(1 - \frac{\gamma\widetilde{A}\hat{\beta}_2}{2\theta})^T$, what concludes the proof. $\qquad\square$

# G  EF21-W-PP: Weighted Error Feedback 2021 with Partial Participation

In this section, we present another extension of error feedback. Again, to maintain brevity, we show our results for EF21-W, however, we believe getting an enhanced rate for standard EF21 should be straightforward.

## G.1  Algorithm

Building upon the delineation of EF21-W in Algorithm 2, we turn our attention to its partial participation variant, EF21-W-PP, and seek to highlight the primary distinctions between them. One salient difference is the introduction of a distribution, denoted as $\mathcal{D}$, across the clients. For clarity, consider the power set $\mathcal{P}$ of the set $[n] \coloneqq \{1, 2, \dots, n\}$, representing all possible subsets of $[n]$. Then, the distribution $\mathcal{D}$ serves as a discrete distribution over $\mathcal{P}$.

While EF21-W-PP runs, at the start of each communication round $t$, the master, having computed a descent step as $x^{t+1} = x^t - \gamma g^t$, samples a client subset $S^t$ from the distribution $\mathcal{D}$. Contrasting with Algorithm 2 where the new iteration $x^{t+1}$ is sent to all clients, in this variant, it is sent exclusively to those in $S^t$.

Any client $i \in S^t$ adheres to procedures akin to EF21-W: it compresses the quantity $\frac{1}{nw_i}\nabla f_i(x^t) - g_i^t$ and transmits this to the master. Conversely, client $j$ omitted in $S^t$, i.e., $j \notin S^t$, is excluded from the training for that iteration. Concluding the round, the master updates $g^{t+1}$ by integrating the averaged compressed variances received from clients in the set $S^t$.

---

**Algorithm 5** EF21-W-PP: Weighted Error Feedback 2021 with Partial Participation

---

1: **Input:** initial model parameters $x^0 \in \mathbb{R}^d$; initial gradient estimates $g_1^0, g_2^0, \dots, g_n^0 \in \mathbb{R}^d$ stored at the clients; weights $w_i = L_i / \sum_j L_j$; stepsize $\gamma > 0$; number of iterations $T > 0$; distribution $\mathcal{D}$ over clients
2: **Initialize:** $g^0 = \sum_{i=1}^n w_i g_i^0$ on the server
3: **for** $t = 0, 1, 2, \dots, T-1$ **do**
4:     Server computes $x^{t+1} = x^t - \gamma g^t$
5:     Server samples a subset $S^t \sim \mathcal{D}$ of clients
6:     Server broadcasts $x^{t+1}$ to clients in $S^t$
7:     **for** $i = 1, \dots, n$ **on the clients in parallel do**
8:         **if** $i \in S^t$ **then**
9:             Compute $u_i^t = \mathcal{C}_i^t\left(\frac{1}{nw_i}\nabla f_i(x^{t+1}) - g_i^t\right)$ and update $g_i^{t+1} = g_i^t + u_i^t$
10:            Send the compressed message $u_i^t$ to the server
11:         **end if**
12:         **if** $i \notin S^t$ **then**
13:            Set $u_i^t = 0$ for the client and the server
14:            Do not change local state $g_i^{t+1} = g_i^t$
15:         **end if**
16:     **end for**
17:     Server updates $g_i^{t+1} = g_i^t + u_i^t$ for all $i \in [n]$, and computes $g^{t+1} = \sum_{i=1}^n w_i g_i^{t+1}$
18: **end for**
19: **Output:** Point $\hat{x}^T$ chosen from the set $\{x^0, \dots, x^{T-1}\}$ uniformly at random

---

Assume $S$ is drawn from the distribution $\mathcal{D}$. Let us denote
$$p_i \coloneqq \mathbf{Prob}(i \in S^t). \tag{62}$$

In other words, $p_i$ represents the probability of client $i$ being selected in any iteration. For given parameters $p_i$ such that $p_i \in (0, 1]$ for $i \in [n]$, we introduce the notations $p_{\min} := \min_i p_i$ and $p_{\max} := \max_i p_i$, respectively.

### G.2 A LEMMA

Having established the necessary definitions, we can now proceed to formulate the lemma.

**Lemma 12.** *Let $\mathcal{C}_i^t \in \mathbb{C}(\alpha)$ for all $i \in [n]$ and $t \geq 0$. Let Assumption 2 hold. Define*

$$G_i^t := \left\| g_i^t - \frac{\nabla f_i(x^t)}{nw_i} \right\|^2, \qquad G^t := \sum_{i=1}^n w_i G_i^t. \tag{63}$$

*For any $s > 0$ and $\rho > 0$, let us define the following quantities:*

$$\begin{aligned}
\theta(\alpha, s) &:= 1 - (1 - \alpha)(1 + s) \\
\beta(\alpha, s) &:= \beta(\alpha, s) = (1 - \alpha)(1 + s^{-1}) \\
\theta_p &:= p_{\min}\rho + \theta(\alpha, s)p_{\max} - \rho - (p_{\max} - p_{\min}) \\
\tilde{B} &:= \left(\beta(\alpha, s)p_{\max} + (1 - p_{\min})(1 + \rho^{-1})\right) L_{\text{AM}}^2.
\end{aligned}$$

*Additionally, assume that*

$$\frac{1 + \rho(1 - p_{\min}) + (p_{\max} - p_{\min})}{p_{\max}} \geq \theta(\alpha, s) > \frac{\rho(1 - p_{\min}) + (p_{\max} - p_{\min})}{p_{\max}}.$$

*Then, we have*

$$\mathbb{E}\left[G^{t+1}\right] \leq (1 - \theta_p)\mathbb{E}\left[G^t\right] + \tilde{B}\mathbb{E}\left[\left\|x^{t+1} - x^t\right\|^2\right]. \tag{64}$$

*Proof.* Let us define $W^t := \{g_1^t, \ldots, g_n^t, x^t, x^{t+1}\}$. If client $i$ participates in the training at iteration $t$, then

$$
\begin{aligned}
\mathbb{E}\left[G_i^{t+1} \mid W^t, i \in S^t\right] &\overset{(63)}{=} \mathbb{E}\left[\left\|g_i^{t+1} - \frac{\nabla f_i(x^{t+1})}{nw_i}\right\|^2 \mid W^t, i \in S^t\right] \\
&\overset{\text{line 9 of Algorithm 5}}{=} \mathbb{E}\left[\left\|g_i^t + \mathcal{C}_i^t\left(\frac{\nabla f_i(x^{t+1})}{nw_i} - g_i^t\right) - \frac{\nabla f_i(x^{t+1})}{nw_i}\right\|^2 \mid W^t, i \in S^t\right] \\
&\overset{(4)}{\leq} (1 - \alpha)\left\|\frac{\nabla f_i(x^{t+1})}{nw_i} - g_i^t\right\|^2 \\
&= (1 - \alpha)\left\|\frac{\nabla f_i(x^{t+1})}{nw_i} - \frac{\nabla f_i(x^t)}{nw_i} + \frac{\nabla f_i(x^t)}{nw_i} - g_i^t\right\|^2 \\
&\overset{(21)}{\leq} (1 - \alpha)(1 + s)\left\|\frac{\nabla f_i(x^t)}{nw_i} - g_i^t\right\|^2 \\
&\qquad + (1 - \alpha)\left(1 + s^{-1}\right)\frac{1}{n^2 w_i^2}\left\|\nabla f_i(x^{t+1}) - \nabla f_i(x^t)\right\|^2 \\
&\overset{(7)}{\leq} (1 - \alpha)(1 + s)\left\|\frac{\nabla f_i(x^t)}{nw_i} - g_i^t\right\|^2 \\
&\qquad + (1 - \alpha)\left(1 + s^{-1}\right)\frac{L_i^2}{n^2 w_i^2}\left\|x^{t+1} - x^t\right\|^2.
\end{aligned}
$$

Utilizing the tower property and taking the expectation with respect to $W^t$, we derive:

$$\mathbb{E}\left[G_i^{t+1} \mid i \in S^t\right] \le (1 - \theta(\alpha, s))\mathbb{E}\left[G_i^t\right] + \beta(\alpha, s)\frac{L_i^2}{n^2 w_i^2}\mathbb{E}\left[\left\|x^{t+1} - x^t\right\|^2\right], \tag{65}$$

where $\theta(\alpha, s) = 1 - (1 - \alpha)(1 + s)$, and $\beta(\alpha, s) = (1 - \alpha)(1 + s^{-1})$. We now aim to bound the quantity $\mathbb{E}\left[G_i^{t+1} \mid i \notin S^t\right]$, starting with an application of the tower property:

$$
\begin{aligned}
\mathbb{E}\left[G_i^{t+1} \mid i \notin S^t\right] &= \mathbb{E}\left[\mathbb{E}\left[G_i^{t+1} \mid W^t, i \notin S^t\right]\right] \\
&\overset{(63)}{=} \mathbb{E}\left[\mathbb{E}\left[\left\|g_i^{t+1} - \frac{\nabla f_i(x^{t+1})}{nw_i}\right\|^2 \mid W^t, i \notin S^t\right]\right] \\
&= \mathbb{E}\left[\mathbb{E}\left[\left\|g_i^t - \frac{\nabla f_i(x^{t+1})}{nw_i} + \frac{\nabla f_i(x^t)}{nw_i} - \frac{\nabla f_i(x^t)}{nw_i}\right\|^2 \mid W^t, i \notin S^t\right]\right] \\
&\overset{(21)}{\le} \mathbb{E}\left[\mathbb{E}\left[(1 + \rho)\left\|g_i^t - \frac{\nabla f_i(x^t)}{nw_i}\right\|^2 + (1 + \rho^{-1})\left\|\frac{\nabla f_i(x^t)}{nw_i} - \frac{\nabla f_i(x^{t+1})}{nw_i}\right\|^2 \mid W^t, i \notin S^t\right]\right] \\
&= (1 + \rho)\mathbb{E}\left[G_i^t\right] + \frac{(1 + \rho^{-1})}{n^2 w_i^2}\mathbb{E}\left[\left\|\nabla f_i(x^{t+1}) - \nabla f_i(x^t)\right\|^2\right].
\end{aligned}
$$

Given that Assumption 2 is satisfied, by applying (7) to the second term, we obtain:

$$\mathbb{E}\left[G_i^{t+1} \mid i \notin S^t\right] \le (1 + \rho)\mathbb{E}\left[G_i^t\right] + \frac{L_i^2(1 + \rho^{-1})}{n^2 w_i^2}\mathbb{E}\left[\left\|x^{t+1} - x^t\right\|^2\right]. \tag{66}$$

We combine the two preceding bounds:

$$
\begin{aligned}
\mathbb{E}\left[G_i^{t+1}\right] &= \mathbf{Prob}(i \in S^t)\mathbb{E}\left[G_i^{t+1} \mid i \in S^t\right] + \mathbf{Prob}(i \notin S^t)\mathbb{E}\left[G_i^{t+1} \mid i \notin S^t\right] \\
&\overset{(62)}{=} p_i\mathbb{E}\left[G_i^{t+1} \mid i \in S^t\right] + (1 - p_i)\mathbb{E}\left[G_i^{t+1} \mid i \notin S^t\right] \\
&\overset{(65)+(66)}{\le} p_i\left[(1 - \theta(\alpha, s))\mathbb{E}\left[G_i^t\right] + \beta(\alpha, s)\frac{L_i^2}{n^2 w_i^2}\mathbb{E}\left[\left\|x^{t+1} - x^t\right\|^2\right]\right] \\
&\quad + (1 - p_i)\left[(1 + \rho)\mathbb{E}\left[G_i^t\right] + \frac{L_i^2(1 + \rho^{-1})}{n^2 w_i^2}\mathbb{E}\left[\left\|x^{t+1} - x^t\right\|^2\right]\right] \\
&= \left((1 - \theta(\alpha, s))p_i + (1 - p_i)(1 + \rho)\right)\mathbb{E}\left[G_i^t\right] \\
&\quad + \left(\beta(\alpha, s)p_i + (1 - p_i)(1 + \rho^{-1})\right)\frac{L_i^2}{n^2 w_i^2}\mathbb{E}\left[\left\|x^{t+1} - x^t\right\|^2\right].
\end{aligned}
$$

Consequently, for $\mathbb{E}\left[G^{t+1}\right]$, we derive the subsequent bound:

$$
\begin{aligned}
\mathbb{E}\left[G^{t+1}\right] &\overset{(63)}{=} \mathbb{E}\left[\sum_{i=1}^n w_i G_i^{t+1}\right] \\
&= \sum_{i=1}^n w_i \mathbb{E}\left[G_i^{t+1}\right] \\
&\leq \sum_{i=1}^n w_i \left((1-\theta(\alpha,s))p_i + (1-p_i)(1+\rho)\right)\mathbb{E}\left[G_i^t\right] \\
&\quad + \sum_{i=1}^n w_i \left(\beta(\alpha,s)p_i + (1-p_i)(1+\rho^{-1})\right)\frac{L_i^2}{n^2 w_i^2}\mathbb{E}\left[\left\|x^{t+1}-x^t\right\|^2\right],
\end{aligned}
$$

where we applied the preceding inequality. Remembering the definitions $p_{\min} := \min_i p_i$ and $p_{\max} := \max_i p_i$, we subsequently obtain:

$$
\begin{aligned}
\mathbb{E}\left[G^{t+1}\right] &\leq \sum_{i=1}^n w_i \left((1-\theta(\alpha,s))p_{\max} + (1-p_{\min})(1+\rho)\right)\mathbb{E}\left[G_i^t\right] \\
&\quad + \sum_{i=1}^n \left(\beta(\alpha,s)p_{\max} + (1-p_{\min})(1+\rho^{-1})\right)\frac{L_i^2}{n^2 w_i}\mathbb{E}\left[\left\|x^{t+1}-x^t\right\|^2\right] \\
&= \left((1-\theta(\alpha,s))p_{\max} + (1-p_{\min})(1+\rho)\right)\sum_{i=1}^n w_i \mathbb{E}\left[G_i^t\right] \\
&\quad + \left(\beta(\alpha,s)p_{\max} + (1-p_{\min})(1+\rho^{-1})\right)\sum_{i=1}^n \frac{L_i^2}{n^2 w_i}\mathbb{E}\left[\left\|x^{t+1}-x^t\right\|^2\right].
\end{aligned}
$$

Applying (63) and (25), we obtain:

$$
\begin{aligned}
\mathbb{E}\left[G^{t+1}\right] &= \left((1-\theta(\alpha,s))p_{\max} + (1-p_{\min})(1+\rho)\right)\mathbb{E}\left[G^t\right] \\
&\quad + \left(\beta(\alpha,s)p_{\max} + (1-p_{\min})(1+\rho^{-1})\right)\sum_{i=1}^n \frac{L_i^2}{n^2 \frac{L_i}{\sum_{j=1}^n L_j}}\mathbb{E}\left[\left\|x^{t+1}-x^t\right\|^2\right] \\
&= \left((1-\theta(\alpha,s))p_{\max} + (1-p_{\min})(1+\rho)\right)\mathbb{E}\left[G^t\right] \\
&\quad + \left(\beta(\alpha,s)p_{\max} + (1-p_{\min})(1+\rho^{-1})\right)\sum_{j=1}^n \frac{L_j}{n}\sum_{i=1}^n \frac{L_i}{n}\mathbb{E}\left[\left\|x^{t+1}-x^t\right\|^2\right] \\
&= \left((1-\theta(\alpha,s))p_{\max} + (1-p_{\min})(1+\rho)\right)\mathbb{E}\left[G^t\right] \\
&\quad + \left(\beta(\alpha,s)p_{\max} + (1-p_{\min})(1+\rho^{-1})\right)L_{\mathrm{AM}}^2\mathbb{E}\left[\left\|x^{t+1}-x^t\right\|^2\right].
\end{aligned}
$$

Subsequently, in order to simplify the last inequality, we introduce the variables $1-\theta_p$ and $\tilde{B}$:

$$
\begin{aligned}
1-\theta_p &:= (1-\theta(\alpha,s))p_{\max} + (1-p_{\min})(1+\rho) \\
&= p_{\max} - p_{\max}\theta(\alpha,s) + 1 - p_{\min} + \rho - p_{\min}\rho \\
&= 1 - (-p_{\max} + p_{\max}\theta(\alpha,s) + p_{\min} - \rho + p_{\min}\rho) \\
&= 1 - (p_{\min}\rho + p_{\max}\theta(\alpha,s) - \rho - (p_{\max} - p_{\min})).
\end{aligned}
$$

Therefore,

$$
\begin{aligned}
\theta_p &= (p_{\max}\theta(\alpha, s) - \rho(1 - p_{\min}) - (p_{\max} - p_{\min})) \\
\tilde{B} &:= \left(\beta(\alpha, s)p_{\max} + (1 - p_{\min})(1 + \rho^{-1})\right) L_{\text{AM}}^2.
\end{aligned}
$$

Expressed in terms of these variables, the final inequality can be reformulated as:

$$
\mathbb{E}\left[G^{t+1}\right] \leq (1 - \theta_p)\mathbb{E}\left[G^t\right] + \tilde{B}\mathbb{E}\left[\left\|x^{t+1} - x^t\right\|^2\right].
$$

Since we need the contraction property over the gradient distortion $\mathbb{E}\left[G^{t+1}\right]$, we require $0 < \theta_p \leq 1$. We rewrite these conditions as follows:

$$
\frac{1 + \rho(1 - p_{\min}) + (p_{\max} - p_{\min})}{p_{\max}} \geq \theta(\alpha, s) > \frac{\rho(1 - p_{\min}) + (p_{\max} - p_{\min})}{p_{\max}}.
$$

$\square$

## G.3   MAIN RESULT

We are ready to prove the main convergence theorem.

**Theorem 10.** *Consider Algorithm 5 (EF21-W-PP) applied to the distributed optimization problem* (1). *Let Assumptions 1, 2, 3 hold, assume that $\mathcal{C}_i^t \in \mathbb{C}(\alpha)$ for all $i \in [n]$ and $t > 0$, set*

$$
G^t := \sum_{i=1}^n w_i \left\| g_i^t - \frac{1}{nw_i}\nabla f_i(x^t)\right\|^2,
$$

*where $w_i = \frac{L_i}{\sum_j L_j}$ for all $i \in [n]$, and let the stepsize satisfy*

$$
0 < \gamma \leq \left(L + \sqrt{\frac{\tilde{B}}{\theta_p}}\right)^{-1}, \tag{67}
$$

*where $s > 0$, $\rho > 0$, and*

$$
\begin{aligned}
\theta(\alpha, s) &:= 1 - (1 - \alpha)(1 + s) \\
\beta(\alpha, s) &:= (1 - \alpha)(1 + s^{-1}) \\
\theta_p &:= p_{\min}\rho + \theta(\alpha, s)p_{\max} - \rho - (p_{\max} - p_{\min}) \\
\tilde{B} &:= \left(\beta(\alpha, s)p_{\max} + (1 - p_{\min})(1 + \rho^{-1})\right) L_{\text{AM}}^2.
\end{aligned}
$$

*Additionally, assume that*

$$
\frac{1 + \rho(1 - p_{\min}) + (p_{\max} - p_{\min})}{p_{\max}} \geq \theta(\alpha, s) > \frac{\rho(1 - p_{\min}) + (p_{\max} - p_{\min})}{p_{\max}}.
$$

*If for $T > 1$ we define $\hat{x}^T$ as an element of the set $\{x^0, x^1, \ldots, x^{T-1}\}$ chosen uniformly at random, then*

$$
\mathbb{E}\left[\|\nabla f(\hat{x}^T)\|^2\right] \leq \frac{2(f(x^0) - f^*)}{\gamma T} + \frac{G^0}{\theta_p T}. \tag{68}
$$

*Proof.* Following the same approach employed in the proof for the SGD case, we obtain

$$
\begin{aligned}
\mathbb{E}\left[f(x^{t+1}) - f^*\right] &\leq \mathbb{E}\left[f(x^t) - f^*\right] - \frac{\gamma}{2}\mathbb{E}\left[\left\|\nabla f(x^t)\right\|^2\right] \\
&\quad - \left(\frac{1}{2\gamma} - \frac{L}{2}\right)\mathbb{E}\left[\left\|x^{t+1} - x^t\right\|^2\right] + \frac{\gamma}{2}\mathbb{E}\left[G^t\right].
\end{aligned}
$$

Let $\delta^t := \mathbb{E}\left[f(x^t) - f^*\right]$, $s^t := \mathbb{E}\left[G^t\right]$ and $r^t := \mathbb{E}\left[\left\|x^{t+1} - x^t\right\|^2\right]$. Applying the previous lemma, we obtain:

$$
\begin{aligned}
\delta^{t+1} + \frac{\gamma}{2\theta_p}s^{t+1} &\leq \delta^t - \frac{\gamma}{2}\mathbb{E}\left[\left\|\nabla f(x^t)\right\|^2\right] - \left(\frac{1}{2\gamma} - \frac{L}{2}\right)r^t + \frac{\gamma}{2}s^t + \frac{\gamma}{2\theta_p}\left(\tilde{B}r^t + (1-\theta_p)s^t\right) \\
&= \delta^t + \frac{\gamma}{2\theta}s^t - \frac{\gamma}{2}\mathbb{E}\left[\left\|\nabla f(x^t)\right\|^2\right] - \underbrace{\left(\frac{1}{2\gamma} - \frac{L}{2} - \frac{\gamma}{2\theta_p}\tilde{B}\right)}_{\geq 0}r^t \\
&\leq \delta^t + \frac{\gamma}{2\theta_p}s^t - \frac{\gamma}{2}\mathbb{E}\left[\left\|\nabla f(x^t)\right\|^2\right].
\end{aligned}
$$

By summing up inequalities for $t = 0, \ldots, T-1$, we get

$$
0 \leq \delta^T + \frac{\gamma}{2\theta_p}s^T \leq \delta^0 + \frac{\gamma}{2\theta_p}s^0 - \frac{\gamma}{2}\sum_{t=0}^{T-1}\mathbb{E}\left[\left\|\nabla f(x^t)\right\|^2\right].
$$

Finally, via multiplying both sides by $\frac{2}{\gamma T}$, after rearranging we get:

$$
\sum_{t=0}^{T-1}\frac{1}{T}\mathbb{E}\left[\left\|\nabla f(x^t)\right\|^2\right] \leq \frac{2\delta^0}{\gamma T} + \frac{s^0}{\theta_p T}.
$$

It remains to notice that the left-hand side can be interpreted as $\mathbb{E}\left[\left\|\nabla f(\hat{x}^T)\right\|^2\right]$, where $\hat{x}^T$ is chosen from the set $\{x^0, x^1, \ldots, x^{T-1}\}$ uniformly at random. $\qquad\square$

Our analysis of this extension follows a similar approach as the one used by Fatkhullin et al. (2021), for algorithm they called EF21-PP. Presented analysis of EF21-W-PP has a better provides a better multiplicative factor in definition of $\widetilde{B} \propto L_{\mathrm{AM}}^2$, where in vanilla EF21-PP had $\widetilde{B} \propto L_{\mathrm{QM}}^2$. This fact improves upper bound on allowable step size in (67).

# H  IMPROVED THEORY FOR EF21 IN THE RARE FEATURES REGIME

In this section, we adapt our new results to the *rare features* regime proposed and studied by Richtárik et al. (2023).

## H.1  ALGORITHM

In this section, we focus on Algorithm 6, which is an adaptation of EF21 (as delineated in Algorithm 1) that specifically employs $\mathrm{Top}K$ operators. This variant is tailored for the rare features scenario, enhancing the convergence rate by shifting from the average of squared Lipschitz constants to the square of their average. The modifications introduced in Algorithm 6, compared to the standard EF21, are twofold and significant.

Primarily, the algorithm exclusively engages $\mathrm{Top}K$ compressors, leveraging the inherent sparsity present in the data. Additionally, the initial gradient estimates $g_i^0$ are confined to the respective subspaces $\mathbb{R}_i^d$, as characterized by equation (71). With the exception of these distinct aspects, the algorithm's execution parallels that of the original EF21.

---

**Algorithm 6** EF21: Error Feedback 2021 with $\mathrm{Top}K$ compressors

1: **Input:** initial model $x^0 \in \mathbb{R}^d$; initial gradient estimates $g_1^0 \in \mathbb{R}_1^d, \ldots, g_n^0 \in \mathbb{R}_n^d$ (as defined in equation (71)) stored at the server and the clients; stepsize $\gamma > 0$; sparsification levels $K_1, \ldots, K_n \in [d]$; number of iterations $T > 0$
2: **Initialize:** $g^0 = \frac{1}{n} \sum_{i=1}^n g_i^0$ on the server
3: **for** $t = 0, 1, 2, \ldots, T-1$ **do**
4:     Server computes $x^{t+1} = x^t - \gamma g^t$ and broadcasts $x^{t+1}$ to all $n$ clients
5:     **for** $i = 1, \ldots, n$ **on the clients in parallel do**
6:         Compute $u_i^t = \mathrm{Top}K_i(\nabla f_i(x^{t+1}) - g_i^t)$ and update $g_i^{t+1} = g_i^t + u_i^t$
7:         Send the compressed message $u_i^t$ to the server
8:     **end for**
9:     Server updates $g_i^{t+1} = g_i^t + u_i^t$ for all $i \in [n]$, and computes $g^{t+1} = \frac{1}{n} \sum_{i=1}^n g_i^{t+1}$
10: **end for**
11: **Output:** Point $\hat{x}^T$ chosen from the set $\{x^0, \ldots, x^{T-1}\}$ uniformly at random

---

## H.2  NEW SPARSITY MEASURE

To extend our results to the *rare features* regime, we need to slightly change the definition of the parameter $c$ in the original paper. The way we do it is unrolled as follows. First, we recall the following definitions from (Richtárik et al., 2023):

$$\mathcal{Z} := \{(i, j) \in [n] \times [d] \mid [\nabla f_i(x)]_j = 0 \ \forall x \in \mathbb{R}^d\}, \tag{69}$$

and

$$\mathcal{I}_j := \{i \in [n] \mid (i, j) \notin \mathcal{Z}\}, \qquad \mathcal{J}_i := \{j \in [d] \mid (i, j) \notin \mathcal{Z}\}. \tag{70}$$

We also need the following definition of $\mathbb{R}_i^d$:

$$\mathbb{R}_i^d := \{u = (u_1, \ldots, u_d) \in \mathbb{R}^d : u_j = 0 \text{ whenever } (i, j) \in \mathcal{Z}\}. \tag{71}$$

Now we are ready for a new definition of the sparsity parameter $c$:

$$c := n \cdot \max_{j \in [d]} \sum_{i \in \mathcal{I}_j} w_i, \tag{72}$$

where $w_i$ is defined as in (25). We note that $c$ recovers the standard definition from (Richtárik et al., 2023) when $w_i = \frac{1}{n}$ for all $i \in [n]$.

## H.3 LEMMAS

We will proceed through several lemmas.

**Lemma 13.** *Let $u_i \in \mathbb{R}_i^d$ for all $i \in [n]$. Then, the following inequality holds:*

$$\left\| \sum_{i=1}^n w_i u_i \right\|^2 \le \frac{c}{n} \sum_{i=1}^n w_i \| u_i \|^2. \tag{73}$$

*Proof.* Initially, we observe that

$$\left\| \sum_{i=1}^n w_i u_i \right\|^2 = \sum_{j=1}^d \left( \sum_{i=1}^n w_i u_{ij} \right)^2. \tag{74}$$

We note that for any $j \in [d]$ it holds that

$$\left( \sum_{i=1}^n w_i u_{ij} \right)^2 = \left( \sum_{i \in \mathcal{I}_j} w_i u_{ij} \right)^2$$

$$= \left( \sum_{i \in \mathcal{I}_j} w_i \right)^2 \left( \sum_{i \in \mathcal{I}_j} \frac{w_i}{\sum_{i' \in \mathcal{I}_j} w_{i'}} u_{ij} \right)^2 \le \left( \sum_{i \in \mathcal{I}_j} w_i \right)^2 \sum_{i \in \mathcal{I}_j} \frac{w_i}{\sum_{i' \in \mathcal{I}_j} w_{i'}} u_{ij}^2, \tag{75}$$

where on the last line we used the Jensen's inequality. Subsequent arithmetic manipulations and the incorporation of definition (72) yield:

$$\left( \sum_{i=1}^n w_i u_{ij} \right)^2 \overset{(75)}{\le} \left( \sum_{i \in \mathcal{I}_j} w_i \right) \cdot \sum_{i \in \mathcal{I}_j} w_i u_{ij}^2$$

$$= \left( \sum_{i \in \mathcal{I}_j} w_i \right) \cdot \sum_{i=1}^n w_i u_{ij}^2$$

$$\le \left[ \max_{j \in [d]} \left( \sum_{i \in \mathcal{I}_j} w_i \right) \right] \cdot \sum_{i=1}^n w_i u_{ij}^2$$

$$\overset{(72)}{=} \frac{c}{n} \sum_{i=1}^n w_i u_{ij}^2. \tag{76}$$

Substituting Equation (76) into Equation (74) completes the proof. $\square$

**Lemma 14.** *Assume that $g_i^0 \in \mathbb{R}_i^d$ for all $i \in [n]$. Then, it holds for all $t > 0$ that*

$$\| g^t - \nabla f(x^t) \|^2 \le \frac{c}{n} G^t. \tag{77}$$

*Proof.* By Lemma 8 in Richtárik et al. (2023), for EF21 the update $g_i^t$ stays in $\mathbb{R}_i^d$ if $g_i^0 \in \mathbb{R}_i^d$. We then proceed as follows:

$$\|g^t - \nabla f(x^t)\|^2 \overset{(9)}{=} \left\| \frac{1}{n} \sum_{i=1}^n g_i^t - \nabla f_i(x^t) \right\|^2 = \left\| \sum_{i=1}^n w_i \left[ \frac{1}{nw_i}(g_i^t - \nabla f_i(x^t)) \right] \right\|^2. \tag{78}$$

Since $g_i^t \in \mathbb{R}_i^d$, as was noted, and $\nabla f_i(x^t) \in \mathbb{R}_i^d$, by the definition of $\mathbb{R}_i^d$, then $\frac{1}{nw_i}(g_i^t - \nabla f_i(x^t))$ also belongs to $\mathbb{R}_i^d$. By Lemma 13, we further proceed:

$$
\begin{aligned}
\|g^t - \nabla f(x^t)\|^2 &\overset{(78)}{=} \left\| \sum_{i=1}^n w_i \left[ \frac{1}{nw_i}(g_i^t - \nabla f_i(x^t)) \right] \right\|^2 \\
&\overset{(73)}{\leq} \frac{c}{n} \sum_{i=1}^n w_i \left\| \frac{1}{nw_i}(g_i^t - \nabla f_i(x^t)) \right\|^2 \\
&= \frac{c}{n} \sum_{i=1}^n \frac{1}{n^2 w_i} \|g_i^t - \nabla f_i(x^t)\|^2 = \frac{c}{n} G^t,
\end{aligned}
$$

which completes the proof. $\qquad\qquad\square$

For the convenience of the reader, we briefly revisit Lemma 6 from (Richtárik et al., 2023).

**Lemma 15** (Lemma 6 from Richtárik et al. (2023))**.** *If Assumption 2 holds, then for $i \in [n]$, we have*

$$\sum_{j:(i,j)\notin\mathcal{Z}} ((\nabla f_i(x))_j - (\nabla f_i(y))_j)^2 \leq L_i^2 \sum_{j:(i,j)\notin\mathcal{Z}} (x_j - y_j)^2 \quad \forall x, y \in \mathbb{R}^d. \tag{79}$$

Now, we proceed to the following lemma, which aims to provide a tighter bound for the quantity $\sum_{i=1}^n \frac{1}{L_i} \|\nabla f_i(x) - \nabla f_i(y)\|^2$.

**Lemma 16.** *If Assumption 2 holds, then*

$$\sum_{i=1}^n \frac{1}{L_i} \|\nabla f_i(x) - \nabla f_i(y)\|^2 \leq c L_{\text{AM}} \|x - y\|^2. \tag{80}$$

*Proof.* The proof commences as follows:

$$
\begin{aligned}
\sum_{i=1}^n \frac{1}{L_i} \|\nabla f_i(x) - \nabla f_i(y)\|^2 &= \sum_{i=1}^n \frac{1}{L_i} \sum_{j:(i,j)\notin\mathcal{Z}} ((\nabla f_i(x))_j - (\nabla f_i(y))_j)^2 \\
&\overset{(79)}{\leq} \sum_{i=1}^n \frac{1}{L_i} L_i^2 \sum_{j:(i,j)\notin\mathcal{Z}} (x_j - y_j)^2 \\
&= \sum_{i=1}^n \sum_{j:(i,j)\notin\mathcal{Z}} L_i (x_j - y_j)^2 \\
&= \sum_{j=1}^d \sum_{i:(i,j)\notin\mathcal{Z}} L_i (x_j - y_j)^2 = \sum_{j=1}^d \left[ (x_j - y_j)^2 \sum_{i:(i,j)\notin\mathcal{Z}} L_i \right] \tag{81}
\end{aligned}
$$

To advance our derivations, we consider the maximum value over $\sum\limits_{i:(i,j)\notin\mathcal{Z}} L_i$:

$$
\begin{aligned}
\sum_{i=1}^{n}\frac{1}{L_i}\|\nabla f_i(x)-\nabla f_i(y)\|^2 \quad &\overset{(81)}{\leq}\quad \sum_{j=1}^{d}\left[(x_j-y_j)^2\sum_{i:(i,j)\notin\mathcal{Z}} L_i\right]\\
&\leq\quad \sum_{j=1}^{d}\left[(x_j-y_j)^2\max_{j\in[d]}\sum_{i:(i,j)\notin\mathcal{Z}} L_i\right]\\
&=\quad \left[\max_{j\in[d]}\sum_{i:(i,j)\notin\mathcal{Z}} L_i\right]\sum_{j=1}^{d}(x_j-y_j)^2\\
&=\quad \left[\max_{j\in[d]}\sum_{i:(i,j)\notin\mathcal{Z}} L_i\right]\|x-y\|^2\\
&=\quad \left[\max_{j\in[d]}\sum_{i\in\mathcal{I}_j} L_i\right]\|x-y\|^2\quad\overset{(72)}{=}\quad cL_{\mathrm{AM}}\|x-y\|^2,
\end{aligned}
$$

what completes the proof. $\qquad\square$

For clarity and easy reference, we recapitulate Lemma 10 from Richtárik et al. (2023).

**Lemma 17** (Lemma 10 from Richtárik et al. (2023))**.** *The iterates of Algorithm 6 method satisfy*

$$
\left\|g_i^{t+1}-\nabla f_i(x^{t+1})\right\|^2 \leq (1-\theta(\alpha))\left\|g_i^t-\nabla f_i(x^t)\right\|^2 + \beta(\alpha)\|\nabla f_i(x^{t+1})-\nabla f_i(x^t)\|^2, \tag{82}
$$

*where* $\alpha=\min\left\{\min\limits_{i\in[n]}\frac{K_i}{|\mathcal{J}_i|},1\right\}$.

**Lemma 18.** *Under Assumption 2, iterates of Algorithm 6 satisfies*

$$
G^{t+1}\leq(1-\theta(\alpha))G^t+\beta(\alpha)\frac{c}{n}L_{\mathrm{AM}}^2\|x-y\|^2. \tag{83}
$$

*Proof.* The proof is a combination of Lemmas 16 and 17:

$$
\begin{aligned}
G^{t+1} \quad &\overset{(36)}{=} \quad \frac{1}{n^2} \sum_{i=1}^{n} \frac{1}{w_i} \left\| g_i^{t+1} - \nabla f_i(x^{t+1}) \right\|^2 \\
&\overset{(82)}{\leq} \quad \frac{1}{n^2} \sum_{i=1}^{n} \frac{1}{w_i} \left[ (1 - \theta(\alpha)) \| g_i^t - \nabla f_i(x^t) \|^2 + \beta(\alpha) \| \nabla f_i(x^{t+1}) - \nabla f_i(x^t) \|^2 \right] \\
&= \quad (1 - \theta(\alpha)) G^t + \frac{\beta(\alpha)}{n^2} \sum_{i=1}^{n} \frac{1}{w_i} \| \nabla f_i(x^{t+1}) - \nabla f_i(x^t) \|^2 \\
&\overset{(25)}{=} \quad (1 - \theta(\alpha)) G^t + \frac{\beta(\alpha)}{n^2} \left( \sum_{j=1}^{n} L_j \right) \sum_{i=1}^{n} \frac{1}{L_i} \| \nabla f_i(x^{t+1}) - \nabla f_i(x^t) \|^2 \\
&\overset{(80)}{\leq} \quad (1 - \theta(\alpha)) G^t + \frac{\beta(\alpha)}{n^2} \frac{c}{n} \left( \sum_{j=1}^{n} L_j \right)^2 \| x - y \|^2 \\
&= \quad (1 - \theta(\alpha)) G^t + \beta(\alpha) \frac{c}{n} L_{\mathrm{AM}}^2 \| x - y \|^2 .
\end{aligned}
$$

$\square$

## H.4 Main result

And now we are ready to formulate the main result.

**Theorem 11.** *Let Assumptions 1, 2 and 3 hold. Let $g_i^0 \in \mathbb{R}_i^d$ for all $i \in [n]$,*

$$
\alpha = \min \left\{ \min_{i \in [n]} \frac{K_i}{|\mathcal{J}_i|}, 1 \right\}, \qquad 0 < \gamma \leq \frac{1}{L + \frac{c}{n} L_{\mathrm{AM}} \xi(\alpha)} .
$$

*Under these conditions, the iterates of Algorithm 6 satisfy*

$$
\frac{1}{T} \sum_{t=0}^{T-1} \| \nabla f(x^t) \|^2 \leq \frac{2(f(x^0) - f^*)}{\gamma T} + \frac{c}{n} \frac{G^0}{\theta(\alpha) T} . \tag{84}
$$

*Proof.* Let us define the Lyapunov function:

$$
\Psi^t := f(x^t) - f^* + \frac{\gamma c}{2\theta n} G^t . \tag{85}
$$

We start the proof as follows:

$$
\begin{aligned}
\Psi^{t+1} \;\overset{(85)}{=}\;\; & f(x^{t+1}) - f^* + \frac{\gamma c}{2\theta n} G^{t+1} \\[4pt]
\overset{(20)}{\leq}\;\; & f(x^t) - f^* - \frac{\gamma}{2}\|\nabla f(x^t)\|^2 - \left(\frac{1}{2\gamma} - \frac{L}{2}\right)\|x^{t+1} - x^t\|^2 + \frac{\gamma}{2}\|g^t - \nabla f(x^t)\|^2 + \frac{\gamma c}{2\theta n} G^{t+1} \\[4pt]
\overset{(77)}{\leq}\;\; & f(x^t) - f^* - \frac{\gamma}{2}\|\nabla f(x^t)\|^2 - \left(\frac{1}{2\gamma} - \frac{L}{2}\right)\|x^{t+1} - x^t\|^2 + \frac{\gamma}{2}\frac{c}{n}G^t + \frac{\gamma c}{2\theta n} G^{t+1} \\[4pt]
\overset{(82)}{\leq}\;\; & f(x^t) - f^* - \frac{\gamma}{2}\|\nabla f(x^t)\|^2 - \left(\frac{1}{2\gamma} - \frac{L}{2}\right)\|x^{t+1} - x^t\|^2 \\[4pt]
& + \frac{\gamma}{2}\frac{c}{n}G^t + \frac{\gamma c}{2\theta n}\left((1-\theta)G^t + \beta\frac{c}{n}L_{\mathrm{AM}}^2\|x^{t+1} - x^t\|^2\right) \\[4pt]
=\;\; & f(x^t) - f^* + \frac{\gamma c}{2\theta n}G^t - \frac{\gamma}{2}\|\nabla f(x^t)\|^2 - \left(\frac{1}{2\gamma} - \frac{L}{2} - \frac{\gamma}{2}\frac{\beta}{\theta}\frac{c^2}{n^2}\cdot L_{\mathrm{AM}}^2\right)\|x^{t+1} - x^t\|^2 \\[4pt]
=\;\; & \Psi^t - \frac{\gamma}{2}\|\nabla f(x^t)\|^2 - \underbrace{\left(\frac{1}{2\gamma} - \frac{L}{2} - \frac{\gamma}{2}\frac{\beta}{\theta}\frac{c^2}{n^2}\cdot L_{\mathrm{AM}}^2\right)}_{\geq 0}\|x^{t+1} - x^t\|^2 \\[4pt]
\leq\;\; & \Psi^t - \frac{\gamma}{2}\|\nabla f(x^t)\|^2.
\end{aligned}
$$

Unrolling the inequality above, we get

$$
0 \leq \Psi^T \leq \Psi^{T-1} - \frac{\gamma}{2}\|\nabla f(x^{T-1})\|^2 \leq \Psi^0 - \frac{\gamma}{2}\sum_{t=0}^{T-1}\|\nabla f(x^t)\|^2,
$$

from what the main result follows. □

# I    EXPERIMENTS: FURTHER DETAILS

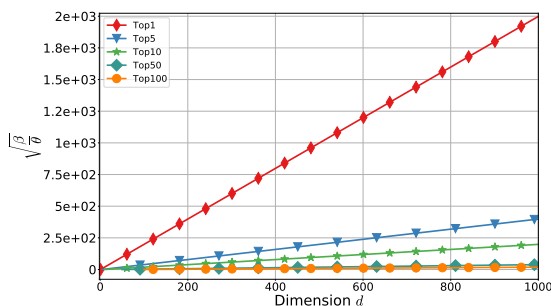

Figure 4: The factor $\xi = \sqrt{\beta/\theta}$ as a function of optimization variable dimension $d$ for several TopK compressors. The behavior is independent of properties of $\{f_1(x), \ldots, f_n(x)\}$ and $f(x)$.

## I.1    COMPUTING AND SOFTWARE ENVIRONMENT

We used the Python software suite `FL_PyTorch` (Burlachenko et al., 2021) to simulate the distributed environment for training. We carried out experiments on a compute node with `Ubuntu 18.04 LTS`, 256 GBytes of DRAM DDR4 memory at 2.9GHz, and 48 cores (2 sockets with 24 cores per socket) of Intel(R) Xeon(R) Gold 6246 CPU at 3.3GHz. We used double-precision arithmetic during computing gradient oracles. All our computations were carried on `CPU`.

## I.2    COMMENTS ON THE IMPROVEMENT

The standard EF21 analysis (Richtárik et al., 2021) allows to utilize EF21 with maximum allowable step size $\gamma$ equal to:

$$\gamma = \left(L + L_{\text{QM}}\sqrt{\frac{\beta(\alpha)}{\theta(\alpha)}}\right)^{-1}, \qquad \theta(\alpha) = 1 - \sqrt{1-\alpha}, \qquad \beta(\alpha) = \frac{1-\alpha}{1-\sqrt{1-\alpha}}.$$

Our analysis allows us to replace the quantity $L_{\text{QM}}$ with $L_{\text{AM}}$. This improvement has an important consequence. The replaced quantity affects the step size by a factor of $\xi(\alpha) = \sqrt{\frac{\beta(\alpha)}{\theta(\alpha)}}$. This factor can be arbitrarily large as $d$ increases, as shown in Figure 4. If $d$ is increasing and the parameter $k$ of TopK compressor is fixed, then even a small improvement in the constant term can have a significant impact in an absolute sense to the computed step size if $\xi(\alpha) \gg L$.

## I.3    WHEN IMPROVED ANALYSIS LEADS TO MORE AGGRESSIVE STEPS

The quantity $L_{\text{QM}} \coloneqq \sqrt{\frac{1}{n}\sum_{i=1}^{n}L_i^2}$ plays essential role in EF21 analysis. As we saw with special consideration this quantity for EF21 and its extensions is improvable. The improved analysis allows us to replace it with $L_{\text{AM}} \coloneqq \frac{1}{n}\sum_{i=1}^{n}L_i$. Clearly, by the arithmetic-quadratic mean inequality,

$$L_{\text{var}} \coloneqq L_{\text{QM}}^2 - L_{\text{AM}}^2 \geq 0.$$

The difference $L_{\mathrm{QM}} - L_{\mathrm{AM}}$ can be expressed as follows:

$$
\begin{aligned}
L_{\mathrm{QM}} - L_{\mathrm{AM}} &= (L_{\mathrm{QM}} - L_{\mathrm{AM}}) \left( \frac{L_{\mathrm{QM}} + L_{\mathrm{AM}}}{L_{\mathrm{QM}} + L_{\mathrm{AM}}} \right) \\
&= \frac{L_{\mathrm{QM}}^2 - L_{\mathrm{AM}}^2}{L_{\mathrm{QM}} + L_{\mathrm{AM}}} = \frac{1}{L_{\mathrm{QM}} + L_{\mathrm{AM}}} \cdot \frac{1}{n} \sum_{i=1}^{n} \left( L_i - \frac{1}{n} \sum_{i=1}^{n} L_i \right)^2 .
\end{aligned}
$$

The coefficient $\frac{1}{L_{\mathrm{QM}} + L_{\mathrm{AM}}}$ in the last equation can be bounded from below and above as follows:

$$
\frac{1}{2 L_{\mathrm{QM}}} = \frac{1}{2 \cdot \max(L_{\mathrm{QM}}, L_{\mathrm{AM}})} \leq \frac{1}{L_{\mathrm{QM}} + L_{\mathrm{AM}}} \leq \frac{1}{2 \cdot \min(L_{\mathrm{QM}}, L_{\mathrm{AM}})} \leq \frac{1}{2 L_{\mathrm{AM}}}.
$$

As a consequence, difference $L_{\mathrm{QM}} - L_{\mathrm{AM}}$ is bound above by the estimated variance of $L_i$ divided by the mean of $L_i$, also known as *Index of Dispersion* in statistics. From this consideration, we can more easily observe that EF21-W can have an arbitrarily better stepsize than vanilla EF21 if the variance of $L_i$ is increasing faster than the mean of $L_i$.

### I.4 DATASET GENERATION FOR SYNTHETIC EXPERIMENT

First, we assume that the user provides two parameters: $\mu \in \mathbb{R}_+$ and $L \in \mathbb{R}_+$. These parameters define the construction of strongly convex function $f_i(x)$, which are modified by meta-parameters $q \in [-1, 1]$ and $z > 0$, described next.

1. Each client initially has

$$
f_i(x) := \frac{1}{n_i} \| \mathbf{A}_i x - b_i \|^2 ,
$$

where $\mathbf{A}_i$ is initialized in such way that $f_i$ is $L_i$ smooth and $\mu_{f_i}$ strongly convex. Parameters are defined in the following way:

$$
L_i = \frac{i}{n} \cdot (L - \mu) + \mu, \qquad \mu_{f_i} = \mu.
$$

2. The scalar value $q \in [-1, +1]$ informally plays the role of meta parameter to change the distribution of $L_i$ and make values of $L_i$ close to one of the following: (i) $\mu$; (ii) $L$; (iii) $(L + \mu)/2$. The exact modification of $L_i$ depends on the sign of meta parameter $q$.

   - Case $q \in [0, 1]$. In this case for first $n/2$ (i.e., $i \in [0, n/2]$) compute the value $L_{i,q} = \mathrm{lerp}(L_i, \mu, q)$, where $\mathrm{lerp}(a, b, t) : \mathbb{R}^d \times \mathbb{R}^d \times [0, 1] \to \mathbb{R}^d$ is standard linear interpolation
   $$
   \mathrm{lerp}(a, b, t) = a(1 - t) + bt.
   $$
   The last $n/2$ ($i \in [n/2 + 1, n]$) compute the value $L_{i,q} = \mathrm{lerp}(L_i, L, q)$. For example, if $q = 0$ then $L_{i,q} = L_i, \forall i \in [n]$, and if $q = 1$ then $L_{i,q} = \mu$ for first $n/2$ clients and $L_{i,q} = L$ for last $n/2$ clients.
   - Case $q \in [-1, 0]$. In this for all $n$ clients the new value $L_{i,q} = \mathrm{lerp}(L_i, (L + \mu)/2, -q)$. In this case for example if $q = 0$ then $L_{i,q} = L_i$ and if $q = -1$ then $L_{i,q} = (L + \mu)/2$.

   The process firstly fills the $\mathbf{A}_i$ in such form that $L_i$ forms a uniform spectrum in $[\mu, L]$ with the center of this spectrum equal to $a = \frac{L + \mu}{2}$. And then as $q \to 1$, the variance of $L_{i,q}$ is increasing.

3. We use these new values $L_{i,q}$ for all $i \in [n]$ clients as a final target $L_i^{\mathrm{new}}$ values. Due to numerical issues, we found that it's worthwhile for the first and last client to add extra scaling. First client scales $L_{1,q}$ by factor $1/z$, and last $n$-th client scales $L_{n,q}$ by factor $z$. Here $z \geq 0$ is an additional meta-parameter.

4. Next obtained values are used to generate $\mathbf{A}_i$ in such way that $\nabla^2 f_i(x)$ has uniform spectrum in $[\mu_{f_i,q,z}, L_{i,q,z}]$.

5. As a last step the objective function $f(x)$ is scaled in such a way that it is $L$ smooth with constant value $L$. The $b_i$ for each client is initialized as $b_i := \mathbf{A}_i \cdot x_{\text{solution}}$, where $x_{\text{solution}}$ is fixed solution.

## I.5 DATASET SHUFFLING STRATEGY FOR LIBSVM DATASET

Our dataset shuffling strategy heuristically splits data points so that $L_{\text{var}}$ is maximized. It consists of the following steps:

1. **Sort data points from the whole dataset according to $L$ constants.** Sort all data points according to the smoothness constants of the loss function for each single data point.

2. **Assign a single data point to each client** Assume that there are total $m$ data points in the datasets, and the total number of clients is $n$. At the beginning each client $i$ holds a single data point $\lfloor (i - 1 + 1/2) \cdot \frac{m}{n} \rfloor$.

3. **Pass through all points.** Initialize set $F = \{\}$. Next, we pass through all points except those assigned from the previous step. For each point we find the best client $i' \in [n] \backslash F$ to assign the point in a way that assignment of point to client $i'$ maximize $L_{\text{var}} := L_{\text{QM}}^2 - L_{\text{AM}}^2$. Once the client already has $\lceil \frac{m}{n} \rceil$ data points assigned to it, the client is added to the set $F$.

The set $F$ in the last step guarantees each client will have $\frac{m}{n}$ data points. In general, this is a heuristic greedy strategy that approximately maximizes $L_{\text{var}}$ under the constraint that each client has the same amount of data points equal to $\lfloor \frac{m}{n} \rfloor$. Due to its heuristic nature, the Algorithm does not provide deep guarantees, but it was good enough for our experiments.

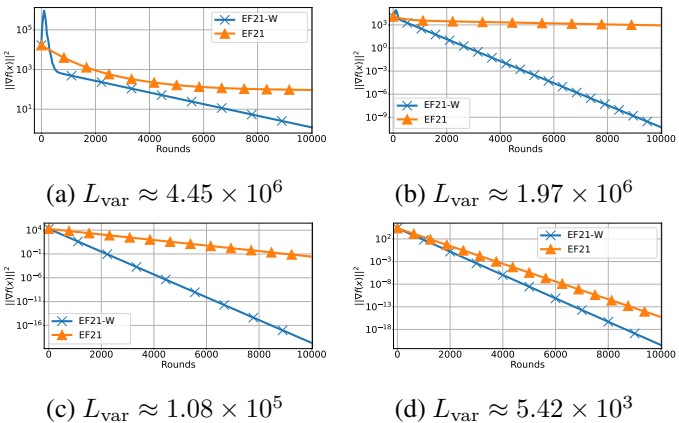

(a) $L_{\text{var}} \approx 4.45 \times 10^6$        (b) $L_{\text{var}} \approx 1.97 \times 10^6$

(c) $L_{\text{var}} \approx 1.08 \times 10^5$        (d) $L_{\text{var}} \approx 5.42 \times 10^3$

Figure 5: Convex smooth optimization. EF21 and EF21-W with Top1 client compressor, $n = 2\,000$, $d = 10$. The objective function is constitute of $f_i(x)$ defined in Eq.(86). Regularization term $\lambda \frac{\|x\|^2}{2}$, where $\lambda = 0.01$. Theoretical step size. Full participation. Extra details are in Table 1.

## J  ADDITIONAL EXPERIMENTS

In this section, we present additional experiments for comparison EF21-W, EF21-W-PP, EF21-W-SGD with their vanilla versions. We applied these algorithms in a series of synthetically generated convex and non-convex optimization problems and for training logistic regression with non-convex regularized with using several LIBSVM datasets (Chang & Lin, 2011). While carrying out additional experiments we will use three quantities. These quantities have already been mentioned in the main part, but we will repeat them here:

$$L_{\text{QM}} := \sqrt{\frac{1}{n} \sum_{i=1}^{n} L_i^2}, \quad L_{\text{AM}} := \frac{1}{n} \sum_{i=1}^{n} L_i, \quad L_{\text{var}} := L_{\text{QM}}^2 - L_{\text{AM}}^2 = \frac{1}{n} \sum_{i=1}^{n} \left( L_i - \frac{1}{n} \sum_{i=1}^{n} L_i \right)^2.$$

The relationship between these quantities was discussed in Appendix I.3. In our experiments we used TopK compressor. The TopK compressor returns sparse vectors filled with zeros, except $K$ positions, which correspond to $K$ maximum values in absolute value and which are unchanged by the compressor. Even if this compressor breaks ties arbitrarily, it is possible to show that $\alpha = \frac{K}{d}$. The compressor parameter $\alpha$ is defined without considering properties of $f_i$. The quantities $\beta$, $\theta$, $\frac{\beta}{\theta}$ are derived from $\alpha$, and they do not depend on $L_i$.

### J.1  ADDITIONAL EXPERIMENTS FOR EF21

**Convex case with synthetic datasets.** We aim to solve optimization problem (1) in the case when the functions $f_1, \ldots, f_n$ are strongly convex. In particular, we work with

$$f_i(x) := \frac{1}{n_i} \|\mathbf{A}_i x - b_i\|^2 + \frac{\lambda}{2} \|x\|^2, \tag{86}$$

where $\lambda = 0.01$. It can be shown that $L_i = \frac{2}{n_i} \lambda_{\max}(\mathbf{A}_i^\top \mathbf{A}_i) + \lambda$. The result of experiments for training linear regression model with a convex regularized is presented in Figure 5. The total number of rounds for

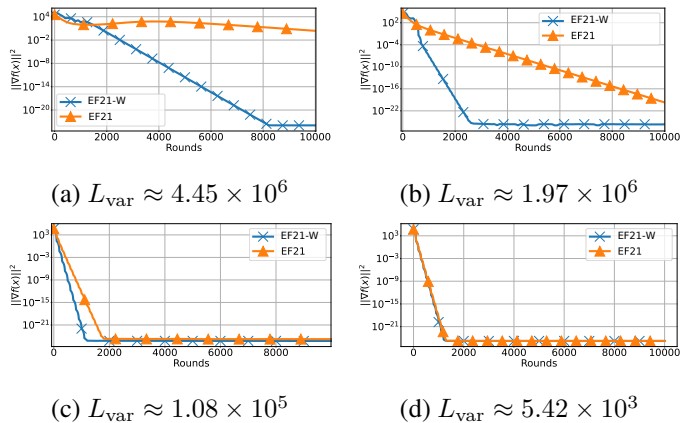

(a) $L_{\text{var}} \approx 4.45 \times 10^6$      (b) $L_{\text{var}} \approx 1.97 \times 10^6$

(c) $L_{\text{var}} \approx 1.08 \times 10^5$      (d) $L_{\text{var}} \approx 5.42 \times 10^3$

Figure 6: Non-Convex smooth optimization. `EF21` and `EF21-W` with `Top1` client compressor, $n = 2,000$, $d = 10$. The objective function is constitute of $f_i(x)$ defined in Eq. (87). Regularization term $\lambda \sum_{j=1}^{d} \frac{x_j^2}{x_j^2+1}$, with $\lambda = 100$. Theoretical step size. Full client participation. Extra details are in Table 2.

simulation is $r = 10,000$. Instances of optimization problems were generated for values $L = 50$, $\mu = 1$ with several values of $q, z$ with using the dataset generation schema described in Appendix I.4. The summary of derived quantities is presented in Table 1. We present several optimization problems to demonstrate the possible different relationships between $L_{\text{QM}}$ and $L_{\text{AM}}$. As we see from experiments, the `EF21-W` is superior as the variance of $L_i$ tends to increase. The plots in Figure 5 (a)–(d) correspond to decreasing variance of $L_i$. As we see, as the variance of $L_i$ decreases, the difference between `EF21-W` and `EF21` also tends to decrease. Finally, `EF21-W` is always at least as best as `EF21`.

Table 1: Convex Optimization experiment in Figure 5. Quantities which define theoretical step size.

| Tag | $L$ | $q$ | $z$ | $L_{\text{var}}$ | $\xi = \sqrt{\frac{\beta}{\theta}}$ | $L_{\text{QM}}$ | $L_{\text{AM}}$ | $\gamma_{\text{EF21}}$ | $\gamma_{\text{EF21}-W}$ |
|---|---|---|---|---|---|---|---|---|---|
| (a) | 50 | 1 | $10^4$ | $4.45 \times 10^6$ | 18.486 | 2111.90 | 52.04 | $2.55 \times 10^{-5}$ | $9.87 \times 10^{-4}$ |
| (b) | 50 | 1 | $10^3$ | $1.97 \times 10^6$ | 18.486 | 1408.49 | 63.56 | $3.83 \times 10^{-5}$ | $8.16 \times 10^{-4}$ |
| (c) | 50 | 1 | $10^2$ | $1.08 \times 10^5$ | 18.486 | 339.34 | 80.97 | $1.58 \times 10^{-4}$ | $6.46 \times 10^{-4}$ |
| (d) | 50 | 0.8 | 1 | $5.42 \times 10^3$ | 18.486 | 112.51 | 85.03 | $4.69 \times 10^{-4}$ | $6.16 \times 10^{-4}$ |

**Non-convex case with synthetic datasets.** We aim to solve optimization problem (1) in the case when the functions $f_1, \ldots, f_n$ are non-convex. In particular, we work with

$$f_i(x) \coloneqq \frac{1}{n_i} \|\mathbf{A}_i x - b_i\|^2 + \lambda \cdot \sum_{j=1}^{d} \frac{x_j^2}{x_j^2 + 1}. \tag{87}$$

The result of experiments for training linear regression model with a non-convex regularization is presented in Figure 6. The regularization coefficient $\lambda = 100$. Instances of optimization problems were generated for values $L = 50$, $\mu = 1$ and several values of $q, z$ for employed dataset generation schema from Appendix I.4.

The summary of derived quantities is presented in Table 2. We present various instances of optimization problems to demonstrate the different relationships between $L_{\text{QM}}$ and $L_{\text{AM}}$. As we see in the case of small variance of $L_i$ algorithm EF21-W is at least as best as standard EF21.

Table 2: Non-convex optimization experiment in Figure 6. Quantities which define theoretical step size.

| Tag | $L$ | $q$ | $z$ | $L_{\text{var}}$ | $\xi = \sqrt{\frac{\beta}{\theta}}$ | $L_{\text{QM}}$ | $L_{\text{AM}}$ | $\gamma_{\text{EF21}}$ | $\gamma_{\text{EF21}-\text{W}}$ |
|---|---|---|---|---|---|---|---|---|---|
| (a) | 50 | 1 | $10^4$ | $4.45 \times 10^6$ | 18.486 | 2126.25 | 252.035 | $2.52 \times 10^{-5}$ | $2.03 \times 10^{-4}$ |
| (b) | 50 | 1 | $10^3$ | $1.97 \times 10^6$ | 18.486 | 1431.53 | 263.55 | $3.74 \times 10^{-5}$ | $1.95 \times 10^{-4}$ |
| (c) | 50 | 1 | $10^2$ | $1.08 \times 10^5$ | 18.486 | 433.05 | 280.958 | $1.21 \times 10^{-4}$ | $1.83 \times 10^{-4}$ |
| (d) | 50 | 0.8 | 1 | $5.42 \times 10^3$ | 18.486 | 294.39 | 285.022 | $1.17 \times 10^{-4}$ | $1.81 \times 10^{-4}$ |

**Non-convex logistic regression on benchmark datasets.** We aim to solve optimization problem (1) in the case when the functions $f_1, \ldots, f_n$ are non-convex. In particular, we work with logistic regression with a non-convex robustifying regularization term:

$$f_i(x) := \frac{1}{n_i} \sum_{j=1}^{n_i} \log\left(1 + \exp(-y_{ij} \cdot a_{ij}^\top x)\right) + \lambda \cdot \sum_{j=1}^{d} \frac{x_j^2}{x_j^2 + 1}, \tag{88}$$

where $(a_{ij}, y_{ij}) \in \mathbb{R}^d \times \{-1, 1\}$.

We used several LIBSVM datasets (Chang & Lin, 2011) for our benchmarking purposes. The results are presented in Figure 7 and Figure 8. The important quantities for these instances of optimization problems are summarized in Table 3. From Figures 7 (a), (b), (c), we can observe that for these datasets, the EF21-W is better, and this effect is observable in practice. For example, from these examples, we can observe that 12.5K rounds of EF21-W corresponds to only 10K rounds of EF21. This improvement is essential for Federated Learning, in which both communication rounds and communicate information during a round represent the main bottlenecks and are the subject of optimization. Figures 7 (d), (e), (f) demonstrate that sometimes the EF21-W can have practical behavior close to EF21, even if there is an improvement in step-size (For exact values of step size see Table 3). The experiment on AUSTRALIAN datasets are presented in Figure 8. This example demonstrates that in this LIBSVM benchmark datasets, the relative improvement in the number of rounds for EF21-W compared to EF21 is considerable. For example 40K rounds of EF21 corresponds to 5K rounds of EF21-W in terms of attainable $\|\nabla f(x^t)\|^2$.

Table 3: Non-convex optimization experiments in Figures 7, 8. Derived quantities which define theoretical step size.

| Tag | $L$ | $L_{\text{var}}$ | $\xi = \sqrt{\frac{\beta}{\theta}}$ | $L_{\text{QM}}$ | $L_{\text{AM}}$ | $\gamma_{\text{EF21}}$ | $\gamma_{\text{EF21}-\text{W}}$ |
|---|---|---|---|---|---|---|---|
| (a) W1A | 0.781 | 3.283 | 602.49 | 2.921 | 2.291 | $5.678 \times 10^{-4}$ | $7.237 \times 10^{-4}$ |
| (b) W2A | 0.784 | 2.041 | 602.49 | 2.402 | 1.931 | $6.905 \times 10^{-4}$ | $8.589 \times 10^{-4}$ |
| (c) W3A | 0.801 | 1.579 | 602.49 | 2.147 | 1.741 | $7.772 \times 10^{-4}$ | $9.523 \times 10^{-4}$ |
| (d) MUSHROOMS | 2.913 | $5.05 \times 10^{-1}$ | 226.498 | 3.771 | 3.704 | $1.166 \times 10^{-3}$ | $1.187 \times 10^{-3}$ |
| (e) SPLICE | 96.082 | $2.23 \times 10^2$ | 122.497 | 114.43 | 113.45 | $7.084 \times 10^{-5}$ | $7.14 \times 10^{-5}$ |
| (f) PHISHING | 0.412 | $9.2 \times 10^{-4}$ | 138.498 | 0.429 | 0.428 | $1.670 \times 10^{-2}$ | $1.674 \times 10^{-2}$ |
| (g) AUSTRALIAN | $3.96 \times 10^6$ | $1.1 \times 10^{16}$ | 18.486 | $3.35 \times 10^7$ | $3.96 \times 10^6$ | $9.733 \times 10^{-10}$ | $8.007 \times 10^{-9}$ |

**Non-convex logistic regression with non-homogeneous compressor.** In this supplementary experiment, we leveraged the AUSTRALIAN LIBSVM datasets (Chang & Lin, 2011) to train logistic regression, incorpo-

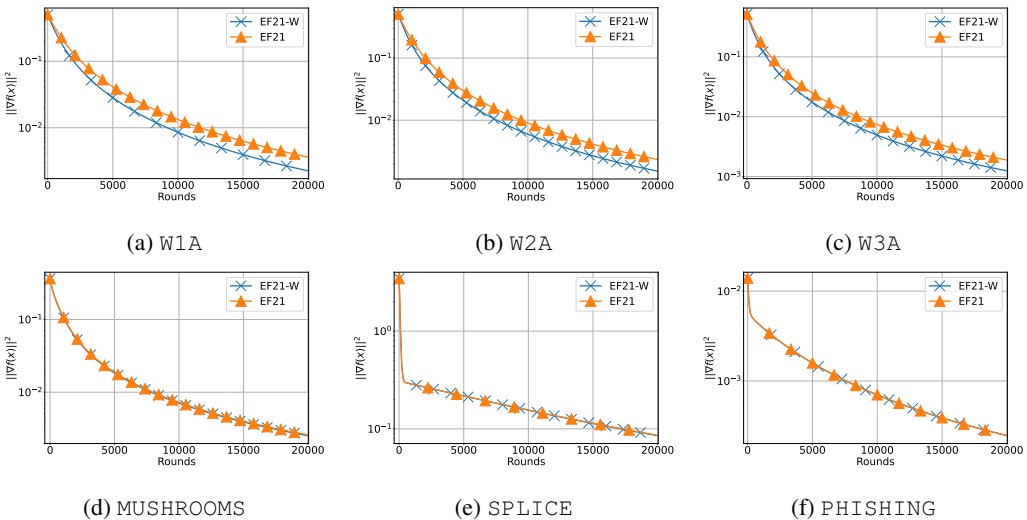

| (a) W1A | (b) W2A | (c) W3A |
|---------|---------|---------|

| (d) MUSHROOMS | (e) SPLICE | (f) PHISHING |
|---------------|------------|--------------|

Figure 7: Non-Convex Logistic Regression: comparison of EF21 and EF21-W. The used compressor is Top1. The number of clients $n = 1,000$. Regularization term $\lambda \sum_{j=1}^{d} \frac{x_j^2}{x_j^2+1}$, with $\lambda = 0.001$. Theoretical step size. Full client participation. The objective function is constitute of $f_i(x)$ defined in Eq. (88). Extra details are in Table 3.

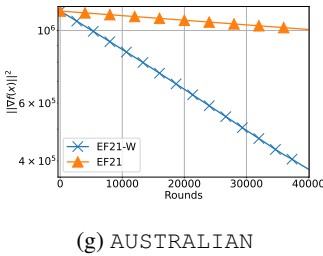

(g) AUSTRALIAN

Figure 8: Non-Convex Logistic Regression: comparison of the performance of standard EF21 and EF21-W. The used compressor is Top1. The number of clients $n = 200$. Regularization term $\lambda \sum_{j=1}^{d} \frac{x_j^2}{x_j^2+1}$, with $\lambda = 1,000$. Theoretical step size. The objective function is constitute of $f_i(x)$ defined in Eq. (88). Extra details are in Table 3.

rating a non-convex sparsity-enhanced regularization term defined in Eq. (88). The experiment featured the utilization of a non-homogeneous compressor known as Natural by Horváth et al. (2022), belonging to the family of unbiased compressors and adhering to Definition 3 with $w = 1/8$. This compressor, in a randomized manner, rounds the exponential part and zeros out the transferred mantissa part when employing the standard IEEE 754 Standard for Floating-Point Arithmetic IEEE Computer Society (2008) representation for floating-point numbers. Consequently, when using a single float-point format (FP32) during communication, only 9 bits of payload per scalar need to be sent to the master, and the remaining 23 bits of mantissa can be entirely dropped.

The experiment results are depicted in Figure 9. In this experiment, we fine-tuned the theoretical step size by multiplying it with a specific constant. As we can see the EF21-W consistently outperforms EF21 across all corresponding step-size multipliers. As we see EF21-W operates effectively by utilizing unbiased non-

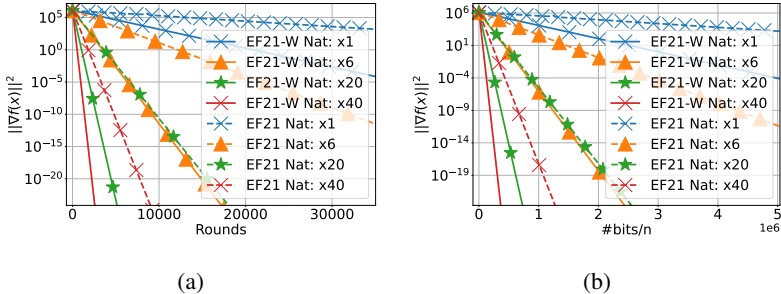

(a)                                          (b)

Figure 9: Non-Convex Logistic Regression: comparison of the performance of standard EF21 and EF21-W. The used compressor for EF21 and EF21-W is Natural compressor Horváth et al. (2022). The number of clients $n = 200$. The objective function is constitute of $f_i(x)$ defined in Eq. (88). Regularization term $\lambda \sum_{j=1}^{d} \frac{x_j^2}{x_j^2+1}$, with $\lambda = 1,000$. Multipliers of theoretical step size. Full participation. Computation format single precision (FP32). Dataset: AUSTRALIAN.

homogeneous compressors, and the advantages over EF21 extend beyond the scope of applying EF21-W solely to homogeneous compressors. Finally, it is worth noting that the increased theoretical step size in EF21-W does not entirely capture the practical scenario of potentially enhancing the step size by a significantly large multiplicative factor (e.g., ×40), which remains a subject for future research.

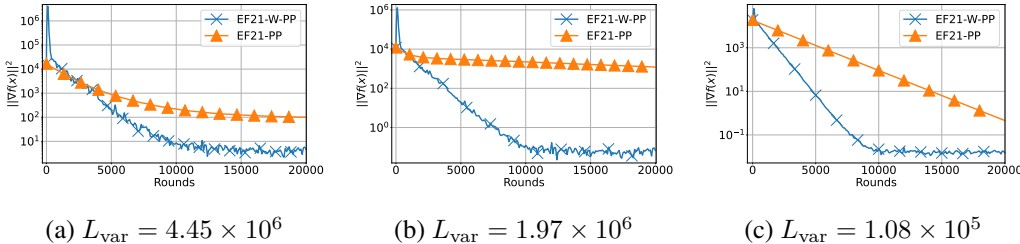

(a) $L_{\mathrm{var}} = 4.45 \times 10^6$      (b) $L_{\mathrm{var}} = 1.97 \times 10^6$      (c) $L_{\mathrm{var}} = 1.08 \times 10^5$

Figure 10: Convex smooth optimization. `EF21-PP` and `EF21-W-PP` with `Top1` client compressor, $n = 2\,000$, $d = 10$. The objective function is constitute of $f_i(x)$ defined in Eq. (89). Regularization term $\lambda\frac{\|x\|^2}{2}$, $\lambda = 0.01$. Theoretical step size. The objective function is constitute of $f_i(x)$ defined in Eq.(89). Each client participates in each round with probability $p_i = 0.5$. Extra details are in Table 4.

## J.2 Additional experiments for `EF21-W-PP`

**Convex case with synthetic datasets.**

Table 4: Convex optimization experiment in Figure 10. Derived quantities which define theoretical step size.

| Tag | $L$ | $q$ | $z$ | $L_{\mathrm{var}}$ | $\sqrt{\frac{\beta}{\theta}}$ | $L_{\mathrm{QM}}$ | $L_{\mathrm{AM}}$ | $\gamma_{\text{EF21-PP}}$ | $\gamma_{\text{EF21-W-PP}}$ |
|-----|-----|-----|-----|-----|-----|-----|-----|-----|-----|
| (a) | 50 | 1 | $10^4$ | $4.45 \times 10^6$ | 18.486 | 2111.90 | 52.04 | $2.55 \times 10^{-5}$ | $9.87 \times 10^{-4}$ |
| (b) | 50 | 1 | $10^3$ | $1.97 \times 10^6$ | 18.486 | 1408.49 | 63.56 | $3.83 \times 10^{-5}$ | $8.16 \times 10^{-4}$ |
| (c) | 50 | 1 | $10^2$ | $1.08 \times 10^5$ | 18.486 | 339.34 | 80.97 | $1.58 \times 10^{-4}$ | $6.46 \times 10^{-4}$ |

We aim to solve optimization problem (1) in the case when the functions $f_1, \ldots, f_n$ are strongly convex. In particular, we choose:

$$f_i(x) := \frac{1}{n_i}\|\mathbf{A}_i x - b_i\|^2 + \frac{\lambda}{2}\|x\|^2. \tag{89}$$

In this synthetic experiment, we have used the maximum allowable step size suggested by the theory of `EF21-PP` and for the proposed `EF21-W-PP` algorithm. The initial gradient estimators have been initialized as $g_i^0 = \nabla f_i(x^0)$ for all $i$. The number of clients in simulation $n = 2000$, dimension of optimization problem $d = 10$, number of samples per client $n_i = 10$, and number of communication rounds is $r = 10,000$. For both `EF21-PP` and `EF21-W-PP` clients we used `Top1` biased contractile compressor. In our experiment, each client's participation in each communication round is governed by an independent Bernoulli trial which takes $p_i = 0.5$. The result of experiments for training linear regression model with a convex regularizer is presented in Figure 10. The regularization constant was chosen to be $\lambda = 0.01$. Instances of optimization problems were generated for values $L = 50, \mu = 1$ with several values of $q$ and $z$. The summary of derived quantities is presented in Table 4. We present several optimization problems to demonstrate the possible different relationships between $L_{\mathrm{QM}}$ and $L_{\mathrm{AM}}$. As we see from experiments, the `EF21-W-PP` is superior as the variance of $L_i$ tends to increase. As we can observe `EF21-W-PP` is always at least as best as `EF21-PP`.

**Non-convex logistic regression on benchmark datasets.** We provide additional numerical experiments in which we compare `EF21-PP` and `EF21-W-PP` for solving (1). We address the problem of training a binary classifier via a logistic model on several `LIBSVM` datasets (Chang & Lin, 2011) with non-convex

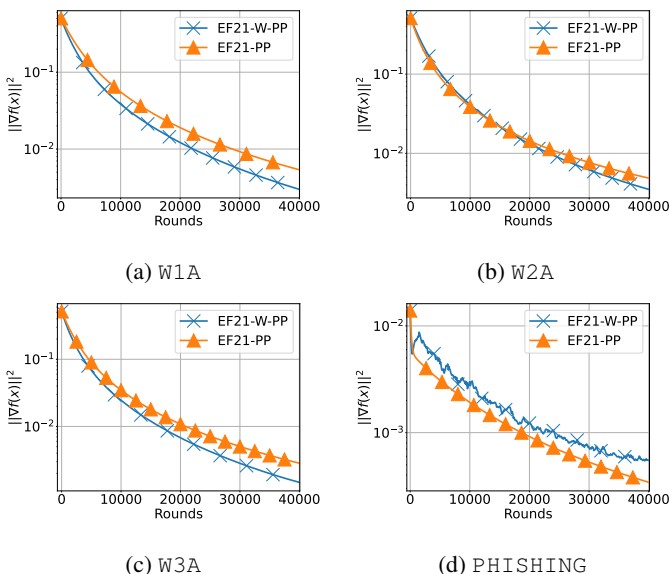

(a) `W1A`

(b) `W2A`

(c) `W3A`

(d) `PHISHING`

Figure 11: Non-Convex Logistic Regression: comparison of `EF21-PP` and `EF21-W-PP`. The used compressor is `Top1`. The number of clients $n = 1,000$. Regularization term $\lambda \sum_{j=1}^{d} \frac{x_j^2}{x_j^2+1}$, $\lambda = 0.001$. Theoretical step size. Each client participates in each round with probability $p_i = 0.5$. The objective function is constitute of $f_i(x)$ defined in Eq.(90). Extra details are in Table 5.

regularization. We consider the case when the functions $f_1, \ldots, f_n$ are non-convex; in particular, we set $f_i(x)$ as follows:

$$f_i(x) := \frac{1}{n_i} \sum_{j=1}^{n_i} \log\left(1 + \exp(-y_{ij} \cdot a_{ij}^\top x)\right) + \lambda \cdot \sum_{j=1}^{d} \frac{x_j^2}{x_j^2 + 1}, \tag{90}$$

where $(a_{ij}, y_{ij}) \in \mathbb{R}^d \times \{-1, 1\}$.

We conducted distributed training of a logistic regression model on `W1A`, `W2A`, `W3A`, `PHISHING`, and `AUSTRALIAN` datasets with non-convex regularization. The initial gradient estimators are set $g_i^0 = \nabla f_i(x^0)$ for all $i \in [n]$. For comparison of `EF21-PP` and `EF21-W-PP`, we used the largest step size allowed by theory. We used the dataset shuffling strategy described in Appendix I.5. The results are presented in Figure 11 and Figure 12. The important quantities for these instances of optimization problems are summarized in Table 5.

Table 5: Non-Convex optimization experiments in Figures 11, 12. Quantities which define theoretical step size.

| Tag | $L$ | $L_{\text{var}}$ | $L_{\text{QM}}$ | $L_{\text{AM}}$ | $\gamma_{\text{EF21-PP}}$ | $\gamma_{\text{EF21-W-PP}}$ |
|---|---|---|---|---|---|---|
| (a) W1A | 0.781 | 3.283 | 2.921 | 2.291 | $2.315 \times 10^{-4}$ | $2.95 \times 10^{-4}$ |
| (b) W2A | 0.784 | 2.041 | 2.402 | 1.931 | $2.816 \times 10^{-4}$ | $3.503 \times 10^{-4}$ |
| (c) W3A | 0.801 | 1.579 | 2.147 | 1.741 | $3.149 \times 10^{-4}$ | $3.884 \times 10^{-4}$ |
| (d) PHISHING | 0.412 | $9.2 \times 10^{-4}$ | 0.429 | 0.428 | $6.806 \times 10^{-3}$ | $6.823 \times 10^{-3}$ |
| (e) AUSTRALIAN | $3.96 \times 10^6$ | $1.1 \times 10^{16}$ | $3.35 \times 10^7$ | $3.96 \times 10^6$ | $3.876 \times 10^{-10}$ | $3.243 \times 10^{-9}$ |

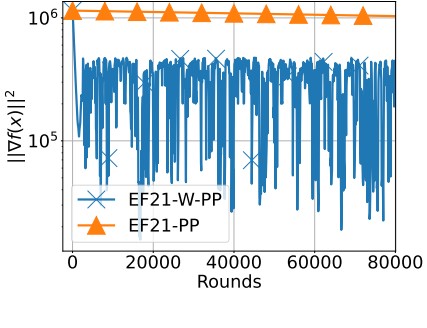

(e) AUSTRALIAN

Figure 12: Non-Convex Logistic Regression: comparison of EF21-PP and EF21-W-PP. The used compressor is Top1. The number of clients $n = 200$. Regularization term $\lambda \sum_{j=1}^{d} \frac{x_j^2}{x_j^2+1}$, with $\lambda = 1,000$. Theoretical step size. Each client participates in each round with probability $p_i = 0.5$. The objective function is constitute of $f_i(x)$ defined in Eq.(90). Extra details are in Table 5.

From Figure 11 (a), (b), (c), we can observe that for these datasets, the EF21-W-PP is better, and this effect is observable in practice and is not negligible. Figures 11 (d), demonstrate that sometimes EF21-W-PP in terms of the full gradient at last iterate can have slightly worse behavior compared to EF21-PP, even though theory allow more aggressive step-size (For exact values of step size see Table 5. The experiment on AUSTRALIAN dataset is presented in Figure 12. This example demonstrates that in this LIBSVM benchmark datasets, the relative improvement in the number of rounds for EF21-W-PP compared to EF21-PP is considerable. The EF21-W-PP exhibits more oscillation behavior in terms of $\|\nabla f(x^t)\|^2$ for AUSTRALIAN dataset, however as we can see observe in expectation $\|\nabla f(x^t)\|^2$ tends to decrease faster compare to EF21-PP.

## J.3 ADDITIONAL EXPERIMENTS FOR EF21-W-SGD

The standard EF21-SGD with the analysis described in Corollary 4 (Fatkhullin et al., 2021) allows performing the optimization procedure with maximum allowable step size up to the factor of 2 equal to:

$$\gamma_{\text{EF21-SGD}} = \left( L + \sqrt{\frac{\hat{\beta}_1}{\hat{\theta}}} L_{\text{QM}} \right)^{-1}.$$

In last expression quantities $\hat{\theta} = 1 - (1-\alpha)(1+s)(1+\nu)$, and $\hat{\beta}_1 = 2(1-\alpha)(1+s)(s+\nu^{-1})$. Improved analysis for EF21-W-SGD allows to apply step size:

$$\gamma_{\text{EF21-W-SGD}} = \left( L + \sqrt{\frac{\hat{\beta}_1}{\hat{\theta}}} L_{\text{AM}} \right)^{-1}.$$

Therefore in terms of step size

$$\gamma_{\text{EF21-W-SGD}} \geq \gamma_{\text{EF21-SGD}}$$

and EF21-W-SGD exhibits a more aggressive step-size.

We conducted distributed training of a logistic regression model on `W1A`, `W2A`, `W3A`, `PHISHING`, `AUSTRALIAN` datasets with non-convex regularization. For all datasets, we consider the optimization problem (1), where

$$f_i(x) := \frac{1}{n_i} \sum_{j=1}^{n_i} \log\left(1 + \exp(-y_{ij} \cdot a_{ij}^\top x)\right) + \lambda \sum_{j=1}^{d} \frac{x_j^2}{x_j^2 + 1}, \tag{91}$$

and $(a_{ij}, y_{ij}) \in \mathbb{R}^d \times \{-1, 1\}$.

The initial gradient estimators are set to $g_i^0 = \nabla f_i(x^0)$ for all $i \in [n]$. For comparison of EF21-SGD and EF21-W-SGD, we used the largest step size allowed by theory. The dataset shuffling strategy repeats the strategy that we have used for EF21-W-PP and EF21-W and it is described in Appendix I.5. The algorithms EF21-SGD and /EF21-W-SGD employed an unbiased gradient estimator, which was estimated by sampling a single training point uniformly at random and independently at each client.

Table 6: Non-Convex optimization experiments in Figures 13, 14. Quantities which define theoretical step size.

| Tag | $L$ | $L_{\text{var}}$ | $L_{\text{QM}}$ | $L_{\text{AM}}$ | $\gamma_{\text{EF21-SGD}}$ | $\gamma_{\text{EF21-W-SGD}}$ |
|---|---|---|---|---|---|---|
| (a) W1A | 0.781 | 3.283 | 2.921 | 2.291 | $4.014 \times 10^{-4}$ | $5.118 \times 10^{-4}$ |
| (b) W2A | 0.784 | 2.041 | 2.402 | 1.931 | $4.882 \times 10^{-4}$ | $6.072 \times 10^{-4}$ |
| (c) W3A | 0.801 | 1.579 | 2.147 | 1.741 | $5.460 \times 10^{-4}$ | $6.733 \times 10^{-4}$ |
| (f) PHISHING | 0.412 | $9.2 \times 10^{-4}$ | 0.429 | 0.428 | $1.183 \times 10^{-2}$ | $1.186 \times 10^{-2}$ |
| (g) AUSTRALIAN | $3.96 \times 10^6$ | $1.1 \times 10^{16}$ | $3.35 \times 10^7$ | $3.96 \times 10^6$ | $3.876 \times 10^{-10}$ | $3.243 \times 10^{-9}$ |

The results are presented in Figure 13 and Figure 14. The important quantities for these instances of optimization problems are summarized in Table 6. In all Figures 13 (a), (b), (c), (d) we can observe that for these datasets, the EF21-W-SGD is better, and this effect is observable in practice. The experiment on `AUSTRALIAN` datasets are presented in Figure 14. This example demonstrates that in this `LIBSVM` benchmark datasets, the relative improvement in the number of rounds for EF21-W-SGD compared to EF21-SGD is considerable. Finally, we address oscillation behavior to the fact that employed step size for EF21-SGD is too pessimistic, and its employed step size removes oscillation of $\|\nabla f(x^t)\|^2$.

## K    REPRODUCIBILITY STATEMENT

To ensure reproducibility, we use the following `FL_PyTorch` simulator features: (i) random seeds were fixed for data synthesis; (ii) random seeds were fixed for the runtime pseudo-random generators involved in EF21-PP and EF21-SGD across clients and the server; (iii) the thread pool size was turned off to avoid the non-deterministic order of client updates in the server.

If you are interested in the source code for all experiments, please contact the authors.

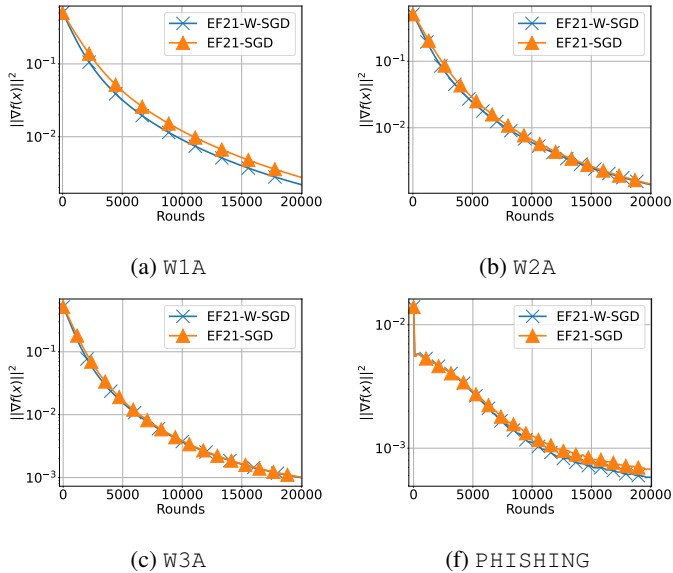

(a) W1A        (b) W2A

(c) W3A        (f) PHISHING

Figure 13: Non-Convex logistic regression: comparison of EF21-SGD and EF21-W-SGD. The used compressor is Top1. The SGD gradient estimator is SGD-US, $\tau = 1$. The number of clients $n = 1,000$. The objective function is constitute of $f_i(x)$ defined in Eq.(91). Regularization term $\lambda \sum_{j=1}^{d} \frac{x_j^2}{x_j^2 + 1}$, $\lambda = 0.001$. Theoretical step size. See also Table 6.

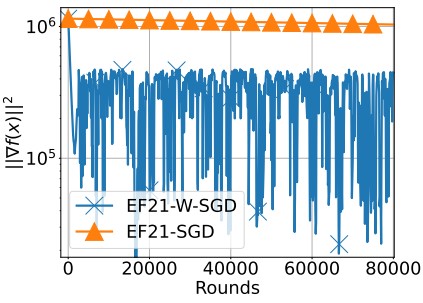

AUSTRALIAN

Figure 14: Non-Convex logistic regression: comparison of EF21-SGD and EF21-W-SGD. The used compressor is Top1. The SGD gradient estimator is SGD-US, $\tau = 1$. The number of clients $n = 200$. The objective function is constitute of $f_i(x)$ defined in Eq.(91). Regularization term $\lambda \sum_{j=1}^{d} \frac{x_j^2}{x_j^2 + 1}$, with $\lambda = 1,000$. Theoretical step size. Full participation. Extra details are in Table 6.

