# OpenReview forum: "Error Feedback Reloaded: From Quadratic to Arithmetic Mean of Smoothness Constants"
_ICLR.cc/2024/Conference — ICLR 2024 poster_

### Official Review · Reviewer_XNZi · 2023-10-24

**Soundness:** 3 good
**Presentation:** 4 excellent
**Contribution:** 3 good
**Rating:** 6
**Confidence:** 2

**Summary:**

The authors offers a refined analysis of the popular EF21 algorithm to show that this algorithm actually depends on the arithmetic mean of Lipschitz constants instead of the quadratic mean. In the process the authors also proposed new versions of EF21, and showed that this analysis can be extended to a wide variety of EF21 variants.

**Strengths:**

1. Improves theoretical understanding of existing popular approach.
2. Presentation is very clear, and easy to follow
3. Contribution is meaningful, because it further promotes the use of EF21 under certain cases.

**Weaknesses:**

1. Did not propose a newer version of EF21 that outperforms the original one (probably not a weakness, but more of a hope)

**Questions:**

N/A

---

> ### Author Response · Authors · 2023-11-22
> **Re "Strengths"**
>
> Thanks!!!

---

> ### Author Response · Authors · 2023-11-22
> **Re "Weaknesses"**
>
> > Did not propose a newer version of EF21 that outperforms the original one (probably not a weakness, but more of a hope)
>
> Well, we actually did propose a new version of EF21 that outperforms the original one: this is what EF21-W does! Indeed, EF21-W (Algorithm 2) is a new variant of the classical EF21 method described in Algorithm 1, and Theorem 3 clearly shows dependence on $L_{AM}$ whereas the classical result for EF21 from Richtarik et al (2021) has a dependence on $L_{QM}$.
>
> We could have stopped our research here, and this would indeed prove that EF21-W is better than EF21!
>
> However, as described in Section 2.4, we proceeded with our research deeper, and noticed that for positively homogeneous compressors, such as TopK, EF21-W is in some sense equivalent to EF21! This means that, at least for this class of compressors, our improvements apply to EF21 as well. We pressed on, and asked whether the positive homogeneity assumption is necessary, or can be lifted. Theorem 4 says that it can be lifted. So, after all, we discovered that original EF21, with no change apart from the choice of a larger stepsize, also enjoys this better rate!
>
> So, we first discovered a new method which outperforms EF21, and then in turn discovered that original EF21 can be analyzed in a tighter manner to obtain the same improvements.
>
> Of course, the new open problem is whether our rate for EF21 or EF21-W can be improved even further, either for the same method, or for a new variant of error feedback.

---

> ### Author Response · Authors · 2023-11-22
> **Did we address your concerns?**
>
> It seems to us that you have not raised any issue with our work, and hence we are surprised by the marginal accept score. If you think it would be more appropriate to assign a higher score because of this, we would certainly appreciate it.
>
> If some questions remain in your mind which you did not ask, please do ask. We are happy to respond.
>
> Thanks for reading our paper.

---

### Official Review · Reviewer_ZKKe · 2023-11-01

**Soundness:** 3 good
**Presentation:** 3 good
**Contribution:** 3 good
**Rating:** 8
**Confidence:** 3

**Summary:**

This paper presents with a weighted version of EF21, a modern form of error feedback, which previously offers the best theoretical guarantees under the weakest assumptions. And the proposed weighted version of EF21 improved the dependence on the quadratic mean of certain smoothness parameters to their arithmetic mean.

In addition to the detailed description of the weighted error feedback algorithms, this paper introduces the steps took to discover the weighted EF21, starting from adding one more client node to enhance the convergence rate in EF21, to generalizing the cloning idea, replacing the quadratic mean of the smoothness parameters to their arithmetic mean, improving convergence rate, to overcoming the shortcomings with cloning clients with weighted version of EF21.

Finally, the paper conducted experiments on non-convex logistic regression on benchmark datasets and non-convex linear regression on synthetic datasets, showing a faster convergence rate in situations of high variance in smoothness constants.

**Strengths:**

This paper proposed an intuitive algorithm based on EF21, improving the convergence rate in cases of high variance in smoothness constants, taking the readers through the journey of discovering the algorithm. The process of discovery is clearly described in the paper with simple and effective examples. The math presented in the main paper is enough to clarify the core idea behind the algorithm, and more detailed explanations can be found in the appendix. After taking the readers through the discovery process, the paper presents with 2 sets of experiments conducted on the new algorithm, accompanied by clear figures illustrating the improvement of the algorithm over the original EF21.

**Weaknesses:**

This paper did not discuss the choices of compressor used in the experiments, as in the original paper of EF21. In the EF21 paper, experiments were conducted on fine-tuning k and the step sizes, which is overlooked in this paper.

This paper conducted experiments under specific settings with limited explanation for the parameters chosen, such as the number of clients in each experiment.

**Questions:**

Why is top1 compressor chosen to be employed in all experiments? If the algorithm is designed for a specific setting, can that be clearly described in the paper?

Why is the number of clients set to be 1000 in the logistic regression experiments, and set to be 2000 in the linear regression experiments? Is it chosen for a specific problem setting? Will the algorithm behave differently with different number of clients?

---

> ### Author Response · Authors · 2023-11-22
> **Strengths**
>
> Thanks!!!

---

> ### Author Response · Authors · 2023-11-22
> **Weaknesses - part 1**
>
> > This paper did not discuss the choices of compressor used in the experiments, as in the original paper of EF21. In the EF21 paper, experiments were conducted on fine-tuning $k$ and the step sizes, which is overlooked in this paper….  Why is the top compressor chosen to be employed in all experiments?
>
> Our theory holds for *any* contractive compressor, and there are tens of them in the literature, most with a hyper-parameter than can be chosen in many ways; see Beznosikov et al, On biased compression for distributed learning, arXiv:2002.12410, 2020. Our goal in this paper is *not* to find the best performing compressor, since our work's contribution are in an orthogonal direction. Instead, our experiments are meant to support our theoretical findings. And this can be done for any choice of the compressor.
>
> - We have used TopK with $K=1$. We described all derived quantities from the optimization problem instances in the tables mentioned in figure captions.
>
> - We did not tune stepsizes, but we are happy to include such experiments as well (they would be less important than the ones we have though). Note that one of our key theoretical findings is that EF21 can be provably used with larger stepsizes (see Theorem 4). In order to illustrate numerically what this theorem says, we need to compare EF21 with standard stepsizes vs EF21 with the new larger stepsizes. We do not have such experiments, and can add them. However, we do have a experiments where we compare EF21 with standard stepsizes vs EF21-W. However, for the compressors we employ, the performance of EF21-W and EF21 with the larger stepsizes from Theorem 4 will be identical. Indeed, this is because the compressors we experiment with are positively homogeneous, and in this case the transition from EF21-W (Sec 2.3) to EF21 with large stepsizes (Sec 2.4) is exact. That is, the two approaches described in the two sections are equivalent for positively homogeneous compressors.  Having said that, one would expect even better performance with "tuned" stepsizes. But this is not what our paper is about, and this is why we did not include such experiments. The observed improvements gained in the step size for Logistic Regression experiments with a non-convex regularization are described in Table 3, p.47. As you can see the improvement is not so dramatic, namely the average improvement of the step size in this benchmark dataset is about 1.37. Our theory does not explain why EF21 can be improved by big factors as significant as $60\times$ mentioned in the empirical study from: https://arxiv.org/pdf/2110.03294.pdf, p.10, Figure 2.

---

> ### Author Response · Authors · 2023-11-22
> **Weaknesses - part 2**
>
> > If the algorithm is designed for a specific setting, can that be clearly described in the paper?  Why is the number of clients set to be 1000 in the logistic regression experiments, and set to be 2000 in the linear regression experiments?  Is it chosen for a specific problem setting?
>
> Our methods and results are generic, and our theory works for all contractive compressors, for any number of clients $n$ and any number of parameters $d$. However, communication compression is typically useful when $d$ is fairly large.
>
> The choice of $n=1000/2000$ clients is arbitrary; we could have chosen some other number.
>
> Our key point was to construct some experiments where the difference between $L_{QM}$ and $L_{AM}$ is  small and some where it is large, so that we can test whether indeed we shall observe better performance in cases when the difference is larger, as our theory predicts. Our experiments confirm this to be the case.
>
> > Will the algorithm behave differently with a different number of clients?
>
> Yes, of course, this is the case for all distributed methods.
>
> - Note that our theoretical improvements for the vanilla version of our method (i.e., those described in the main paper as opposed to the appendix) do not explicitly depend on $n$; they explicitly depend on the constants $L_{QM}$ and $L_{AM}$. This is why we control for these smoothness averages instead. By changing the number of clients, the data needs to be partitioned differently, and this will influence the smoothness constants $L_i$, and indirectly also the smoothness averages $L_{QM}$ and $L_{AM}$. So yes, the number of clients will influence the performance; and this is captured in our theory in this way. However, for our purposes it is easier to control for these smoothness averages directly and not by playing with $n$.
>
>
> - For EF21-W and EF21-PP the dependence on $n$ comes from the average of $L_i$ across the clients, and during computing the average of $L_i$ there is a division by factor $n$. [See Theorem 3, page 6; and Theorem 7, p.33.].
>
> - The convergence of EF21-W-SGD is driven by Theorem 6, p.27. The upper bound on the stepsize does not depend on $n$ explicitly, only implicitly via $L_{AM}$. For EF21-W-SGD, dependence on $n$ also comes through $\widetilde{A}$. So changing $n$ can potentially affect $\widetilde{A}$.
>
> - The convergence of EF21-W in the Rare Features regime is driven by Theorem 8, p.39. Firstly, the upper bound on the stepsize does not depend on $n$ explicitly, only implicitly via the mean of $L_i$. Second, the dependence on $n$ comes through the multiplier $c/n$. If you consider problems instances with increasing $n$with fixed $c$, $c/n$ decreases, this allows larger stepsize, and this improved the rate -- see Equation (76).

---

### Official Review · Reviewer_uTDX · 2023-11-01

**Soundness:** 3 good
**Presentation:** 1 poor
**Contribution:** 2 fair
**Rating:** 3
**Confidence:** 3

**Summary:**

This paper proposes a new weighted version of EF21 with better theoretical guarantees. Experiments show that the proposed EF21-W outperforms the baselines.

**Strengths:**

1. This paper proposes a new weighted version of EF21 with better theoretical guarantees.

2. Experiments show that the proposed EF21-W outperforms the baselines.

**Weaknesses:**

1. It seems that the proposed EF21-W is simply a method that uses the local smoothness constants to obtain a weighted average of the local gradients sent by the workers, which actually has nothing to do with EF21. I mean, if we totally remove the communication compression and EF21 parts, we could still obtain a distributed gradient descent algorithms with weighted average of gradients on the server side and the same improvement in the theoretical results (from quadratic mean to arithmetic mean of the smoothness constants). Adding communication compression and EF21 seems too deliberate to me and prevents the paper from presenting the core idea in a clean and crispy manner.

2. The experiments seem too simple and small to me. I mean, the linear/logistic regression models on libsvm datasets could be easily trained in a short time on some modern and cheap hardware such as a normal laptop, which doesn't require distributed training or communication at all. I understand these experiments are meant to verify the theoretical results, but the experiment settings are just too simple and synthetic, which makes the experiments seem meaningless to me.

3. The proposed algorithm, EF21-W is far from practical, since in real-world complicated models and training tasks, typically it is very difficult to obtain the smoothness constants, while the smoothness constants are essential for EF21-W.

**Questions:**

1. If we simply re-weight the gradients with $w_i$ in the gradient averaging without using communication compression or EF21, would we obtain the same improvement in the theoretical results (from quadratic mean to arithmetic mean of the smoothness constants)?

2. For the more complicated models or tasks, such as neural networks, how to obtain the smoothness constants and apply EF21-W?

---

> ### Author Response · Authors · 2023-11-22
> **Weakness 1**
>
> > 1. It seems that the proposed EF21-W is simply a method that uses the local smoothness constants to obtain a weighted average of the local gradients sent by the workers, which actually has nothing to do with EF21. I mean, if we totally remove the communication compression and EF21 parts, we could still obtain a distributed gradient descent algorithms with weighted average of gradients on the server side and the same improvement in the theoretical results (from quadratic mean to arithmetic mean of the smoothness constants). Adding communication compression and EF21 seems too deliberate to me and prevents the paper from presenting the core idea in a clean and crispy manner.
>
> This is simply just not true. Let's explain:
> - Notice that our improvements hold for EF21 as well, see Theorem 4 in Section 2.4. This was ignored by the reviewer. EF21 does not do any weighting! (Weighting only appears in the proof, not in the method). So, at least in this case, our improvement clearly can not be explained the way you try to explain it. The same logic applies to EF21 applied to the cloning reformulation described in Section 2.2.
> - Note that if $C_i^t(x)\equiv x$ for all $i$ and $t$, i.e., if no compression is performed, then the gradient estimator $g^{t+1}$ in EF21-W has the form $$ g^{t+1} = \sum_{i=1}^n w_i g_i^{t+1} = \sum_{i=1}^n w_i (g_i^t + u_i^t) = \sum_{i=1}^n w_i g_i^t + \sum_{i=1}^n w_i \left(\frac{1}{n w_i} \nabla f_i(x^{t+1}) - g_i^{t} \right) = \frac{1}{n}\sum_{i=1}^n \nabla f_i(x^{t+1}) = \nabla f(x^{t+1}).$$
> In other words, EF21-W turns into vanilla gradient descent. There are no weights in the method! **Your claim that "we could still obtain a distributed gradient descent algorithms with weighted average of gradients on the server side and the same improvement in the theoretical results" is simply false, and a trivial calculation reveals this.** So, it is not the case that "adding communication compression and EF21 seems too deliberate to me and prevents the paper from presenting the core idea in a clean and crispy manner.", as you claim. **The core idea does not exist without communication compression.**
>
> **This criticism is invalid, and trivially so, we would appreciate if you could retract it.**
>
> We wish to draw the attention of the other reviewers and the AC to this criticism, which we find entirely unfounded.

---

> ### Author Response · Authors · 2023-11-22
> **Weakness 2**
>
> > The experiments seem too simple and small to me. I mean, the linear/logistic regression models on libsvm datasets could be easily trained in a short time on some modern and cheap hardware such as a normal laptop, which doesn't require distributed training or communication at all. I understand these experiments are meant to verify the theoretical results, but the experiment settings are just too simple and synthetic, which makes the experiments seem meaningless to me.
>
> **We very strongly disagree.** We will explain why:
> - Our key result is the last result in the chain Theorem 2 -> Theorem 3 -> Theorem 4, progressing through EF21 applied with client cloning (Theorem 2), to EF21 with weights in the algorithm (Theorem 3), and finally to EF21 with weights in the proof (Theorem 4). This result shows that the state-of-the-art theoretical result for error feedback, which has a dependence on $L_{QM}$, can be improved to a dependence on the smaller (and potentially orders-of-magnitude-smaller) quantity $L_{AM}$. **So, our key result is an improvement of a theoretical complexity of a known method.** The only difference between EF21 captured by our Theorem 4 and the classical result is in an improved stepsize, which directly translates into an improved complexity.
> - **However, note that error feedback, both in the EF14 and EF21 forms are *already* used in practice, and they are used with large fine-tuned stepsizes. There is no need for us to redo all these experiments which were already done in prior works. Please note there are nearly 10 years of research into error feedback (EF14 was developed in 2014), and error feedback is know to work well in practice! Our key claim is completely orthogonal to a practicality claim, which we take for granted.**
> - So, as in any good theoretical paper, **our experimental results are designed to support the key findings  = our theory**. And our experiments do precisely that. We play with multiple (yes, fairly small) datasets, and in each measure $L_{AM}$ and $L_{QM}$. Our theory predicts large improvements in cases when $L_{AM} \ll L_{QM}$, and small or no improvements of these quantities are nearly equal. And this is precisely what we observe. None of this necessitates the need for very large datasets; and there is no reason to believe something would break if $d$ or $n$ increased. Our key contributions can be clearly and conclusively supported with small experiments, and this is also cheaper and more friendly to the environment.
> - **In summary, our experiments are exactly opposite to meaningless. They are designed well to support our theory, and they do so.** They are *not* designed to make claims which are orthogonal to our contributions, and which were made and confirmed many times in prior works over the last decade -- claims that error feedback works in practice on large problems. This is well know and we do not need to replicate these results. In fact, we believe it would be meaningless to do so.
> - We have experiments with $n=100$, $n = 1000$, and $n= 2000$. What is "small" is subjective. We were limited to available hardware and prototype/simulator implementation. However, the $n\approx 100$ is typical in papers, and we are already beyond this with simulation across $n=2000$ clients. To carry and reproduce synthetic experiments with $n=100,000$ clients is extremely time and compute demanding and also expensive. It is of course possible to increase $n$ and $d$ and run larger experiments. These quantities increase the costs of iterations from Theorem 2, but fundamentally they do not change the situation.
>
>
> We wish to draw the attention of the other reviewers and the AC to this criticism, which we find entirely unfounded.

---

> > ### Comment · Reviewer_JbXd · 2023-11-22
> >
> > I agree with the authors that this criticism is unjustified. Given prior research, it is not necessary to go out of their ways to provide large scale experiments to make points about the general behavior of error feedback algorithms if the points verifying the theory can be made also with smaller scale experiments.

---

> ### Author Response · Authors · 2023-11-22
> **Weakness 3**
>
> > The proposed algorithm, EF21-W is far from practical, since in real-world complicated models and training tasks, typically it is very difficult to obtain the smoothness constants, while the smoothness constants are essential for EF21-W.
>
> - Please note that even if it *was* hard to estimate the local smoothness constants $L_1, \dots,L_n$, you ignore the fact that standard EF21 enjoys the same rate through our new analysis! Indeed, you seemed to have completely missed our Theorem 4. There is no need for weights in the algorithm -- the weights appear in the analysis only! We make this very clear in the paper. Note also that this is our main contribution: it is the final step in our chain of results from Theorem 2 (EF21 with client cloning), Theorem 3 (EF21 with weights in the algorithm) and Theorem 4 (EF21 with weights in the proofs). The criticism you raise does not apply to our main result, which you seem to ignore.
>
> **We thus kindly request that thus criticism be retracted; it is unfair as it ignores our main result which does not suffer from such issues.**
>
> - Having said that, we claim that whenever it is possible to estimate Lipschitz constants (virtually all gradient-based methods -- apart from a few adaptive methods such as AdaGrad and Adam rely on such estimations), it is possible in our setup as well: one just does it on each client before EF21-W runs.
>
> **So, you are not criticizing our work -- your criticism applies virtually to all works in gradient-type methods; that is, you are criticizing an entire field. This is unfair to us. If this kind of criticism was allowed to make accept/reject decisions, the field of gradient based optimization would be almost entirely erased. For this reason, we believe that this criticism is unfair.**
>
> We wish to draw the attention of the other reviewers and the AC to this criticism, which we find very misleading and unfair.

---

> > ### Comment · Reviewer_JbXd · 2023-11-23
> >
> > I agree with the authors that this criticism is not appropriate.

---

> ### Author Response · Authors · 2023-11-22
> **Questions**
>
> > 1. If we simply re-weight the gradients with $w_i$ in the gradient averaging without using communication compression or EF21, would we obtain the same improvement in the theoretical results (from quadratic mean to arithmetic mean of the smoothness constants)?
>
> We addressed this in our response to Weakness 1. Your observation is incorrect. This is not a weakness of our paper, but a misunderstanding by the reviewer.
>
> > 2. For the more complicated models or tasks, such as neural networks, how to obtain the smoothness constants and apply EF21-W?
>
> We addressed this in our response to Weakness 3. Note that our main method is EF21, covered by our new Theorem 4. There is no need for such estimation there. We agree setting weights in EF21-W (Theorem 3) more complicated to not setting any weights in EF21 (Theorem 4). Moreover, we agree that cloning (Theorem 2) may be often impractical (e.g., in federated learning). This is precisely why we progressed in our research from Theorem 2, to Theorem 3 to Theorem 4, which is our main result.
>
> Moreover, EF14 and EF21 already are used in practice, on very large models, including on LLMs. See the original work of Seide et al (2014) which in its abstract already in 2014 said:
> - *"For a typical Switchboard DNN with 46M parameters, we reach computation speeds of 27k frames per second (kfps) when using 2880 samples per minibatch, and 51kfps with 16k, on a server with 8 K20X GPUs. This corresponds to speed-ups over a single GPU of 3.6 and 6.3, respectively. 7 training passes over 309h of data complete in under 7h. A 160M-parameter model training processes 3300h of data in under 16h on 20 dual-GPU servers—a 10 times speed-up—albeit at a small accuracy loss."*
> - Moreover, EF21 was used as one component of a slightly more complicated method by J Wang, Y Lu, B Yuan, B Chen, P Liang, C De Sa, C Re, C Zhang in their paper "CocktailSGD: Fine-tuning Foundation Models over 500Mbps Networks", ICML 2023, to fine-tune LLMs in a geographically distributed setup.
>
> There are hundreds of papers using error feedback in practice. Our work is theoretical: by improving the dependence from $L_{QM}$ to $L_{AM}$, we obtain the current SOTA in theory. We need not prove the practicality of a method which is known to work well in practice.

---

### Official Review · Reviewer_JbXd · 2023-11-03

**Soundness:** 4 excellent
**Presentation:** 4 excellent
**Contribution:** 3 good
**Rating:** 6
**Confidence:** 4

**Summary:**

Error Feedback (EF) uses contractive compressors to improve distributed training methods. EF has been around for almost a decade(Siede et al 2014), and suitable theory had been developed for its variant EF21 (Richtárik et al 2021). In this work, the authors improve the analysis of EF21 by allowing for larger step sizes through a refined analysis of varying smoothness constants of the different client loss, leading to an improvement from the quadratic mean to the arithmetic mean of the respective constants.
They describe how they came up with this idea, and by doing this they propose a variant of EF21 called EF21-W which weights the different client gradient inversely proportional to their smoothness constant. They also provide n improved analysis of EF21 is achieved by extending an analogous analysis of EF21-W.
In the supplementary material they extend the algorithm to more advanced versions like EF21 SGD which uses stochastic gradients instead of gradients and EF21 PP which requires partial participation of clients. The authors also provide experimental results for logistic regression with non-convex regularization. They show significant improvement for largely varying smoothness constants and very limited improvements for more balanced cases.

**Strengths:**

The paper is very well written, in particular, the narrative that led the authors to the results is instructive and plausible. The claimed convergence theory is rather exhaustive and applies under quite general assumptions (Assumptions 1-3 only require bounds on the smoothness constants and a lower bound on the optimal function value; a version of the results that leads to linear convergence uses Assumption 4, Polyak-Lojasiewicz inequality, which is a rather weak condition for linear convergence). The improvement in the admissible stepsize for EF21 will be potentially relevant for distributed optimization with largely varying smoothness constants. Furthermore, the authors also provide versions of their algorithms in the stochastic setting (EF21-SGD) and a setting of partial participation (EF-21PP).

**Weaknesses:**

From their theory and experiments, it remains unclear whether the Improvement of the results are coming from the provably larger admissible step size or from different weighting of the clients (as in EF21-W) -- the experiments only seem to compare EF21 with the original stepsize to EF21-W with the improved stepsize (similarly, EF21-W-SGD vs. EF21-SGD).
Furthermore, the novelty of the underlying ideas is not sufficiently discussed. In particular, the usage of differently weighted clients’ gradients based on Lipschitz constants seems to be very related to similar ideas used in importance sampling fro stochastic optimization. (e.g., Zhao, P., & Zhang, T., "Stochastic optimization with importance sampling for regularized loss minimization", ICML 2015). Ideally, a discussion of the connection of their work with this body of literature should be part of the paper.

**Questions:**

- Can you provide experiments that clarify whether the improvement of the results in the setting of largely varying smoothness constants is coming from larger admissible step size or from the different weighting of the clients?
- Please include a discussion of the connection between the idea underlying EF21-W and importance sampling.

---

> ### Author Response · Authors · 2023-11-22
> **Re "Strengths"**
>
> Thanks for appreciating our results!

---

> ### Author Response · Authors · 2023-11-22
> **Re "Weaknesses" - part 1**
>
> > From their theory and experiments, it remains unclear whether the Improvement of the results are coming from the provably larger admissible step size or from different weighting of the clients (as in EF21-W) -- the experiments only seem to compare EF21 with the original stepsize to EF21-W with the improved stepsize (similarly, EF21-W-SGD vs. EF21-SGD).
>
> We disagree that it is unclear whether the theoretical improvements come from provably larger admissible stepsize or from a different weighting of the clients. Please note we obtain essentially (i.e., up to small differences) the same theoretical results in all our three approaches: the client cloning approach described in Section 2.2 and summarized in Theorem 2, the client weighting approach described in Section 2.3 and summarized in Theorem 3, and the "weights in the analysis" approach described in Section 2.4 and summarized in Theorem 4. In all three cases, the stepsize depends on $L_{AM}$ as opposed to $L_{QM}$, and this translates into complexity that depends on this smaller quantity. So, in all three our approaches, the improvement is ultimately due to a larger stepsize. However, these three approaches are very different.
>
> - Indeed, as we describe in the paper, in the cloning approach we apply standard EF21 to a "cloning reformulation" (9) of the underlying problem (1), i.e., we need to work with more than $n$ clients (but no more than $2n$ clients). So, the original source of the speedup here is the fact that we use more clients, and do so in a smart way.
>
> - In the weighting approach we solve the original formulation (1), but use a new method, which we call EF21-W. So, the original source of the speedup here is the smart "double" weighting scheme: we first apply inverse weights to the gradient on each client before compression, and then aggregate the compressed messages in a weighted fashion.
>
> - In the last approach, we also solve the original formulation (1), but apply the vanilla EF21 method. So, there is no change in the algorithm at all (apart from the larger admissible stepsize). The improvement here is solely in the mathematical analysis of the algorithm, which allows for a larger stepsize to be used.
>
> We believe that all of this is clear from a careful reading of the theorems and the accompanying text.
>
> In our experiments, we decided to compare EF21-W (Sec 2.3; a new method proposed in this paper; with its large stepsize) to standard EF21 (with classical stepsize). We *could* have compared any of the three approaches against the standard EF21 (with classical stepsize) approach, but since all three enjoy the same theoretical result, we deemed it sufficient to choose a single representative instead. We can easily include the first (Sec 2.2) and/or the third (Sec 2.4) approach as well, but the empirical trajectories would be almost indistinguishable.
>
> We hope this resolves your concern. We are happy to add this explanation to the paper as well for increased clarity.

---

> ### Author Response · Authors · 2023-11-22
> **Re "Weaknesses" - part 2**
>
> > Furthermore, the novelty of the underlying ideas is not sufficiently discussed. In particular, the usage of differently weighted clients’ gradients based on Lipschitz constants seems to be very related to similar ideas used in importance sampling fro stochastic optimization. (e.g., Zhao, P., & Zhang, T., "Stochastic optimization with importance sampling for regularized loss minimization", ICML 2015). Ideally, a discussion of the connection of their work with this body of literature should be part of the paper.
>
> We described our discovery story as it indeed happened. Our story started with a numerical insight similar to the one described in Example 1 (Section 2.1), which motivated the general client cloning approach described in Section 2.2, and so on. We did not rely on any known results or approaches from other papers. Having said that, it might be possible some of these or similar ideas appeared in some other contexts in some papers we do not know about. To the best of our knowledge, this is not the case.
>
> We are familiar with importance sampling techniques (and have contributed to this research in the past), but we do not see a connection between the weighting approach from Algorithm 2 and importance sampling. First, notice that we do not sample clients: all clients participate in each communication round. So, our approach is not in any direct way related to sampling, let alone importance sampling.
>
> Recall that in the strongly convex regime, the rate of SGD with uniform sampling (let's call it SGD-US) applied to (1) depends on $\max_i L_i$, while with importance sampling (let's call it SGD-IS) the rate depends on $\frac{1}{n}\sum_{i=1}^n L_i$ (see Gower et al, "SGD: general analysis and improved rates", ICML 2019).  The importance sampling probabilities that achieve this are $p_i = L_i/\sum_i L_j$. So, importance sampling can be seen as a trick to improve the dependence of SGD on the smoothness parameters.
>
> In our work, importance sampling as a trick is not useful. Indeed, unlike the SGD scenario, which makes most sense in a *sequential* setting, we work in a *parallel* setting with all $n$ workers available to work at all times in parallel. In this setting, GD is superior to both SGD-US and SGD-IS. Since all $n$ workers are available to work at all times, in parallel, it would be inefficient from a resource point of view to *not* use them all. In distributed training, it is communication complexity that matters. Just like sampling as a trick makes sense in sequential settings, (possibly randomized) gradient compression makes sense in parallel settings. This is why communication compression is used in EF14 and EF21. Just like SGD can improve dramatically over GD in a sequential setting, EF14 and EF21 improve dramatically over GD in the parallel setting.
>
> In the parallel setting, the best performing method that can work with contractive compressors is EF21, and depends on $L_{QM}$. In some sense, in EF21 all of the workers are treated "equally". This is analogous (but not the same!) to SGD-US depending on $\max_i L_i$ since it treats all data samples "equally".
>
> **Someone familiar with importance sampling as a technique for improving this dependence of SGD on the smoothness parameters may wonder whether there might be an analogous technique that could improve the dependence of EF21 on the smoothness constants from $L_{QM}$ to some smaller quantity. Our method EF21-W is an answer to this question! The dependence is reduced to $L_{AM}$.**
>
> **The new technique discovered in our paper could perhaps be called "importance weighting".**
>
> **So, we conclude that EF21-W being an "importance-weighting" variant of EF21 is analogous to SGD-IS being an "importance sampling" variant of SGD-US.**
>
> We believe the relation is that of "analogy" rather than a direct one where one technique can be deduced from the other via some reduction.

---

> ### Author Response · Authors · 2023-11-22
> **Questions - part 1**
>
> > Can you provide experiments that clarify whether the improvement of the results in the setting of largely varying smoothness constants is coming from larger admissible step size or from the different weighting of the clients?
>
> We handled this in detail in our response "Re "Weaknesses" - part 1".

---

> ### Author Response · Authors · 2023-11-22
> **Questions - part 2**
>
> > Please include a discussion of the connection between the idea underlying EF21-W and importance sampling.
>
> In the camera-ready version of the paper, we will include a discussion similar to the one we included in our post "Weaknesses" - part 2.

---

> ### Comment · Reviewer_JbXd · 2023-11-23
>
> Thank you for your reply.
>
> What remains to be unclear for me is the question whether there is any benefit in using EF21-W instead of EF21 if both were to use the enlarged stepsizes based on $L_{AM}$ instead of $L_{QM}$. An exploration of this point in experiments remains to be desirable form my perspective. Furthermore, comparing Theorem 3 and Theorem 4, they are not _exactly_ the same as the constant $G_0$ is slightly different in both cases.
>
> Maybe you can showcase the benefit of EF21-W over EF21 (both with large stepsizes) in the case of non-positively homogenous compressors, when they are not equivalent? I am not an expert of federated learning, which is why I am not aware of the question whether the positive homogeneity assumption is restrictive or not.
>
> Otherwise, it would make it in my opinion a cleaner narrative and message of the paper if it was clear whether EF21-W brings any benefits compared to EF21 in practice or not. This does not do any harm to the improved theory that you obtain for EF21, which I acknowledge as a good scientific contribution.

---

> > ### Author Response · Authors · 2023-11-23
> > **EF21-W vs EF21 with larger stepsize**
> >
> > Thanks for engaging with us.
> >
> > > What remains to be unclear for me is the question whether there is any benefit in using EF21-W instead of EF21 if both were to use the enlarged stepsizes based on  instead of . An exploration of this point in experiments remains to be desirable form my perspective. Furthermore, comparing Theorem 3 and Theorem 4, they are not exactly the same as the constant  is slightly different in both cases. Maybe you can showcase the benefit of EF21-W over EF21 (both with large stepsizes) in the case of non-positively homogenous compressors, when they are not equivalent? I am not an expert of federated learning, which is why I am not aware of the question whether the positive homogeneity assumption is restrictive or not.
> >
> > Indeed, we agree.
> >
> > We focused on experiments with a positively homogeneous compressor (TopK) in which case both these approaches are equivalent, so there is no difference there. But for compressors that do not have this property, the situation may be different. This is a good point. In fact, we briefly considered adding such experiments when writing the paper, but ultimately decided not to do so since we thought this would be a relatively minor investigation due to the fact that most practical compressors used in practice are positively homogeneous. For example, the sparsifiers RandK and TopK are positively homogeneous for any K. Moreover, all typical quantization strategies, which operate by first normalizing the vector to be compressed, followed by randomized rounding of the absolute value of each entry to one of the $l$ levels $0 = s_1 < s_2 < \cdots < s_l = 1$ subdividing the interval $[0,1]$, are positively homogeneous by construction as well.
> >
> > An example of a compressor that is not positively homogeneous is the "natural compressor" from Horvath et al, Natural compression for distributed deep learning, Mathematical and Scientific Machine Learning, 2022, which randomly rounds each entry of the vector to a (positive or negative) power of two. We will include an experiment comparing EF21-W (Theorem 3) and EF21 with large stepsizes (Theorem 4) using this compressor in the camera ready version of the paper.
> >
> > > Otherwise, it would make it in my opinion a cleaner narrative and message of the paper if it was clear whether EF21-W brings any benefits compared to EF21 in practice or not.
> >
> > This is a good point; we shall do so.
> >
> > > This does not do any harm to the improved theory that you obtain for EF21, which I acknowledge as a good scientific contribution.
> >
> > We agree, and thanks for the nice words.

---

> ### Comment · Reviewer_JbXd · 2023-11-23
>
> It is potentially instructive that you presented your results in the way your obtained them. However, for a comprehensive scientific contribution, it is important to discuss similar results and proof techniques as they might have been used in a different context.
> I disagree with the comment that similar proof techniques have not arisen in the literature on important sampling.
> It is clear that you are not applying sampling in your approach, but on the other hand, you apply importance weighting, which doesn't change the fact that there are similarities between known proofs for stochastic optimization with importance sampling and your proof.
> For example, in Corollary 2 of https://arxiv.org/pdf/1401.2753.pdf, importance sampling with probabilities based on Lipschitz constants similar to your weights $w_i$ lead to an improvement of the dependence of a bound from quadratic mean of the Lipschitz constants to the arithmetic mean (cf. Remark after Corollary 2).
>
> I strongly recommend you to acknowledge this if the paper is to be accepted.
>
> As a side comment, it does not matter for your argument if you previously contributed to importance sampling techniques in the past. To keep the anonymity of this review process, it would have actually been better if you had refrained from this remark.

---

> > ### Author Response · Authors · 2023-11-23
> > **Importance sampling vs importance weighting**
> >
> > > It is potentially instructive that you presented your results in the way your obtained them.
> >
> > Thanks, we enjoyed writing the paper this way and hope our readers will enjoy reading it.
> >
> > > However, for a comprehensive scientific contribution, it is important to discuss similar results and proof techniques as they might have been used in a different context.
> >
> > Of course; we agree and are happy to acknowledge all closely related works.
> >
> > > I disagree with the comment that similar proof techniques have not arisen in the literature on important sampling. It is clear that you are not applying sampling in your approach, but on the other hand, you apply importance weighting, which doesn't change the fact that there are similarities between known proofs for stochastic optimization with importance sampling and your proof.
> > For example, in Corollary 2 of https://arxiv.org/pdf/1401.2753.pdf, importance sampling with probabilities based on Lipschitz constants similar to your weights $w_i$ lead to an improvement of the dependence of a bound from quadratic mean of the Lipschitz constants to the arithmetic mean (cf. Remark after Corollary 2).
> >
> > We have looked at the remark (and also at their equation (4)), thanks for pointing this out to us. Indeed, this is something we should and will cite, there is some similarity here.
> >
> > The "abstract" calculation (we elaborate on this below) they do in equation (4) is the same calculation we do when introducing the cloning idea in the paragraph just before Theorem 2. However, they do it in very different context - they want to minimize the variance of a stochastic gradient estimator by adjusting sampling probabilities, and we want to minimize the quadratic mean of the smoothness constants of our cloning reformulation (9) of the original problem (1) {because this quadratic mean appears in the original complexity result for EF21} by adjusting the client cloning weights. While our setups and motivations are very different, for some reason (a coincidence?) both of these goals can be formalized via the same abstract optimization problem of the form
> >
> > $$\min \{ \sum_{i=1}^n \frac{a_i^2}{w_i} \quad \text{subject to} \quad w_1\geq 0,\dots,w_n\geq 0; \sum_{i=1}^n w_i =1 \},$$
> > where $a_i>0$ are some positive constants. In their paper, $w_i$ are sampling probabilities, and $a_i$ are the norms of the sample gradients evaluated at some iteration $t$. In our case, $w_i$ are the cloning weights, and $a_i$ are the Lipschitz constants of the client loss functions.
> >
> > We will cite this similarity in our paper in the appropriate place -- the paragraph immediately preceding Theorem 2.
> >
> > In summary, this abstract minimization problem indeed appears in both our work and theirs; thanks for pointing this out. However, other than that, the algorithms and proof techniques are entirely different. For example, we do not sample, they do, we compress gradients, they do not, our gradient estimator is biased, theirs is unbiased, and so on. Moreover, besides the similarity related to the above abstract optimization problem you point out (which we shall clearly mention), we do not see any resemblance between our lemmas / proofs and the lemmas and proofs in the Zhao-Zhang paper. At this point it is not clear to us whether there is a deeper connection between their importance sampling and our importance weighting approach than this connection, but it is certainly something worth investigating.
> >
> > > I strongly recommend you to acknowledge this if the paper is to be accepted.
> >
> > Absolutely. We now see what you meany before; it was not clear to us why you pointed out the Zhao-Zhang paper before.
> >
> > > As a side comment, it does not matter for your argument if you previously contributed to importance sampling techniques in the past. To keep the anonymity of this review process, it would have actually been better if you had refrained from this remark.
> >
> > Point taken. What we really wanted to say is that while we are familiar with importance sampling literature, we do not see a connection between our proof techniques and the proof techniques in that literature. You were originally suggesting there was a connection but did not say what it was, and since we do not know of any such connection despite knowing the literature, we said what we said. But now that you elaborated on what you really meant (remark after Corollary 2), we see a certain connection which is indeed worth describing and citing, and we shall do so.

---

### Meta-Review · Area_Chair_2oAX · 2023-12-08

**Metareview:**

The paper considers distributed optimization settings with gradient compression and the error feedback (EF) mechanism. The main contribution of the paper is in the improvement of the dependence of the convergence bound on the smoothness constants from $\|\|\mathbf{L}\|\|_2/\sqrt{n}$ to $\|\|\mathbf{L}\|\|_1/n,$ where $\mathbf{L} = (L_1, L_2, \dots, L_n)$ is the vector of component smoothness constants. By the relationship between the $p$-norms, when $\mathbf{L}$ is highly non-uniform, 1- and 2-norm are close in value, so the bound is tighter by a factor of the order $\sqrt{n}.$ When $\mathbf{L}$ is close to uniform, the 1-norm is larger by a factor of the order $\sqrt{n}$ than the 2-norm, and so these two constants are of the same order in this case. The paper starts from proposing a new variant of EF to obtain these tighter bounds and then in the process manages to obtain improved, matching results for the original EF mechanism, with the improvements being the result of a tighter analysis that allows for a larger step size. The theoretical results are corroborated by numerical experiments.

The paper is written clearly and explains the main ideas well. The discussion with the reviewers also revealed an interesting parallel between importance weighting used in the present work and the importance sampling in stochastic optimization, which the authors promised to discuss in the camera-ready. Overall, the paper provides a solid contribution to distributed optimization.

**Justification For Why Not Higher Score:**

I am ok with bumping up, but I didn't get the impression of a big idea/result to recommend spotlight.

**Justification For Why Not Lower Score:**

There is certainly enough contribution to accept the paper.

---

### Decision · Program_Chairs · 2024-01-16

Accept (poster)